# Modelling decadal trends and the impact of extreme events on carbon fluxes in a deciduous temperate forest using the QUINCY model

Tea Thum[1], Tuuli Miinalainen[1], Outi Seppälä[1], Holly Croft[2], Cheryl Rogers[3], Ralf Staebler[4], Silvia Caldararu[5], and Sönke Zaehle[6]

[1]Finnish Meteorological Institute, P.O. Box 503, 00101 Helsinki, Finland
[2]University of Sheffield, Western Bank, Sheffield S10 2TN, The United Kingdom
[3]Toronto Metropolitan University, Jorgenson Hall, 350 Victoria Street Toronto, ON M5B 2K3, Canada
[4]Environment and Climate Change Canada, 867 Lakeshore Rd Burlington ON L7S 1A1, Canada
[5]Trinity College, College Green, Dublin 2, Ireland
[6]Max Planck Institute for Biogeochemistry, Hans-Knöll-Straße 10, 07745 Jena, Germany

**Correspondence:** Tea Thum (tea.thum@fmi.fi)

**Abstract.** Changing climatic conditions pose a challenge to accurately estimate the carbon sequestration potential of terrestrial vegetation, which is often mediated by Nitrogen availability. The close coupling between the Nitrogen and Carbon cycles controls plant productivity and shapes the structure and functional dynamics of ecosystems. However, how carbon and nitrogen interactions affect both carbon fluxes and plant functional traits in dynamic ecotones, which are experiencing biotic and abiotic

changes, remains unclear. In this work, we use in-situ measurements of leaf chlorophyll content ($Chl_{Leaf}$, years 2013-2016) and leaf area index (LAI, years 1998-2018) to parameterise the seasonal dynamics of the QUINCY ('QUantifying Interactions between terrestrial Nutrient CYcles and the climate system') terrestrial biosphere model (TBM) to simulate the carbon fluxes at the Borden Forest Research Station flux tower site, Ontario, Canada, over 22 years from 1996-2018. Our goals are to assess the additional value of using $Chl_{Leaf}$ in the model parametrization, to study how well QUINCY can capture observed trends

related to the carbon cycle at the site, and investigate how well the processes associated with a drought year and its legacy effects are captured by the model.

QUINCY was able to simulate leaf-level maximum carboxylation capacity ($V_{c(max),25}$), $Chl_{Leaf}$ and leaf nitrogen quite consistent with observations. The model with the improved parameterization captured observed daily gross primary production (GPP) well ($r^2$=0.80, root mean square error (RMSE)=2.2 $\mu mol\,m^{-2}\,s^{-1}$). Nevertheless, we found that although observed GPP

increased significantly during the study period (22.4 $gC\,m^2\,yr^{-1}\,yr^{-1}$), and net ecosystem exchange (NEE) shifted towards a stronger sink, these trends were not captured in the model. Instead, QUINCY showed a significant increasing trend for total ecosystem respiration (TER), that was not present in the observations. The severe drought in 2007 affected observed carbon fluxes strongly, lowering both GPP and TER also in the following year. QUINCY was able to capture some of the decrease in GPP and TER in 2007. However, the legacy effect of the drought in 2008 was not captured by the model. These results call for

further work on representing legacy effects in TBMs, as these can have long-lasting impacts on ecosystem functioning.

## 1   Introduction

Climate change impacts the exchange of carbon (C), water and energy between vegetation and the atmosphere, as well as the biogeochemical cycles and carbon storage potential of ecosystems (Canadell et al., 2022). The future land carbon balance can be predicted by the terrestrial biosphere models (TBMs), which are versatile tools for studying the effects of climate on biogeochemical cycles (Blyth et al., 2021). However, TBMs exhibit significant inconsistencies in their simulated results over space and time due to divergent representations of important biogeochemical processes (O'Sullivan et al., 2022). One way to address these inconsistencies and understand the most important development needs is to confront these models with data to gain a better understanding of their performance. The nitrogen (N) cycle related variables and long-term observations are particularly relevant (Zaehle and Dalmonech, 2011), as the role of the N cycle can constrain the C storage capacity of terrestrial ecosystems (Zaehle et al., 2015) and long-term observations can provide information on how changing climate affects the ecosystem. These long-term observations may also include anomalous years, such as years with droughts, one of the major stressors of extreme events that can have profound impacts on the carbon cycle (Piao et al., 2019). Droughts can also have legacy effects for years to come and these legacy effects can vary between forests according to species and structure (Yu et al., 2022).

The C and N cycles are closely interconnected, with N being a significant component of plants and a vital macronutrient. N is required for growth, development and metabolic processes, and is a fundamental constituent of DNA and various plant structural and photosynthetic components, such as light harvesting complexes and the electron transport chain (ETC). Additionally, it is an integral component of many enzymes involved in the Calvin cycle. N deficit therefore limits photosynthesis, which ultimately decreases plant productivity (LeBauer and Treseder, 2008). The future carbon pools and budgets in the coming decades depend, in part, on the N cycle and availability of N to vegetation (Arora et al., 2020; Huntingford et al., 2022; Zaehle, 2013).

In recent decades, many TBMs have incorporated elements of the N cycle, to varying degrees (Thornton et al., 2009; Sokolov et al., 2008; Zaehle and Friend, 2010). However, this inclusion presents many challenges, e.g. in terms of N uptake (Davies-Barnard et al., 2022) and losses (Meyerholt and Zaehle, 2018) and particularly in how models represent the N limitation of photosynthesis (Medlyn et al., 2015; Thomas et al., 2015; Walker et al., 2021). N limitation may directly affect photosynthesis rates or its effects may be buffered via different stoichiometric related implementations (Thomas et al., 2015) and the different hypotheses and parameter values related to N cycle processes lead to differences between models (Medlyn et al., 2015). Various modelling approaches to the N cycle have resulted in different ecosystem responses e.g. to carbon dioxide ($CO_2$) fertilization (Arora et al., 2020; Meyerholt et al., 2020a; Thomas et al., 2013). A model intercomparison study of five CMIP6 models showed a wide range of response in net primary productivity for increased atmospheric $CO_2$ and atmospheric N deposition (Davies-Barnard et al., 2020).

Given the importance of N in physiological, biochemical and structural processes, novel data sources linked to the N cycle and its connections to the C cycle are highly needed to help in model development (Kou-Giesbrecht et al., 2023; Meyerholt

et al., 2020b). Remotely-sensed observations of ecosystem traits are an important means of parameterising the temporal dynamics of these models in a spatially-explicit manner (Rogers et al., 2017). Satellite observations have shown that the LAI has increased globally and has contributed to an increase in the land carbon sink (Chen et al., 2019) and change in the Bowen ratio of the energy fluxes (Forzieri et al., 2020). Simultaneously, lowering of leaf N content has been observed through intensive long-term monitoring plots in European forests (Jonard et al., 2015). Climate change-induced changes led to increases in leaf area index (LAI) (Chen et al., 2019) and tree productivity (Jonard et al., 2015), and the changes in N availability and demand have led to declining N availability relative to demand in terrestrial ecosystems (Mason et al., 2022).

One candidate that brings N observations and remote sensing together is leaf chlorophyll content ($Chl_{Leaf}$), which can be used as a proxy of the photosynthetic N component (Croft et al., 2017) and can be accurately retrieved at ecologically-relevant time and space resolution from remote sensing data, due to the presence of large absorption features in spectral bands typically sampled by optical sensors (Croft and Chen, 2018). There is a large body of literature on leaf chlorophyll retrieval from remote sensing data (Sims and Gamon, 2002; Dash and Curran, 2004), leading to the creation of large-scale national and global products spanning several years (e.g. Croft et al. (2020)). Most TBMs simulate photosynthesis through the Farquhar-von Caemmerer-Berry (FcB) kinetic enzyme model (Farquhar et al., 1980), where photosynthetic capacity is represented by the maximum carboxylation capacity (normalised to 25 °C; $V_{c(max),25}$) and the maximum rate of electron transport (normalised to 25 °C; $J_{(max),25}$). Integrating physiological information through $Chl_{Leaf}$ data has led to developments to improve modelling gross primary productivity (GPP). For example, Houborg et al. (2013) developed a semi-empirical relationship between $Chl_{Leaf}$ and $V_{c(max),25}$ and used remotely sensed $Chl_{Leaf}$ to replace $V_{c(max),25}$ in the CLM model to improve simulations of GPP for a maize field. Lu et al. (2022) used observations of $Chl_{Leaf}$ and $V_{c(max),25}$ to create plant functional type (PFT)-dependent linear relationships and successfully retrieved the $V_{c(max),25}$ parameter at several ecosystems. At the site level, at the Borden Forest Research Station (hereafter referred to as Borden Forest), Luo et al. (2018) improved GPP modelling by directly linking the seasonal cycle of the $V_{c(max),25}$ parameter to $Chl_{Leaf}$. Our present study differs from previous literature in that we model $Chl_{Leaf}$ explicitly, and it is based on a predicted nitrogen cycle.

In this research, we use the long time series of in situ LAI and $Chl_{Leaf}$ observations at Borden Forest together with eddy covariance observations of carbon fluxes over a decadal time scale. The long-term nature of the continuous flux record at Borden Forest with 22 years of near-continuous data provides an almost unparalleled opportunity to examine longer-term trends in ecosystem processes at a deciduous forest site due to warming temperature against a background of temperature and drought variability and extremes and their legacy effects. The seasonal cycle of LAI has several implications to the climate-vegetation exchange, including influencing canopy conductance and fluxes of water, energy and carbon dioxide (Richardson et al., 2013). The development of $Chl_{Leaf}$ and canopy structural parameters, such as LAI decouple during the shoulder seasons in deciduous forests, necessitating the separate parameterisation of leaf-level physiological processes and LAI in TBMs (Croft et al., 2015). Continuous long-term ground-based observations of LAI at site scale (Rogers et al., 2021) provide a means to assess these phenomenon in one forest.

In our research, the observational data are combined with a terrestrial biosphere model QUINCY (QUantifying Interactions between terrestrial Nutrient CYcles and the climate system) (Thum et al., 2019). QUINCY simulates fully coupled carbon,

nitrogen, phosphorus cycles of terrestrial ecosystems along with water and energy budgets. QUINCY is one of the few TBMs that explicitly represents relationships between time-varying foliar nitrogen content, which vary given the ecosystems nutrient availability and carbon-uptake capacity, and $Chl_{Leaf}$ and photosynthetic activity (such as $V_{c(max),25}$). The model treats the impact of leaf chlorophyll and its vertical distribution on leaf- and canopy-level photosynthesis using an extension of the FcB model (Kull and Kruijt, 1998). QUINCY includes a representation of plant growth separating sink and source processes, acclimation of many ecophysiological processes to meteorological and/or nutrient availability and explicit representation of vertical soil processes. Whilst the impacts of climatic events and longer-term climatic shifts are complex to model, we hypothesize that QUINCY can capture changes in ecosystem function if they are related to meteorological conditions and atmospheric $CO_2$. Additionally QUINCY has potential to capture legacy effects of extreme events via its carbohydrate pool structure. This improved modelling capacity will enable to us better understand the nitrogen and carbon cycles under both episodic events, and over inter-annual and decadal timescales in a temperate Deciduous-Boreal ecotone. The diverse observations available at the site allow us to evaluate the model's performance in several aspects. The following four research questions are addressed in this work:

- How does the decoupling of LAI and $Chl_{Leaf}$ seasonal development in the model affect the estimation of annual carbon fluxes?

- Is the QUINCY model able to simulate any long-term changes in seasonal shifts in carbon fluxes and LAI values?

- Is there a nitrogen constraint on carbon fluxes at Borden Forest, and does it change over the 23-year period?

- Can the QUINCY model simulate the effects of drought events on the carbon cycle?

## 2  Materials and methods

### 2.1  Study site

Borden Forest (44° 19′ N, 79° 56′ W) is a mixed forest situated in Southern Ontario, Canada. This forest is located in the Great Lakes-St. Lawrence forest ecotone, which is a transition zone that includes both southern temperate forest species and northern boreal species (Froelich et al., 2015). Based on the 2006 vegetation survey (Teklemariam et al., 2009), the forest species composition was primarily composed of red maple (*Acer rubrum*) and eastern white pine (*Pinus strobus*) with 52% and 14% respectively. Other species included large-tooth aspen (*Populus grandidentata*, 8%) white ash (*Fraxinus americana,* 7%), and trembling aspen (*Populus tremuloides*, 3%). The forest has 15-20% evergreen coniferous vegetation. The understory consists of short ferns, small shrubs and saplings (Halliday, 2010). The forest has been naturally regrown from farmland that was abandoned in the early 20th century, with a canopy height of approximately 22 m (Froelich et al., 2015). The soil consists mainly of sand with a thin layer of organic matter. The mean annual temperature at the site is 7.4 °C and mean annual precipitation is 784 mm over 2000-2014 period (Froelich et al., 2015). The site is a member of the AmeriFlux network (site-ID: CA-Cbo).

## 2.2 Site level observations

Site-level measurements of carbon fluxes, meteorology and soil moisture, LAI, leaf nitrogen and $Chl_{Leaf}$ and biochemical model parameters ($J_{(max),25}$ and $V_{c(max),25}$) were used in the study. In this study, we define LAI as half of the total (all-sided) leaf area per unit of ground area (Chen and Black, 1992). Three data sets were used, two of which were long-term: i) carbon fluxes, meteorology (1996-2018) and ii) LAI (1998-2018), along with a leaf-level biochemical dataset from 2013 to 2016. Meteorological data was used to force our model simulations, while soil moisture and temperature observations (from years 2005-2015) were used to evaluate the model's performance.

### 2.2.1 Net ecosystem exchange (NEE) of $CO_2$, meteorological and soil moisture observations

$CO_2$ flux data from half-hourly eddy covariance measurements sampled at Borden Forest tower at 33 m height between 1996 and 2018 were used for model evaluation. The instrumentation is described in detail in Froelich et al. (2015). No observations were made in 2004 due to instrument and tower replacement. The fluxes were determined on a half-hourly time scale using a program developed at SUNY Albany (Froelich et al., 2015) up to 2013, and using EddyPro (Fratini and Mauder, 2014) thereafter. The vegetation remains uninterrupted from 1.5-4 km towards the southeast and southwest and 1 km towards the northeast. However, there is a cropland less than 400 m in the northwest direction. Data for wind directions between $285°$ and $20°$ were excluded from analysis due to insufficient fetch and gap-filled (Luo et al., 2018). Also observations recorded when the friction velocity was less than $0.3\,\mathrm{m\,s^{-1}}$ were removed according to Froelich et al. (2015), and the data were gap-filled and the measured carbon dioxide net ecosystem exchange (NEE) flux was partitioned into half-hourly GPP and total ecosystem respiration (TER) according to Barr et al. (2004). This is the standard method used for the Fluxnet-Canada sites (Pierrat et al., 2021) and has been used in the previous publications from this site. This procedure first derives the component fluxes from the NEE and then uses simple empirical models constrained by the measured data (Barr et al., 2004; Rogers, 2022). The other empirical relationship is between TER and soil temperature at shallow depth, and the other is between GPP and photosynthetically active radiation (PAR) above the canopy, including also a time-varying parameter (Barr et al., 2004; Rogers, 2022). These time-varying parameters are determined using a flexible moving window approach, including 100 points (Barr et al., 2004; Rogers, 2022). The exact formulations used are as in Rogers (2022).

Air temperature, relative humidity, air pressure, longwave and shortwave radiation, wind speed and direction were also measured by instruments on the flux tower (Froelich et al., 2015). For the air temperature, relative humidity, radiation and wind speed we used observations made at 42 meters, for air pressure observations at 2 meters. Soil temperature (from 5 to 100 cm depth) and profiles (depths from 2 to 100 cm) were measured at two locations Froelich et al. (2015), one located 40 m southwest of the flux tower, the other one was located 50 m west. Precipitation data was obtained from the nearby Egbert weather station ($44°\,23'$ N, $79°\,78'$ W), which has provided hourly observations since 2014. Prior to 2014, the hourly precipitation was obtained from the ERA5-Land product (Muñoz Sabater et al., 2021) and scaled to match the average precipitation values as estimated from the hourly Egbert observations (more information in S1). In addition, the ERA5-Land product was used in the gapfilling of observed meteorological data.

### 2.2.2 LAI observations

We used a daily LAI time series of 1999-2018 from Rogers et al. (2021), estimated from photosynthetically active radiation (PAR) observations collected above and below the canopy. PAR was measured above the canopy by LI-COR LI-190SA (LI-COR, Lincoln Nebraska) sensor and below the canopy with a LI-COR LI-191 sensor. We estimated daily LAI values from half-hourly observations of above-canopy and transmitted PAR using the Miller integral (Miller, 1967), as recommended by Rogers et al. (2021). To improve spatial representativeness of the daily LAI estimates at the site, the values were then calibrated to match effective LAI ($L_e$) measured along a 100m transect using a handheld LI-COR LAI-2000 plant canopy analyzer (LI-COR, Lincoln Nebraska) using a linear relationship. True LAI was estimated from all the observations as:

$$LAI = \frac{[(1-\alpha)L_e\gamma_E]}{\Omega_E} \tag{1}$$

where $\alpha$ is the ratio of woody area to total area and $\gamma_E$ is the ratio of needle area to shoot area (taken as unity in deciduous forests). The value of $\alpha$ (0.17) was taken from the literature (Gower et al., 1999), similar to Croft et al. (2015). The clumping index ($\Omega_E$) is 0.95, was measured using a TRAC (Tracing Radiation and Architecture of Canopies) instrument (Huiming Instrumentation Limited, Nanjing, China) (Rogers et al., 2021).

### 2.2.3 Leaf-level trait measurements

In situ leaf-level data included measurements of $Chl_{Leaf}$ and nitrogen content, maximum electron transport and carboxylation capacities ($J_{max,25}$, $V_{c(max),25}$), and specific leaf area (SLA). The measurements were collected at an average interval of 9 days during the growing seasons (day of year 130-290) from 2013 to 2016. In 2013, only leaf chlorophyll and SLA data were collected. In 2014, all six variables were measured. In the following years, only chlorophyll, $J_{max,25}$ and $V_{c(max),25}$ were measured. For these biochemical measurements, leaves were sampled from the top of the canopy from the flux tower. Leaf-level gas exchange measurements were made using a LI-6400 portable infrared gas analyzer (LI-COR, Lincoln, Nebraska, USA) (Croft et al., 2017). $Chl_{Leaf}$ and leaf nitrogen content were destructively analyzed in the laboratory from five sampled leaves per measurement date. The methodology and the measurements are described in detail in Croft et al. (2013, 2014) and Wellburn (1994). The SLA was calculated as the ratio of leaf area to leaf dry mass. In this study, we used a species-weighted canopy average of the leaf-level parameters, based on the species composition of the forest (i.e. red maple 60.4 %, large-tooth aspen 12.9 %, trembling aspen 12.4 % and ash 14.2 %) (Croft et al., 2015). The values for the different tree species are given in Table S2 and the calculation of the species-weighted average is explained in Section S2.

## 2.3 The QUINCY model

### 2.3.1 General description

The QUINCY model (Thum et al., 2019) was used to simulate ecosystem functioning in the study area. QUINCY simulates fully coupled carbon, nitrogen and phosphorus cycles as well as water and energy balances in vegetation and soil on a half-

hourly time scale. Here we give a brief description of the model, focusing on the parts relevant to this study. A more detailed description can be found in Thum et al. (2019).

Vegetation is grouped by Plant Functional Type (PFT), and represented as an average individual composed of structural pools (leaves, sapwood, heartwood, coarse roots, fine roots and fruits), a labile pool (a fast overturning and respiring non-structural pool) and a reserve pool (seasonal, non-respiring and non-structural storage pool). The non-structural pools represent storage pools for non-structural carbohydrates and associated nutrients. Trees are also characterized by height (m), diameter (m) and stand density ($m^{-2}$). The tree canopy is composed of ten canopy layers, which increase in depth (of LAI) exponentially with

layer depth ($LAI_{cl}$). Photosynthesis and stomatal conductance are calculated separately for the sunlit and shaded leaves in each layer, as estimated by a radiative transfer model (see below) (Zaehle and Friend, 2010) using a Farquhar model based scheme from Kull and Kruijt (1998). According to this scheme, the role of leaf chlorophyll is explicitly taken into account in the photosynthesis calculation and determines the proportion of the leaf area at each canopy layer that is light-saturated. The photosynthesis for the non-light-saturated part is calculated using the light-limited rate of photosynthesis (relying on the

$J_{max,25}$ parameter). The photosynthesis in the light-saturated part is calculated as a co-limited rate of the electron transport capacity (determined using $J_{max,25}$) and carboxylation capacity (via Rubisco; determined using $V_{c(max,25)}$). Therefore both biochemical model parameters and leaf chlorophyll influence photosynthesis. Determination of these variables from leaf nitrogen is described in Section 2.3.3. The leaf stoichiometry can be set fixed or it can be dynamic, when it is varied in response to nutrient demand and supply.

Stomatal conductance is modelled after Medlyn et al. (2011) and in addition to stomatal conductance, soil moisture can limit photosynthesis directly through a modifier in calculation of the biochemical parameters (Egea et al., 2011). Photosynthesis can also be down-regulated by the sink limitation (example in Fig. S1 in Thum et al. (2019)).

     Maintenance respiration is a linear function of the N content of each pool. Temperature acclimation for photosynthesis is as in Friend (2010) and for maintenance respiration as in Atkin et al. (2014). Tissue growth is defined by allometric equations, and

the allometric relationship between leaves and fine roots responds to N and water limitation by increasing the uptake capacity under nutrient limitation.

     Soil biogeochemistry is largely based on a CENTURY-style (Parton et al., 1993) approach, except that the vertical soil profile of biogeochemical pools, including metabolic, structural and woody litter as well as fast and slow overturning soil organic matter (SOM), is explicitly represented. Each soil layer also includes N pools of ammonium ($NH_4$) and nitrate ($NO_3$).

The soil profile consists of 15 layers, reaching a depth of 9.5 meters. The depth of each layer layer increases exponentially as it goes deeper. The stoichiometry of the litter is determined by the stoichiometry of the plant pool it comes from. The fast pool's stoichiometry is dependent on the availability of inorganic nutrients, while the slow pool has a fixed stoichiometry. Plants and microbes compete for the nutrients based on their respective demand and uptake capacity. $NH_4$ is oxidized to $NO_3$ through nitrification in the aerobic part of the soil and $NO_3$ is reduced to diatomic nitrogen $N_2$ through denitrification in the anaerobic

part of the soil (Zaehle et al., 2011). Both processes also produce nitrogen oxide $NO_y$ and nitrous oxide $N_2O$. Biological nitrogen fixation (BNF) is considered as an asymbiotic and symbiotic process (Meyerholt et al., 2016).

Soil temperature and moisture are calculated for each layer based on soil physical characteristics, as well as the transport and atmospheric exchange of energy and water. The radiative transfer scheme has been developed following the two-stream approach by Spitters (1986) and the original implementation to terrestrial biosphere model OCN (Zaehle and Friend, 2010) has been extended to include diagnostic canopy albedo, clumping and attenuation of the shortwave backscatter from the soil. The radiative transfer is calculated separately for the visible and near-infrared radiation bands. It estimates the sunlit and shaded leaves for each canopy layer and separates the incoming radiation into direct and diffuse components. Snow dynamics is a later addition to the model (Lacroix et al., 2022) and the model can be run with or without the snow. The snow dynamics are represented in a five-layer scheme that accounts for flows of heat and water within and between the snow layers.

### 2.3.2 Phenology of the deciduous trees

The seasonal development of leaf biomass is affected by the plant's ability to grow new tissues and the fractional allocation to plant organs. The start and end of the growing season are determined by meteorological and soil moisture values, which are averaged over seven days to mitigate the impact of daily climate variability. The start of the growing season is determined by the accumulated growing degree days ($GDD_{acc}$), which represents the current number of growing degree days above the temperature threshold, $t_{air}^{GDD}$, since the beginning of last dormancy period) as:

$$GDD_{acc} > GDD_{req}^{max} \times exp^{-k_{dormancy}^{GDD} \times NDD}, where \tag{2a}$$

$$\frac{GDD_{acc}}{dt} = GDD_{acc} + MAX(t_{air} - t_{tair}^{GDD}, 0.0) \tag{2b}$$

$NDD$ is the number of dormancy days, taken as days since the last growing season, and $k_{dormancy}^{GDD}$ (value 0.007 $days^{-1}$) relates dormancy to the maximum growing degree-day requirement ($GDD_{req}^{max}$, 800 degree-days) to account for the chilling requirements of the buds (Krinner et al., 2005), and $dt$ denotes the time step in days. The growing season ends when the decreasing average air temperature falls below the temperature threshold of ($t_{air}^{sen}$, 8.5 °C) and then senescence occurs.

### 2.3.3 Leaf N partitioning

While the overall amount of leaf nitrogen is largely driven by phenological development, the leaf N concentration per leaf area ($N_{leaf}$) responds to soil nutrient availability, as plants take up mineral N from the soil pools. This uptake is determined by the amount of N in each soil pool and fine root biomass, and is further modulated by plant N demand. Leaf nitrogen has a vertical gradient that decreases exponentially towards the bottom of the canopy, in accordance with observations (Niinemets et al., 1998). Leaf N in each layer ($N_{leaf,cl}$) is divided into structural and photosynthetic parts (Friend et al., 1997). The fraction of structural N ($fN_{struc,cl}$) is calculated for each canopy layer as a function of the total leaf N in the respective layer (Zaehle and Friend, 2010):

$$fN_{struc,cl} = k_0^{struc} - k_1^{struc} N_{leaf,cl} \tag{3}$$

$k_0^{struc}$ is the maximum fraction of structural leaf N (0.63 for deciduous forest (Friend et al., 1997; Kattge et al., 2011)) and $k_1^{struc}$ is the slope of structural leaf N with total N ($7.14 \times 10^3$ g$^{-1}$N) (Friend et al., 1997) .

The photosynthetic N pools have three compartments: Rubisco associated ($fN_{rub}$), electron transport associated ($fN_{et}$), and chlorophyll associated ($fN_{chl}$). The photosynthetic fractions all have a role in the calculation of photosynthesis (Kull and Kruijt, 1998). The fractions are used directly in the calculation of the photosynthetic parameters $V_{c(max),25}$ and $J_{max,25}$, where the leaf N in each content is multiplied with these fractions and some other modifiers (equations S7 and S10 in Thum et al. (2019)). According to Zaehle and Friend (2010), the fraction of leaf N in chlorophyll is calculated to increase with canopy depth:

$$fN_{chl} = \frac{k_0^{chl} - k_1^{chl} e^{-k_{fn}^{chl} LAI_c}}{a_{chl}^n}, \tag{4}$$

where $k_0^{chl}$ (value 6.0 (Zaehle and Friend, 2010)), $k_1^{chl}$ (value 3.6 (Zaehle and Friend, 2010)) and $k_{fn}^{chl}$ (value 0.7 (Friend, 2001)) are empirical parameters. $a_{chl}^n$ is the molecular N content of chlorophyll (25.12 $\frac{mol}{mmol^{-1}}$ (Evans, 1989)). $LAI_c$ is the cumulative leaf area.

The photosynthetic parameters $V_{c(max),25}$ and $J_{max,25}$ are assumed to have a fixed ratio of 1.97 (Wullschleger, 1993). Based on this ratio, $fN_{rub}$ and $fN_{et}$ are calculated, using the calculated values of the structural and chlorophyll fractions.

## 2.3.4 Modelling protocol

The QUINCY model requires half-hourly meteorological forcing, including short and longwave radiation, air temperature, precipitation, air pressure, humidity and wind speed as well as atmospheric $CO_2$ concentration (hereafter referred to as [$CO_2$]), and N and P deposition rates. Meteorological forcing was measured at the site and the [$CO_2$] was obtained from Friedlingstein et al. (2019) and N deposition data from Lamarque et al. (2010, 2011). The Borden forest is described as a broadleaf deciduous forest PFT and we performed point scale simulations.

To obtain a near-equilibrium state of soil and vegetation, the model was spun up for 500 years, using atmospheric $CO_2$ concentration from a randomly selected year from the 1901-1930 period and a random year of observed meteorological data. This was followed by a transient simulation starting in 1901, which was using meteorological data randomly picked from the observed meteorology. The transient simulation used the atmospheric [$CO_2$] and N deposition values derived from data sources mentioned above and, from 1996 onwards, the measured site-level meteorology for the respective years. For the purposes of this study, we ran simulations where only the C or both the C and N cycles were active. In the simulation where only C was active, the plants had access to all the N that they needed. The P concentration was kept at a level, where it did not limit plant uptake or SOM decomposition. The temperature response of the BNF (Bytnerowicz et al., 2022) was set to have an optimum temperature of 18 °C, replacing the default value of 32 °C. The default value is based on observations in the tropics and the shape of the curve predicts very low BNF for more northern regions with the default optimum temperature. Lowering this value to typical air temperature at the Borden site provides more realistic BNF for the site and assumes local temperature acclimation. The SLA is a constant value of 320 cm g$^{-1}$ for the broadleaf deciduous forest PFT in the model simulations. To compare to

**Table 1.** Abbreviations used for the model runs.

| Abbreviation | Explanation |
|---|---|
| C:orig | C cycle only enabled model simulation with original parameters, with dynamic leaf stoichiometry |
| C:LAI | C cycle only enabled model simulation with parameter tuning based on LAI, with dynamic leaf stoichiometry |
| C:LAI&chl | On top of C:LAI simulation also parameter tuning based on $Chl_{Leaf}$, with dynamic leaf stoichiometry |
| Cfix:orig | C cycle only enabled model simulation with original parameters, with fixed leaf stoichiometry |
| Cfix:LAI | C cycle only enabled model simulation with parameter tuning based on LAI, with fixed leaf stoichiometry |
| Cfix:LAI&chl | On top of C:LAI simulation also parameter tuning based on $Chl_{Leaf}$, with fixed leaf stoichiometry |
| CN:orig | Simulation including N cycle with original parameters |
| CN:LAI | Simulation including N cycle with parameter tuning based on LAI |
| CN:LAI&chl | On top of CN:LAI simulation also parameter tuning based on $Chl_{Leaf}$ |
| CN:LAI&chl&snow | On top of CN:LAI&chl simulation also snow model turned on |

the leaf level observations ($Chl_{Leaf}$, $V_{c(max),25}$, $J_{max,25}$) made at the top of the canopy, we used only the top canopy layer values from the model.

The model was run with several different parametrizations to study the influence of using LAI and $Chl_{Leaf}$ on the parameterisation and how their decoupling influences the results. For the carbon cycle-only (C) simulation, we show results from the original model formulation (orig), then with the simulation using LAI to tune phenology (LAI tuning, C:LAI) and finally a simulation using both LAI and $Chl_{Leaf}$ for tuning (C:LAI&chl). These simulations have been done with dynamic leaf stoichiometry, that is basically showing forest at N saturation. This simulation is shown in the first sections of this paper, since the

nitrogen cycle enabled version showed too low GPP at the site and we also wanted to show the influence of model tuning with LAI and $Chl_{Leaf}$ with GPP levels comparable to the observations. For comprehensiveness, we also report the values for the carbon cycle-only simulations with fixed stoichiometry, which reflects the C cycle with average N availability, but with no N limitation on growth and soil processes. We made an additional simulation of adding snow (CN:LAI&chl&snow) to assess its influence on the soil temperature and carbon fluxes.

For the C and CN simulation, we use the results after both LAI and $Chl_{Leaf}$ in the main analysis. The abbreviations of the model simulations are found in Table 1. A schematic figure (Fig. S1) showing the work flow of the study, with the parameter tuning and then comparison to the observations is found in the SI.

The original parameter values and the tuned parameter values are shown in Table 2. We adjusted the parameters by comparing the modelled LAI and $Chl_{Leaf}$ to the observations (Fig. 1b, e) and tried to match those. The autumn decline of the simulated

LAI was adjusted to match the observations by modifying the parameter controlling leaf senescence ($t_{air}^{sen}$) from the default value of 8.5 °C to 15.0 °C. It is likely that leaf senescence at the site is partially controlled by light availability, a process which is not yet present in the model, therefore the higher temperature threshold is accounting for this missing factor. To adjust the summertime magnitude of $Chl_{Leaf}$ parameters $k_0^{struc}$ (Eq. 3) and $k_0^{chl}$ (Eq. 4) were adjusted. In the model the whole leaf

**Table 2.** Parameter values in different simulations. The unit of the parameter in parenthesis after the parameter name.

| Simulation | $t_{air}^{sen}$ (°C) | $k_0^{struc}$ (-) | $k_0^{chl}$ (-) |
|------------|---------|----------|---------|
| C:orig | 8.5 | 0.63 | 6.0 |
| C:LAI | 15.0 | 0.63 | 6.0 |
| C:LAI&chl | 15.0 | 0.68 | 5.2 |
| Cfix:orig | 8.5 | 0.63 | 6.0 |
| Cfix:LAI | 15.0 | 0.63 | 6.0 |
| Cfix:LAI&chl | 15.0 | 0.50 | 7.0 |
| CN:orig | 8.5 | 0.63 | 6.0 |
| CN:LAI | 15.0 | 0.63 | 6.0 |
| CN:LAI&chl | 15.0 | 0.58 | 6.5 |

nitrogen is initially allocated to the structural nitrogen during the initial stages of the growing season. Subsequently, the amount allocated to the photosynthetic compartments (including leaf chlorophyll, $V_{c(max),25}$ and $J_{max,25}$) begins to increase as the season progresses. The model was modified to incorporate a delay in the transition from structural nitrogen to photosynthetic nitrogen. We added in a delay of 20 days by introducing a leaf age factor to the simulations (equations for this change: Eqs. S2-S4) in the C simulations and delay of 15 days in the Cfix:LAI&chl and CN:LAI&chl -simulations.

### 2.3.5 Estimation of seasonal metrics and trends

We estimated the growing season metrics separately using GPP or LAI. The start and end of the season (SOS and EOS) estimated from GPP were calculated from the first and last pass of the threshold, which was defined as the 30 % of the year's $90^{th}$ percentile value (an example year of 2013 in Fig. S2a). For the LAI the threshold was calculated as being the 20 % of the difference between the summer and winter values, starting from the winter value (Fig. S2b). Length of the growing season (LOS) is the time between SOS and EOS. These calculations were made on smoothed data using an averaging weekly window to minimise anomalies. The trend assessment was carried out with a particular focus on statistically significant trends, which were identified through the application of Student's t-test on slope values obtained from the linear regression (p < 0.05).

## 3 Results

### 3.1 Dynamic parameterisation of LAI and leaf chlorophyll content improves modelled GPP estimates

The average modelled GPP, LAI and $Chl_{Leaf}$ values for the three only C cycle model simulations (C:orig, C:LAI and C:LAI&chl) across a growing season from 1996-2018 are shown alongside the measured data in Fig. 1 a, c, e. The observed GPP starts to increase already after day of year (DOY) 100, whereas in all the simulations, the increase begins later and at a

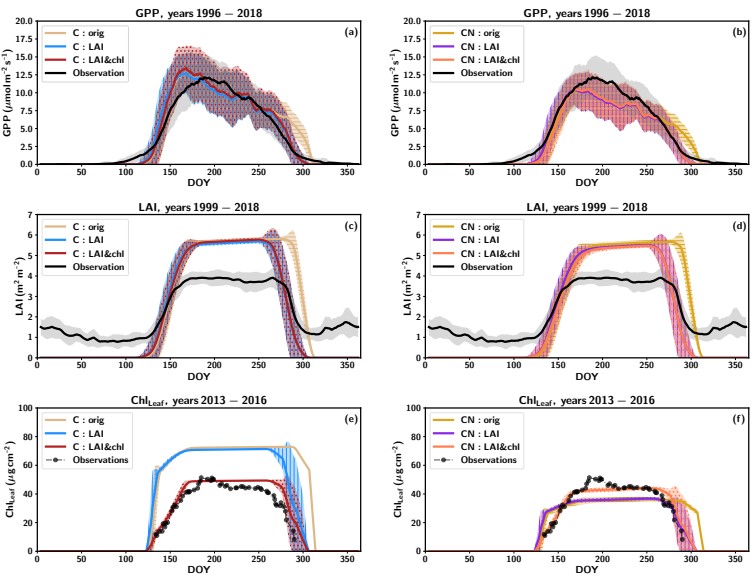

**Figure 1.** Averaged yearly cycles of a) GPP, c) LAI and e) leaf chlorophyll from the C simulations and for the CN-simulations GPP in b), LAI in d) and leaf chlorophyll in f). The shaded regions show the standard deviations between the years. In (a) and (b), the data represents the mean values for the period 1996-2018. In (c) and (d), the data represents the mean values for the period 1999-2018 and in (e) and (f) for the period 2013-2016. The lines have been smoothed with a seven-day averaging window, except for the observed leaf chlorophyll which has been smoothed with a three-day window.

faster rate. The increase to maximum summer time values in the simulations happens rapidly and the maximum summer values occur early in the season, around DOY 160 (early June). The observations show a more gradual decline in increasing GPP before the peak, with maximum summertime values occurring around DOY 200 (mid-July).

At the end of the season, the inclusion of LAI (C:LAI) and $Chl_{Leaf}$ (C:LAI&chl) data improved the representation of senescence at the end of the season, and the consequent decline in GPP, compared to the C:orig simulation (Fig. 1 a, S3). The summertime average was accurately simulated by the model, with an average overestimation of 1.1 % for June, July and August with the C:orig and C:LAI simulations and 5.0 % overestimation with the C:LAI&chl simulation (Fig. 1a). The annual carbon flux values together with root mean square error (RMSE) and R-squared ($r^2$) are shown in Table 3. The different C

model simulations (i.e. C:orig, C:LAI and C:LAI&chl) using dynamic stoichiometry (i.e. not including simulations using fixed stoichiometry, Cfix) did not largely impact RMSE and $r^2$ values (Table 3).

    The simulations with the N constraint on carbon fluxes (CN simulation) demonstrate a reduction in summertime GPP values, with the averaged GPP during July-August underestimated by 14 % compared to observations (Fig. 1 b). The same modification to the phenology parameter was made as in the C simulation, with the objective of improving the fit of the simulated LAI

(simulation CN:LAI). This resulted in a more accurate representation with better timing of senescence of the observed seasonal cycles of GPP, LAI and $Chl_{Leaf}$ in the simulations (Fig. 1 b, d, f, S3).

**Table 3.** Annual averaged carbon fluxes for different model parameterizations, together with their standard deviations, root mean square error (RMSE) and $r^2$ values. RMSE and $r^2$ have been calculated from the daily averages for the time period 1996-2018. Also the percentual discrepancy between the simulation result and observation is shown, positive values denoting overestimation and negative underestimation.

| GPP | Annual value (gC m$^{-2}$yr$^{-1}$) | RMSE (μmol m$^{-2}$s$^{-1}$) | $r^2$ (-) | Over/underestimation (%) |
|---|---|---|---|---|
| Observation | 1459 ± 224 | - | - | - |
| C:orig | 1574 ± 189 | 2.58 | 0.71 | 7.9 |
| C:LAI | 1484 ± 184 | 2.50 | 0.72 | 1.7 |
| C:LAI&chl | 1511 ± 192 | 2.56 | 0.71 | 3.6 |
| Cfix:orig | 1243 ± 148 | 2.26 | 0.78 | -14.8 |
| Cfix:LAI | 1166 ± 144 | 2.22 | 0.78 | -20.1 |
| Cfix:LAI&chl | 1226 ± 158 | 2.19 | 0.79 | -15.9 |
| CN:orig | 1297 ± 171 | 2.23 | 0.78 | -11.1 |
| CN:LAI | 1225 ± 160 | 2.18 | 0.79 | -16.0 |
| CN:LAI&chl | 1209 ± 163 | 2.16 | 0.80 | -17.1 |
| CN:LAI&chl&snow | 1236 ± 174 | 2.22 | 0.78 | -15.3 |

| TER | Annual value (gC m$^{-2}$yr$^{-1}$) | RMSE (μmol m$^{-2}$s$^{-1}$) | $r^2$ (-) | Over/underestimation (%) |
|---|---|---|---|---|
| Observation | 1254 ± 220 | - | - | - |
| C:orig | 1534 ± 132 | 1.77 | 0.59 | 22.3 |
| C:LAI | 1450 ± 130 | 1.64 | 0.65 | 15.6 |
| C:LAI&chl | 1474 ± 133 | 1.68 | 0.63 | 17.5 |
| Cfix:orig | 1199 ± 96.6 | 1.41 | 0.74 | -4.3 |
| Cfix:LAI | 1129 ± 94.9 | 1.44 | 0.73 | -9.9 |
| Cfix:LAI&chl | 1187 ± 101 | 1.43 | 0.74 | -5.4 |
| CN:orig | 1255 ± 107 | 1.40 | 0.75 | 0.0 |
| CN:LAI | 1190 ± 101 | 1.39 | 0.75 | -5.1 |
| CN:LAI&chl | 1174 ± 100 | 1.38 | 0.75 | -6.4 |
| CN:LAI&chl&snow | 1193 ± 111 | 1.40 | 0.75 | -4.9 |

| NEE | Annual value (gC m$^{-2}$yr$^{-1}$) | RMSE (μmol m$^{-2}$s$^{-1}$) | $r^2$ (-) | Over/underestimation (%) |
|---|---|---|---|---|
| Observation | -205 ± 140 | - | - | - |
| C:orig | -40 ± 106 | 2.08 | 0.37 | -80.5 |
| C:LAI | -35 ± 100 | 2.01 | 0.42 | -82.9 |
| C:LAI&chl | -37 ± 106 | 2.06 | 0.39 | -82.0 |
| Cfix:orig | -44 ± 91 | 1.88 | 0.49 | -78.5 |
| Cfix:LAI | -37 ± 87 | 1.73 | 0.57 | -82.0 |
| Cfix:LAI&chl | -39 ± 95 | 1.81 | 0.53 | -80.1 |
| CN:orig | -42 ± 97 | 1.88 | 0.49 | -79.5 |
| CN:LAI | -36 ± 91 | 1.76 | 0.55 | -82.4 |
| CN:LAI&chl | -35 ± 94 | 1.78 | 0.54 | -82.9 |
| CN:LAI&chl&snow | -43 ± 99 | 1.85 | 0.50 | -79.0 |

The use of the LAI in the model tuning had a more pronounced impact on the GPP fluxes in both the C and CN simulations because of shortening the growing season, even though the changes in $r^2$ (order of 0.01) or RMSE remain relatively minor. After the LAI and $Chl_{Leaf}$ tuning, the simulation had a 3.6 % overestimation in the annual GPP in the C simulations and an underestimation by 17 % in the CN simulations (Table 3). The impact of the N constraint on the annual GPP was 302 $gC\,m^{-2}\,yr^{-1}$ (Table 3), representing a 20 % decrease in the annual GPP value relative to the C simulation. When considering the Cfix:LAI&chl simulation, the estimated GPP was very similar to the CN-simulations, with only a 1.4 % larger value (Table 3).

From this point onward in the paper, when we refer to the C simulations, we refer to the results from the simulations with the LAI and $Chl_{Leaf}$ tuning (C:LAI&chl), and similarly for CN simulations. CN:LAI&chl was the most successful of the simulations in terms of r$^2$ and RMSE when compared against observed GPP (Table 3).

## 3.2 Other carbon fluxes: TER and NEE

TER was decreased compared to the original simulation after altering the parameters (Table 2) for both C and CN-simulations (Table 3). This is connected to the declining GPP, as the litter input influences the amount of soil carbon. The seasonal cycle was not influenced by these changes (Fig. S4). The CN simulations had better r$^2$ values for TER than the C simulations (Table 3). This occurred because the magnitude of TER was better captured by these simulations (Fig. S5 b). The most pronounced underestimation of the TER in the CN simulations by the model compared to the observations occurs during the summer months July and August (Fig. 2b, S5 b).

The observed annual TER was overestimated by the C simulation by 18 % (Table 3). The CN-simulation yielded a more accurate representation of the annual estimate(Table 3). The Cfix:LAI&chl simulation gave similar values to the CN-simulation (Table 3). The nitrogen cycle was found to constrain annual TER by 300 $gC\,m^{-2}\,yr^{-1}$, representing a 20 % decrease from the C simulation.

The observations indicated that the forest acted as a sink, with a net carbon uptake of -205 $\pm$ 140 $gC\,m^{-2}\,yr^{-1}$ over the measurement period. The simulations indicated that it was a weak sink, for both the C and the CN simulations, with very small difference (Table 3). However, it should be noted that the interannual variation in the simulations was considerable. Steep increase in GPP in the C simulations led to a short-term overestimation of the ecosystem carbon sink in early summer (Fig. S5c, S6), but generally during the summer the ecosystem carbon sink was underestimated in the simulations. Despite the GPP summertime magnitude being underestimated in the CN-simulations, the performance of the model as estimated by the r$^2$ and RMSE, was for NEE and the component fluxes better with the CN than C simulations (r$^2$ better by 0.08 for GPP, 0.12 for TER, 0.15 for NEE) (Table 3, Fig. S5).

The early season pattern observed in the simulated NEE is attributed to too late onset of GPP and too early onset of heterotrophic respiration (Fig. 2a and b, S7), which is regulated by the soil moisture and soil temperature in the model. QUINCY is simulating these soil conditions based on the meteorological conditions and soil texture. The majority of the simulated heterotrophic respiration originates from the uppermost soil layers, as most of the organic material is located in these layers in the model. The soil temperature of the upmost layer is significantly underestimated during the winter months, and increase

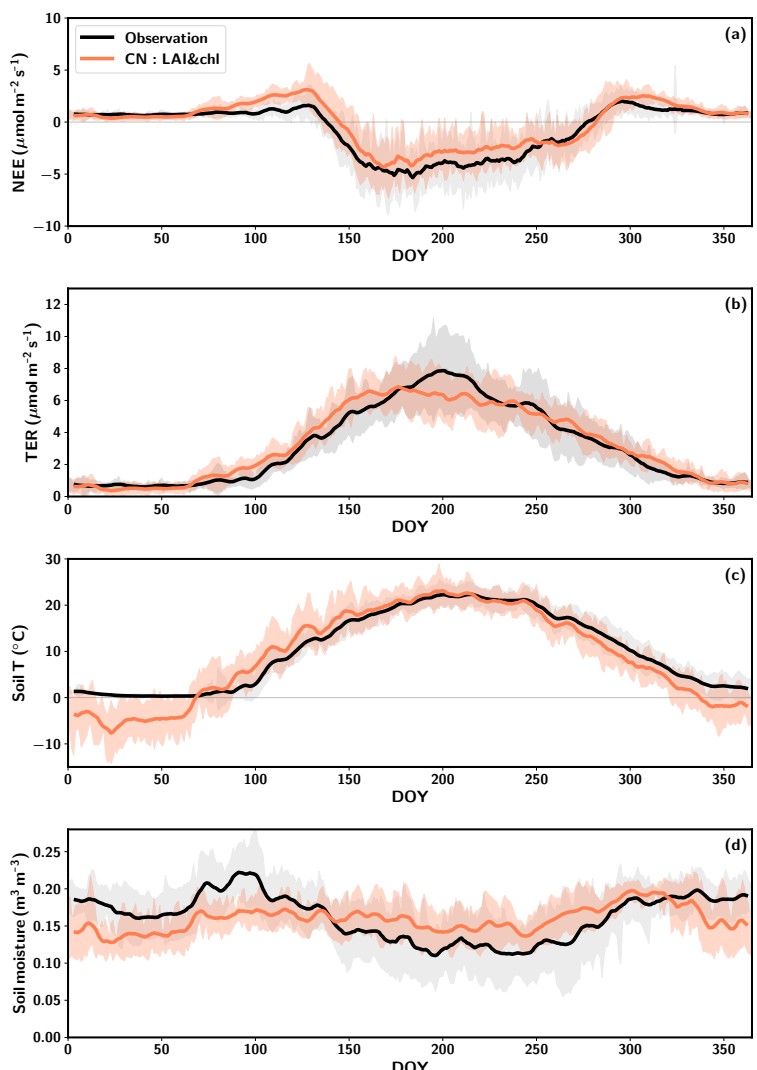

**Figure 2.** Averaged seasonal cycles of net ecosystem exchange, NEE (a), total ecosystem respiration, TER (b), soil temperature (c) and soil moisture both at 5 cm depth (d) averaged over the period 2005-2015. The observations are represented in black, the CN:LAI&chl model results in orange and the standard deviation is shown by the shaded regions. Both the observations and simulations are smoothed with a seven-day window.

to the summertime levels occurs earlier in the simulations than in the observations (Fig. 2c). This phenomenon occurs during the period between DOY 70 and 150. It is during this time that the TER is being overestimated in the spring (Fig 2b, c). The maintenance respiration is also activated during this time period, although its increase is less pronounced due to the temperature acclimation implemented in the model (Fig. S7). In autumn, the simulated drawdown of the TER occurs simultaneously with the observations, despite the simulated soil temperature decreasing at a faster rate than that observed (Fig. 2).

The soil conditions are relevant for estimation of the TER, as heterotrophic respiration is an important part of it (Fig. S7). The too early increase of simulated soil temperatures in spring occurs also at deeper depths (Fig. S8 b, c). Also at deeper levels at 10 and 20 cm the wintertime soil moisture is underestimated (Fig. S8 e, f). The observed summertime variability of soil moisture is better captured by the model in deeper layers than in the 5 cm depth, even though the summertime magnitude is overestimated at 10 cm depth (Fig. S8).

The simulation with snow (CN:LAI&chl&snow) had an influence on the spring soil conditions and thus on the TER (Figs. S8, S9). Overall the simulations successfully captured the observed snow depth, with some underestimation in some years, which also resulted in too early snow clearance dates (Fig. S10). Snow influenced both soil temperature and soil moisture (Fig. S8). In winter, the soil temperatures were colder in the CN:LAI&chl&snow simulation than in the CN:LAI&chl simulation, and despite a fairly synchronous increase from the winter levels, the simulations with snow were slower to reach above-zero levels and therefore better matched the observations (Fig. S8a-c). Soil moisture was also affected, with snow delaying the increase to the summer levels (Fig. S8d-f). At 10 cm depth, the simulation without snow appears to be more consistent with spring observations of soil moisture (Fig. S8e). Snow in the simulations caused a slight delay in the increase of TER during spring (Fig. S9b), and the RMSE of TER during mid-March to mid-April (DOY 74-105) improved by 0.1 $\mu$mol m$^{-2}$ s$^{-1}$ for the 2005-2015 period. Overall, the GPP and TER do not begin to increase to summer levels before snow clearance (Fig. S9).

### 3.3 Simulated structural and biochemical parameters

The continuous observations of LAI provide values of 3.78 $\pm$ 0.43 m$^2$ m$^{-2}$ in summer (averaged over the June-July period, along with the standard deviation). The simulated values were closer to the LAI-2000 observations than to the values obtained from the continuous observations. The LAI-2000 observations have a summertime average of 4.63 $\pm$ 0.71 m$^2$ m$^{-2}$ for years 2013-2018. The C simulation overestimated continuous observations by 39 %, while the overestimation in CN simulations was 26 % (Table 4). The LAI from the CN simulations was closer to the value estimated from the LAI-2000 observations, with only 3 % overestimation. In the model the specific leaf area (SLA) is an important factor in converting the leaf biomass to LAI and is held constant. In the observations the SLA showed a dynamic change, with higher values ( 303 cm g$^{-1}$) observed in the early season and a subsequent decrease to a summer value of 162 cm g$^{-1}$ within about one month.

The C simulation overestimated observed leaf N by approximately threefold (Table 4). The CN-simulation overestimated the observed value by 59 %. The C simulation estimated a 2.5-fold overestimation of observed $J_{max,25}$ (Fig. 3a, Table 4). Both the CN and C simulations overestimated the values of $J_{max,25}$ and $Vc_{c(max),25}$, with considerably higher overestimation taking place in the C-simulations (Fig. 3, Table 4). These high predicted value are not unexpected, given the direct link between plant N and photosynthetic parameters in the model and the implicit unlimited N availability in the C model.

The tuning of the model did not have a pronounced effect on the summertime magnitude of the photosynthetic parameters (Fig. 3), because the changes in the nitrogen allocated to leaf chlorophyll were derived from the structural nitrogen. The springtime delay imposed on the leaf chlorophyll also influenced the photosynthesis parameters, resulting in an improved seasonal cycle compared to observations (Figs. 3).

**Table 4.** Observed and simulated (C:LAI&chl and CN:LAI&chl simulations) leaf traits. Values are estimated for the June-July means with standard deviations. The unit of the parameter in parenthesis after the parameter name. For the LAI the values 1998-2018, for leaf N and the biochemical parameters 2014 and for $Chl_{Leaf}$ 2013-2016.

| Variable (unit) | Observed | Simulation (C:LAI&chl) | Simulation (CN:LAI&chl) |
|---|---|---|---|
| LAI (continuous) $(\mathrm{m^2\,m^{-2}})$ | $3.78 \pm 0.43$ | $5.27 \pm 0.77$ | $4.77 \pm 0.91$ |
| Leaf N $(\mathrm{gm^{-2}})$ | $1.32 \pm 0.13$ | $4.21 \pm 0.13$ | $2.10 \pm 0.09$ |
| $J_{max,25}$ $(\mathrm{\mu mol\,m^{-2}\,s^{-1}})$ | $117.1 \pm 24.2$ | $296.6 \pm 53.5$ | $164.7 \pm 27.1$ |
| $V_{c(max),25}$ $(\mathrm{\mu mol\,m^{-2}\,s^{-1}})$ | $63.5 \pm 14.4$ | $156.8 \pm 28.2$ | $87.1 \pm 14.2$ |
| $Chl_{Leaf}$ $(\mathrm{\mu g\,cm^{-2}})$ | $45.6 \pm 7.9$ | $45.9 \pm 6.1$ | $40.8 \pm 4.0$ |

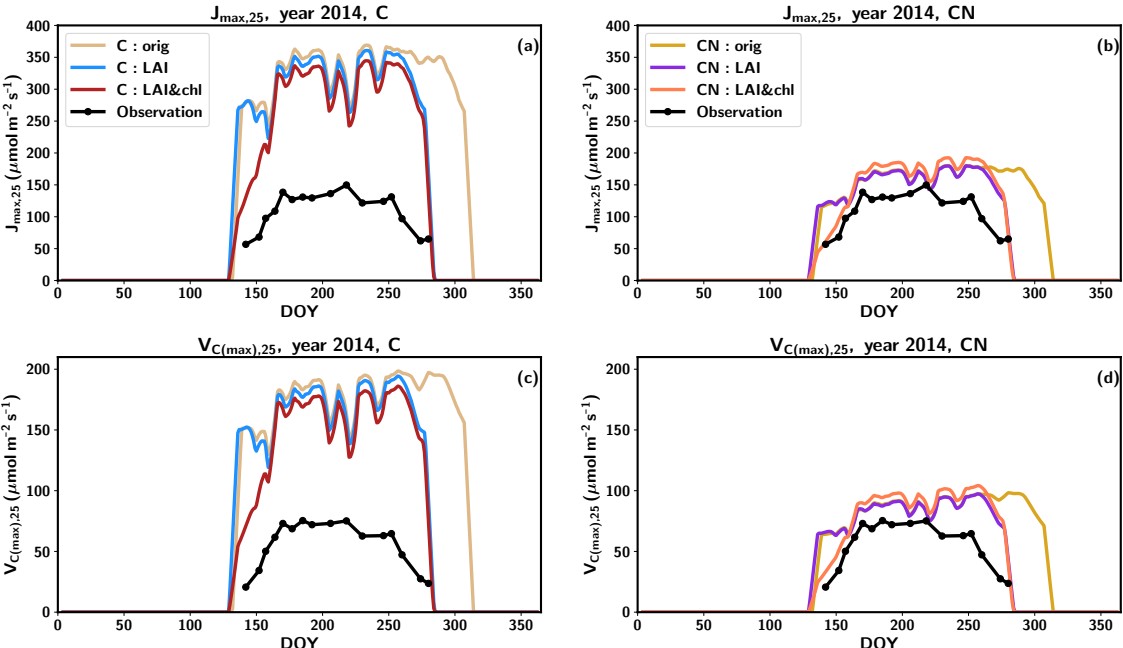

**Figure 3.** The seasonal cycle of $J_{max,25}$ (a, c) and $V_{c(max),25}$ (b, c) in 2014 with C simulations (a, c) and CN simulations (b, d). The observations are represented in black, the original QUINCY results from the C simulations in light brown, with LAI tuning in blue and with both LAI and leaf chlorophyll tuning in dark red. The CN:orig simulation results are in dark yellow, the CN:LAI in magenta and CN:LAI&chl in orange. The modelling results have been averaged with a seven-day smoothing window.

## 3.4 The influence of drought on carbon fluxes

A severe drought occurred at the site in 2007, when the precipitation was approximately 20 % lower than in a regular year, at $608\,\mathrm{mm\,yr^{-1}}$ (Fig. S11). Fig. 4 depicts the averages over the time period 2005-2015 (with the exception of 2007 and 2008)

for GPP, TER and soil moisture at 5 cm, as the soil moisture observations were available for this time period. The following year 2008 was characterized by higher-than average precipitation (923 $\mathrm{mm\,yr^{-1}}$), 21 % above average (Fig. S11).

The annual observed TER in 2007 was clearly below the average annual TER by 37 % (i.e. 794 $\mathrm{gC\,m^{-2}yr^{-1}}$) and the level remained low throughout the summer (Fig. 4a). Furthermore, the summertime maximum TER values in 2008 were below average (Fig. 4b). Additionally, the values of TER exhibited a slower rate of increase and a more rapid decrease to winter levels after mid-summer compared to regular years (Fig. 4b). This resulted in the observed TER for 2008 being 36 % below the averaged annual TER (i.e. 804 $\mathrm{gC\,m^{-2}yr^{-1}}$). In contrast to the measurements, the simulations did not predict low TER for the beginning of the season in 2007 or 2008. Instead, the behaviour in the CN simulations was similar to that observed in other years (Fig. 4a-b). Only, when there was a pronounced decrease in soil moisture around DOY 160 in 2007, decrease in the simulated TER resulted. Following the precipitation event around DOY 200, the TER values exhibited a recovery to typical summertime levels for several days. After this a decline to lower levels occurred. The precipitation events that occurred around DOY 240 resulted in TER returning to a regular level. In 2008 the simulated TER followed pattern similar to regular years. The annual simulated TER was 9 % lower in 2007 compared to regular years and 10 % higher in 2008.

Observed GPP exhibited a decline in 2007, with the annual value being 24 % lower (i.e. 1112 $\mathrm{gC\,m^{-2}yr^{-1}}$) compared to the averaged annual GPP. In 2008, the observed GPP was found to be lower than in 2007, with a value of 1086 $\mathrm{gC\,m^{-2}yr^{-1}}$, representing a 26% decline below the average. A later increase of GPP to summer values in spring and an earlier decrease to winter values were responsible for the reduction in the annual value, with peak season GPP being higher than in 2007 (Fig. 4c-d). The simulated GPP exhibited a regular pattern in both years until DOY 160 in 2007 (Fig. 4c). The annual simulated GPP was 14 % lower in 2007 compared to regular years and 15 % higher in 2008.

The measured soil moisture was consistently below the typical values throughout 2007 and that continued until early summer of 2008 (DOY 150, Fig. 4e). In 2007 the simulated soil moisture was generally at a higher level compared to observations, but showed rather similar responses to precipitation events than observations (Fig. 4e). Overall, the simulated soil moisture exhibits a narrower range than the observed soil moisture data in regular years, and declines less in the drought year than observations.

The model demonstrated a less pronounced effect of drought on the carbon fluxes than was observed in 2007. The water potential of soil was lowering the simulated GPP (Fig. S12), but the drawdown was not large enough (Fig. 4). Furthermore, the likely legacy effects of drought that were observed in 2008 were not replicated by the model. The non-structural carbohydrate pools were affected by the drought. The labile pool was 25 % lower in 2007 than in normal years, but had fully recovered by 2008. The reserve pool was in 2007 at 18 % lower in 2007 than in normal years and was still 10 % lower in 2008. However, this did not significantly affect the LAI values or the annual GPP levels. Because the reduction in GPP was not pronounced enough, the legacy effect was not seen in the TER values. Overall, the simulated soil moisture at 5 cm depth did not reach the lower values in the observed range (Fig. S13 a), so although the simulated carbon fluxes do respond to soil moisture (Fig. S13), the legacy effects were not captured.

In regular years a hysteresis effect of TER versus the soil temperature relationship was observed, with the values in the later half of the year having lower values (Fig. S14a). In drought year 2007 this effect was not visible, with values staying at a low level. In 2008 the hysteresis effect was more pronounced than in 2006. The model simulations were not able to replicate

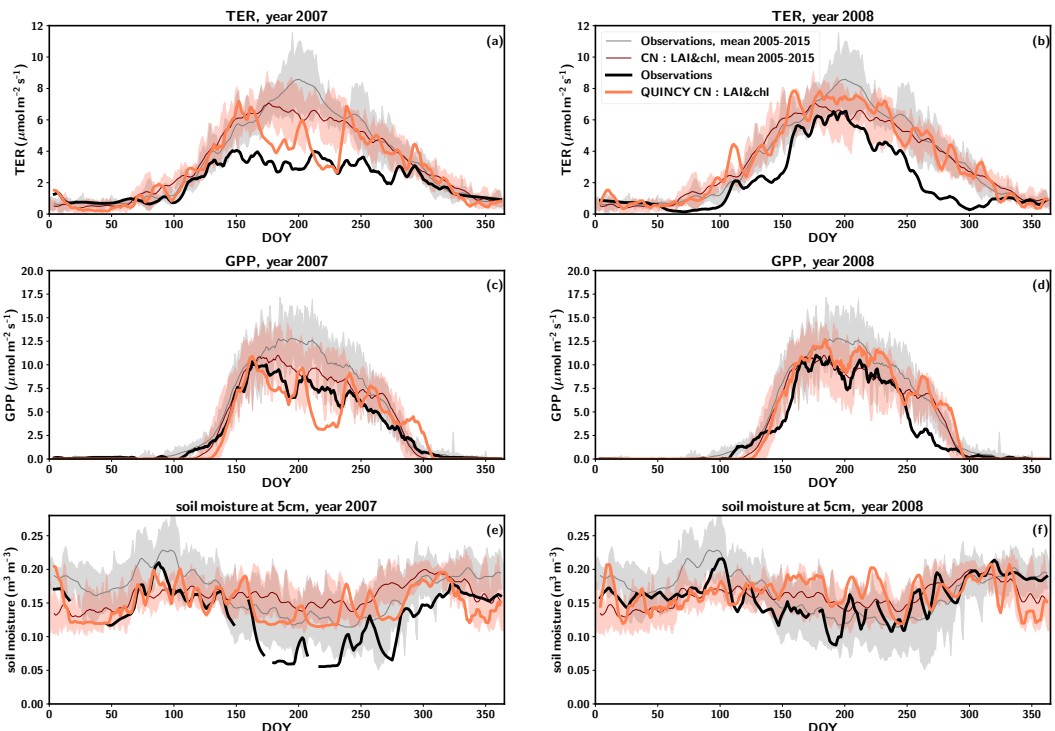

**Figure 4.** The averaged seasonal cycles for observations and CN-simulations for total ecosystem respiration, TER (a in 2007 and b in 2008), gross primary production, GPP (c in 2007 and d in 2008), and soil moisture at 5 cm (e in 2007 and f in 2008). The observations are in thick black line and simulations in thick orange line for these years. The averaged seasonal cycle for 2005-2015 without 2007 and 2008 is in thin gray line and simulations in thin red line. Standard deviation of the averaged seasonal cycle is shown as shaded regions. Both the observations and the model results have been smoothed using a seven-day averaging window.

this kind of behaviour. The observed soil moisture at 5 cm depth did not explain the observed hysteresis effect (data not shown). As a possible explanation to the hysteresis effect, one could think about strong connection between the photosynthesis and respiration. The observations indeed show a higher GPP in early summer months compared to later months in the year, especially in 2008 (Fig. S15). This phenomenon does not take place in the simulations (Fig. S15b, d, f).

### 3.5 Interannual variability and longer term trends in annual carbon fluxes

The growing season metrics were estimated based on both LAI and GPP from the observations and CN:LAI&chl simulation. The start of season (SOS) takes place almost at the same time according to the GPP and LAI observation-based estimates. The end of season (EOS) is estimated to be some days later in LAI based estimates (Table 5). The simulations agree on both observation-based estimates of SOS by one to five days, and there is a larger difference between the observed and simulated EOS estimates. The simulated EOS based on LAI is earlier than observed (Table 5). In the simulations the GPP and LAI are tightly coupled. However, in the observations the EOS estimated from LAI takes place on average 13 days later than the EOS

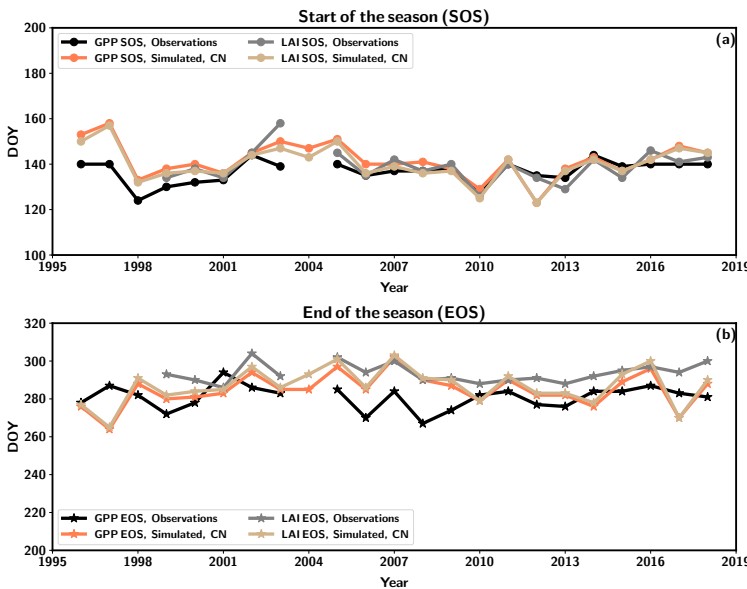

**Figure 5.** The start of season (SOS) (a) and end of season (EOS) (b) as estimated from observed and simulated (CN:LAI&chl) time series of GPP (black observation, orange simulation) and LAI (gray observation, light brown simulation).

according to the GPP. The observed LAI remains high, despite decreasing GPP. LAI is therefore not so tightly coupled to the seasonality of GPP in autumn as it is in the model. C simulations provide similar estimates for these growing season metrics.

The time series of the metrics indicates that the EOS from simulations has a larger range of variability (38 days from GPP estimated EOS) than the observations (27 days from GPP estimated, 18 days from LAI estimated) (Fig. 5b). Additionally, the simulations demonstrated a greater interannual variation in the SOS estimation (range 34 days for LAI-based estimation) than seen in the observations (range 20 days for GPP-based estimation, 32 days for LAI-based estimation) (Fig. 5a). Stronger interannual variability in model estimates of LAI is driven by the phenological parameters governed by air temperature. The

real forest with several species may be more resilient to different environmental conditions, and therefore different species may be able to benefit from different environmental conditions during the shoulder seasons. No discernible trends are evident in any of the time series under consideration. The cold spring of 2017 resulted in the simulation estimates of SOS occurring at a later point in time, although the impact was not as pronounced in the observations of GPP (Fig. 5).

     The next step involved the calculation of trends for the annual values of GPP, TER and NEE (Fig. S16, Table S3) derived

from the CN:LAI&chl simulations. In order to assess the ability of QUINCY to capture observed interannual fluctuations in the annual carbon balance, the changes were investigated over time. This was done with a particular focus on statistically significant trends, which were identified through the application of Student's t-test ($p < 0.05$). During the time period 1996-2018 the observed GPP exhibited a significant increasing trend (Fig. S16, Table S3), with a magnitude of 22.4 $\mathrm{gC\,m^2\,yr^{-1}\,yr^{-1}}$. A significant trend was also present in the summertime and autumn observations (Table S3). QUINCY showed a minor and

non-significant upward trend for GPP (Table S3). When the final five years of the data series were arbitrarily excluded, the

**Table 5.** The average start of season (SOS), end of season (EOS) and length of season (LOS) as determined from the observations and the QUINCY model simulation (CN:LAI&chl) of GPP and LAI.

| Variable | unit | Observations | QUINCY |
|----------|------|--------------|--------|
| SOS (GPP) | DOY | 137 | 142 |
| SOS (LAI) | DOY | 139 | 140 |
| EOS (GPP) | DOY | 281 | 285 |
| EOS (LAI) | DOY | 294 | 287 |
| LOS (GPP) | days | 144 | 143 |
| LOS (LAI) | days | 155 | 147 |

observed GPP trend was no longer statistically significant and had decreased to 11.6 $\mathrm{gC\,m^2\,yr^{-1}\,yr^{-1}}$. This was comparable to the QUINCY estimation for the same period, which was 8.6 $\mathrm{gC\,m^2\,yr^{-1}\,yr^{-1}}$.

The observed TER in the period from 1996 to 2018 showed a non-significant trend of 9.0 $\mathrm{gC\,m^2\,yr^{-1}\,yr^{-1}}$, which was a bit larger than the significant trend seen in TER from QUINCY, 6.9 $\mathrm{gC\,m^2\,yr^{-1}\,yr^{-1}}$. The observations indicated a significant trend for NEE towards a larger sink, with a rate of -13.4 $\mathrm{gC\,m^2\,yr^{-1}\,yr^{-1}}$ over the same time period. The simulations instead proposed an increasing source, with a magnitude of 5.2 $\mathrm{gC\,m^2\,yr^{-1}\,yr^{-1}}$. The continuous observations of summertime LAI (averaged over June, July and August) showed a significant trend of 0.033 $\mathrm{m^2\,m^{-2}\,yr^{-1}}$ (Fig. S16d). The QUINCY model estimated a minor non-significant negative trend (-0.001 $\mathrm{m^2\,m^{-2}\,yr^{-1}}$). Tuning done for the LAI and $Chl_{Leaf}$ did not influence the simulated trends.

The GPP estimates derived from the CN simulation were found to be in close agreement with the observations until 2010 (Fig. S16a). However, a divergence is observed between the simulations and observations from that point onwards. The observations indicate an increase, whereas the simulations remain at a consistent level before declining in the last three years (Fig. S16a). In the observations the interannual variation (in terms of standard deviation) is found to be in similar magnitude for both TER (220 $\mathrm{gC\,m^2\,yr^{-1}}$) and GPP (224 $\mathrm{gC\,m^2\,yr^{-1}}$) (Table 3). In contrast, the simulations indicate that the interannual variation is greater in the GPP (163 $\mathrm{gC\,m^2\,yr^{-1}}$) than TER (100 $\mathrm{gC\,m^2\,yr^{-1}}$) (Table 3). The nitrogen deposition that was input to the model showed a continuous decline in the study period (Fig. S17).

## 4 Discussion

In this study we employed a variety of observational data sources in conjunction with a terrestrial biosphere model. Our objective was to assess the utility of these data in enhancing and evaluating the model's performance, as well as to ascertain the model's capacity to simulate the biogeochemical cycles within the Borden forest. Our focus extends beyond the carbon cycle, as our modelling approach incorporates the nitrogen cycle, for which the available observational data provides valuable insights.

## 4.1 Using the continuous LAI observations

The site-level continuous LAI observations provide a valuable data source for model evaluation and development. Our simulations showed a discrepancy between the simulated absolute value of LAI and these observations. The LAI estimated from litterfall at the site was 5.1 $\mathrm{m^2\,m^{-2}}$ (Neumann et al., 1989) and in close agreement with the model estimate of 5.3 $\mathrm{m^2\,m^{-2}}$ derived from the C-simulations and 4.8 $\mathrm{m^2\,m^{-2}}$ derived from the CN-simulations. The changes in the biochemical model parameters are more influential than LAI values alone explaining the differences in GPP than LAI alone. The winter LAI will remain overestimated in our current modelling setup because only deciduous forest is being simulated. Changing the allocation pattern in QUINCY would have allowed a reduction in simulated LAI, but this would have resulted in lower GPP values.

Tuning of the senescence parameter (Table 2) resulted in substantially higher temperature threshold for the start of the senescence period at the site compared to the standard parameterisation. As described in Section 2.3.2, the phenology model of QUINCY is a simple growing-degree based formulation and does not take into account other environmental conditions such as light availability or day-length, which might be contributing to early leaf senescence. Day-length in particular has been shown to be another important variable controlling senescence in these regions (Bowling et al., 2024). Future work will address whether implementing such a dependency in QUINCY improves the seasonality of LAI estimation without adjusting the temperature threshold for senescence.

The continuous measurements are of particular value for assessing model seasonality, as they provide continuous data, unlike the point values obtained from LAI-2000 or litterfall. Therefore, using these data to improve the seasonality of the modelled LAI is a logical approach. By adjusting the senescence in accordance with the LAI observations, it is possible to improve the seasonal cycle of GPP, as the senescence was occurring too late in the default model at this site (Fig. 1). In this context, the continuous measurement of LAI is of particular significance, as it represents an independent measure from the carbon fluxes that can also be observed from space.

The parameterization of QUINCY often relies on multiple sites in order to perform successful large-scale simulations. An ongoing study evaluating the QUINCY seasonal cycles with flux tower and remote sensing data (T. Miinalainen, pers. comm.) has shown that a similar bias in autumn phenology for temperate broadleaf deciduous trees occurs at several other sites, so this PFT could benefit from parameter tuning. However, the satellite observations have some issues in predicting the autumn phenology (Wang et al., 2024) and the long-term in situ observations at the Borden site can serve as a valuable verification resource.

## 4.2 Other leaf-level observations

In addition to LAI, there were several other leaf-level observations at the site, which we used in our model evaluation. In the original QUINCY formulation the development of leaf chlorophyll is fully coupled to the LAI development (Thum et al., 2019). However, observations reveal a clear decoupling taking place during the early season (Croft et al., 2017). The approach we have adopted here is to delay the development of leaf chlorophyll from the structural, i.e. non-photosynthetic, nitrogen. Previous studies at the site indicated that leaf chlorophyll lags 30 days behind LAI in reaching its maximum summertime value

(Croft et al., 2015). However, the difference observed in the modelling was less pronounced, as the objective was to accurately represent the seasonal development of leaf chlorophyll correctly, while the observed maximum value of $Chl_{Leaf}$ was reached later than the simulated summertime maximum level (Figs. 1c). The observed $Chl_{Leaf}$ values exhibit elevated levels during the midsummer period, spanning approximately one month. In the modelling conducted, the objective was not to replicate this specific behaviour but rather to capture the average summertime level. The observed midsummer peak in $Chl_{Leaf}$ was not simultaneously accompanied by increases in the biochemical model parameters $V_{c(max),25}$ and $J_{max,25}$ (Fig. 3), which would have a stronger effect on the simulated photosynthesis than $Chl_{Leaf}$ alone. We could test the effect of increased midsummer $Chl_{Leaf}$ on photosynthesis using the canopy module of QUINCY, which calculates only the canopy part of the model. Given the structure of the model, this effect is likely to be small. Luo et al. (2018) used the fact that $Chl_{Leaf}$ and $V_{c(max),25}$ have a linear relationship at the Borden site to improve their model results for GPP and evapotranspiration. In QUINCY we currently do not have this linear relationship, although it has been observed in some studies (e.g. Qian et al. (2021)). A further study with remote sensing data and more sites will explore the implementation of such a description of leaf N partitioning in QUINCY. At the Borden site the linear relationship between $Chl_{Leaf}$ and $V_{c(max),25}$ was found to be off due to their different rates of light acclimation (Yu et al., 2024). Once we have a linear relationship between $Chl_{Leaf}$ and $V_{c(max),25}$ implemented in QUINCY, we can use the canopy module to estimate what effect this would have on photosynthesis in our model. Currently, the way we have described the decoupling of LAI and $Chl_{Leaf}$ in QUINCY does not lead to pronounced differences in the annual C balance (Table 3).

Another variable that undergoes changes during the spring season in the observations is the specific leaf area (SLA). Rather than delaying the onset of chlorophyll development from the structural nitrogen, it would be possible to introduce a dynamically changing SLA to QUINCY, as the leaves are thinner immediately following budburst. The SLA in QUINCY is set to a PFT-specific, time-invariant constant representative of the average observed value in Kattge et al. (2011), which is almost double the observed summertime value at Borden forest. This overestimation may contribute to the large LAI predicted for the site by QUINCY, as the SLA is used in the leaf area calculation (Thum et al., 2019). Testing dynamically changing SLA in the model falls outside the scope of our current study, but is an important future step for improving predictions of leaf process seasonality.

## 4.3 Limitations of the model

It should be noted that, like with any other modelling study, our modelling approach is subject to certain limitations, due to the necessity of making certain simplifications. Borden forest is a mixed forest, and the different deciduous species differ in their leaf chlorophyll contents and SLA values (Croft et al., 2017), and likely in their responses to climate variability. Since we do not have the ability to model different tree species, we model a deciduous forest composed of trees with identical traits. The rationale behind our approach can be attributed to the large scale that we are aiming to model and the focus on utilisation of parameters that can be derived from remote sensing observations which are unavailable at a scale where individual trees (and their species) can be resolved.

Another challenge in characterising the forest as a deciduous PFT is the exclusion of the coniferous trees at the site. This leads to discrepancies in our wintertime estimates of LAI, as the simulated deciduous trees lack leaves during this season. Another

effect can be seen in the delay of the simulated GPP increase in the spring, which occurs in the observations after DOY 100 (Fig. 1a, b). This is partly due to the understorey vegetation, which we do not simulate, and partly due to the coniferous trees at the site, which can start photosynthesis as soon as meteorological conditions and the release of possible winter acclimation allow. The shape of the seasonal cycle of GPP is also different in the simulations compared to the observations (Fig. 1 b). The increase in the simulations is more abrupt to the summer levels and decline from early summer values occurs quite early, probably due to dry periods occurring during summer. This could be due to the fact that the observed transitions are more smooth in time as several tree species contribute to the trend, which is represented only by one functional type in the model. Issues with species mixtures are common in TBMs and while this is certainly an area that needs further improvement it is not an issue unique to QUINCY.

Testing the performance of a TBM designed for large-scale, site-level simulation is challenging. The model must necessarily make generalizations in the process representation due to the limited knowledge and data required for large-scale parameterization. Of particular importance to this study is the parameterisation of the partitioning of nitrogen into different compartments, which has a theoretical basis (Evans, 1989). However, the approach applied in QUINCY is simple and relies on PFT-specific parameterization, with very limited consideration of leaf-level data. The dataset available for the Borden site is very valuable in this respect, as it allows evaluating the division of nitrogen to different compartments. The parameterization used in QUINCY does not take all factors into account. For example, the phosphorus content has been shown to influence the relationship between the $V_{c(max),25}$ and leaf nitrogen (Walker et al., 2014). Furthermore, the description of the canopy nitrogen gradient in QUINCY appears to be sound, but it may not account for all possible variation (Niinemets et al., 2015). Borden forest is located in the temperate boreal forest ecotone and many species are close to the limits of their temperature and moisture ranges (Froelich et al., 2015). The tree species composition has undergone changes at the site during our study period, e.g. the red maple was reported to have coverage of 36 % in 1995 (Lee et al., 1999) and 52 % in 2006 (Teklemariam et al., 2009). The impacts that these changes in the tree composition have on the carbon fluxes could be studied by a demographic model with sufficient granularity in the description of tree functional diversity (see Fisher et al. (2018) for a review).

### 4.4 Legacy effects of drought

Carbon fluxes at the site were strongly influenced by the 2007 drought, which also led to legacy effects visible in 2008 (Fig. 4). Often the effect of drought on GPP is stronger than on TER (Schwalm et al., 2010; Piao et al., 2019), but here a stronger effect on TER was observed. GPP can be reduced by drought through both physiological and structural effects (van der Molen et al., 2011). The inclusion of non-structural carbohydrate pools in models has shown improvements in the ability of models to capture drought responses (Jones et al., 2020).

No decrease in the LAI was observed at the site in 2007 or 2008. One of the recognised mechanisms for legacy effects is that the drought-induced decrease in GPP can lead to a decrease in the carbohydrate pool and therefore influence the LAI development in the following year (Yu et al., 2022). QUINCY has an explicit reserve pool and could theoretically simulate this type of behaviour, but did not suggest that such legacy effects affected the GPP in 2008. Although the simulated soil moisture shows a similar dynamic behaviour to the observations (Fig. 4), simulated top-soil moisture did not show the same dynamic

range of moisture seasonal cycle magnitude and minimum in QUINCY when compared to the in-situ observations, possibly explaining underestimated response to soil moisture stress by the model. However, we note that the modelled range was more similar to the observations at deeper depths than the top layer (Fig. S8e, f). At present, it is unclear whether this reflects shortcomings in the representation of soil physical processes or is a result of the lack of in-situ precipitation observations during the study period (see Methods).

There are many possible explanations for the strong legacy effect observed in TER in 2008. Soil microbial activity is dependent on soil moisture (Gaumont-Guay et al., 2006; Liu et al., 2009; Orchard and Cook, 1983) and drought can thus strongly reduce soil respiration directly and indirectly through several different mechanisms (von Buttlar et al., 2018). Direct effects include the dependence on the presence of water films for substrate diffusion and exo-enzyme activity (Davidson and Janssens, 2006) as well as microbial dormancy and death (Orchard and Cook, 1983). Indirect effects affect microbial activity through, for example, changes in soil nutrient retention and availability (Bloor and Bardgett, 2012) or changes in microbial community structure (Frank et al., 2015). In addition, GPP and TER fluxes are tightly coupled, as heterotrophic respiration is also driven by the recently assimilated carbon and not only by environmental conditions (Ruehr et al., 2012). QUINCY was not able to capture the drought-induced decrease in TER in 2007, which could either be due to a too low impact of the drought on soil moisture, or the fact that the version of QUINCY applied here does not simulate microbial activity and root exudation (Yu et al., 2020). This also contributes to the failure to simulate potential legacy effects in the observed TER (Fig. 4).

The drought response at the site could potentially be improved by calibrating the soil moisture response functions in the model, but probably some structural changes in the description of soil physics would also be required. These changes may include modifications in water-retention curve, pedotransfer functions (Weber et al., 2024) or infiltration properties (Vereecken et al., 2019). Soil moisture is a challenge for many models and often in need of improvement (De Pue et al., 2023). Future research will investigate whether a more sophisticated soil biogeochemical model can better represent the effects on microbial communities and through them the legacy effects on respiration (Yu et al., 2020). Overall, terrestrial biosphere models are not yet well equipped to capture the legacy effects (Bastos et al., 2021) and more work is needed to better understand the processes governing ecosystem recovery in order to improve models in this respect.

## 4.5 Seasonality in the total ecosystem respiration

QUINCY is generally capable of modelling the observed magnitude and seasonal amplitude of observed TER, based on empirical responses of soil organic turnover to soil temperature and moisture (Table 3). The premature increase in the simulated soil respiration occurs during average years due to a too early increase in the soil temperature in the simulations (Fig. 2). This behaviour suggests that the coupling between the atmosphere and the soil in the model is too strong, which may be associated with parameters controlling heat diffusion in the soil.

One interesting feature observed in the TER is the strong seasonality in respect to soil temperature in normal years, which cannot be explained by the soil moisture (Fig. S14). This behaviour was first discussed by Lee et al. (1999) at this site and they called it the hysteresis effect. It is most pronounced in the year following the drought, 2008 (Fig. S14e). Based on the data available to Lee et al. (1999), they found that the early season had lower respiration values than the late season and

speculated that this difference might be due to warmer soil temperatures in deeper soil layers as the season progressed, as well as greater litter accumulation. The data available to-date for a longer period shows an "inverse" hysteresis effect, in which the later season has lower TER than the early season (Fig. S14a and e). Soil moisture does not provide an explanation for such a shift. Rather, this behaviour could be driven by the seasonality of photosynthesis, as the early season GPP co-incides with lower TER values compared to the late season (Fig. S15). A fast coupling between GPP and soil respiration, e.g through photosynthesis supplying carbohydrates to rhizosphere respiration (Zhang et al., 2018), observed already early in a girdling experiment (Högberg et al., 2001), could explain the observed hysteresis effect. Hysteresis effects on soil respiration versus soil temperature are quite common (Zhang et al., 2018), and further explanations for dynamics taking place at Borden could be caused by substrate depletion late in summer (Kirschbaum, 2006) or by greater root productivity in early season (Oe et al., 2011). Models generally describe soil respiration as a function of soil temperature responses and would not capture the hysteresis effects (Zhang et al., 2018). This is also the case for QUINCY, which is not able to capture this hysteresis effect (Fig. S14). Future work should evaluate whether including for instance increased vegetation-soil coupling via root exudates would improve the representation of the interannual variability of TER.

## 4.6 Trends and growing season length

There were no significant changes in the growing season metrics SOS, EOS and LOS over the period studied, similar to Gonsamo et al. (2015). QUINCY was generally successful in simulating these metrics, but the end of season as estimated from GPP and LAI differed in the observations, whereas it was coupled in the model. The previous studies at the site that have assessed the growing season metrics (Froelich et al., 2015; Gonsamo et al., 2015) also used the carbon uptake period (CUP). We did not assess CUP because its onset would have been biased in the simulations due to the premature increase in heterotrophic respiration caused by the premature increase in soil temperature (Fig. 2).

Froelich et al. (2015) found a significant increase in summertime GPP and Gonsamo et al. (2015) significant increase in carbon uptake between 1996 and 2012 at the Borden site. These are consistent with our observational results, which additionally also showed a small but significant increase in the summertime LAI. The increase in net carbon uptake is attributed to increased PAR (photosynthetically active radiation, 400-700 nm), which leads to increased photosynthetic activity at this site (Gonsamo et al., 2015). When we looked at this PAR data in more detail, we noticed a significant increasing trend that had been reported. However, its magnitude was less than 1 % per year, which would not lead to large increases in annual GPP according to our model. Also, when the time period was extended to 2018, the significant trend disappeared. The input to QUINCY was shortwave radiation, and it did not show an increasing trend for the time period, a very small decrease in annual values.

There have been reductions in atmospheric sulfur (S), nitrogen oxides, total nitrates and ozone deposition since 1992 in Borden (as measured at the nearby Egbert station) and the brightening seen in the PAR observations has been attributed to reductions in gaseous and particulate emissions, while declines in ozone emissions reduce the damages to the leaves (Gonsamo et al., 2015). The N and S deposition was found to reduce net primary production in southern Ontario (Aherne and Posch, 2013), where also the Borden forest is located. Decreasing S deposition was found to increase forest productivity in the northeastern U.S. region (Dalton et al., 2024; Phelan et al., 2024), which is close to the Borden forest. Decreasing S and N deposition will

also affect the soil pH (Dalton et al., 2024), which can potentially affect the N dynamics. The N deposition may be useful for forests under N deficit (Horn et al., 2018), however according to our model results, the Borden forest would not suffer from N deficiency. Recovery from S deposition may be one of the reasons for the increasing trend in GPP. Long-term S exposure could also reduce the drought resilience of sites (Dalton et al., 2024), which may be another explanation for why our model failed to capture the drought effects at this site.

Furthermore, the QUINCY model does not take into account potential damage to the leaves caused by ozone. Ozone influences both photosynthesis and stomatal conductance and can cause them to become decoupled (Novak et al., 2005; Lombardozzi et al., 2015). Estimates of decreases in the photosynthesis by 21 % and in stomatal conductance by 11 % after chronic ozone exposure have been estimated (Lombardozzi et al., 2013), but also lower estimates have been presented (6-10 % for Europe) (Franz et al., 2017). Ozone exposure can also have an impact on the N cycle (Simpson et al., 2014). The impact of ozone has been modelled by direct influence on $V_{c(max),25}$ and stomatal conductance (Lombardozzi et al., 2012) or then on photosynthesis, which then has feedbacks on stomatal conductance (Franz et al., 2017; Lombardozzi et al., 2015). Decreasing ozone deposition at the site could further affect the increase in photosynthesis.

QUINCY approximately reproduces the leaf-level photosynthetic parameters in the CN version of the model, but at the same time overestimates LAI (compared to the continuous observations) and underestimates GPP. The uncertainty in the annual flux estimates by the eddy covariance method is usually around 10-20 % (Loescher et al., 2006), so QUINCY's estimates are within the uncertainty of the observations. Another source of uncertainty in the observations are the use of gapfilling (Mahabbati et al., 2021) and partitioning methods (Desai et al., 2008), which introduce uncertainty in the annual carbon balance estimates.

Possible reasons for discrepancy between QUINCY and the observations are the lack of understorey representation in the model, the simplified representation of the mixed forest with a single deciduous plant functional type, and possible biases introduced by the assumed within-canopy gradient of leaf nitrogen, which might not hold for this diverse forest. It is interesting to note that the differences between the measured and simulated annual GPP are not apparent in the early years of the record, but emerge in the later years (Fig. S16). The underestimation of annual GPP is 13 % by the CN:LAI&chl simulations for the whole period, but only 8 % for the years 1996-2010. Therefore, the increasing trend in observed GPP that the model fails to reproduce is contributing strongly to the model-data discrepancy.

### 4.7 Impact of nitrogen cycle on the carbon fluxes

Including the nitrogen cycle in the simulations did not cause a change in the net carbon balance of the ecosystem, as both the GPP and TER were both attenuated by approximately the same amount, about 20 % when comparing to the C version with N saturation. With the fixed stoichiometry the C model gave similar values to the CN simulations (Table 3). This means that there was no nitrogen constraint on the carbon fluxes according to the QUINCY model. The leaf C:N was only 9.7 in the C:LAI&chl saturated case simulation, which is an unrealistically low value, but we chose to show these results here to assess the influence of parameterization with simulations with magnitudes comparable to the observed GPP. The foliar C:N ratio was 20.4 in the CN:LAI&chl simulation and 22.4 in the Cfix:LAI&chl, showing that they are similar.

Nitrogen availability limits the carbon cycle (Du et al., 2020), especially in the boreal region (Högberg et al., 2017). The estimated effect of the N cycle on the carbon fluxes is not as large as some previous estimates (Thornton et al., 2007). The summertime variation in GPP values is more pronounced in the C simulations with dynamic stoichiometry than in the CN simulations, highlighting the more stable behaviour of the model when including the N constraint and compared to the N saturated case. Although the annual GPP was underestimated with the inclusion of the N cycle, after tuning by both LAI and

$Chl_{Leaf}$, the CN simulation gave best $r^2$ and RMSE metrics for both GPP and TER, in line with the C-cycle simulations with fixed stoichiometry (Table 3).

### 4.8   Using site level observations in model development

The different site level observations available at the site provided means to evaluate the model performance from different aspects. The QUINCY model is a large scale model and cannot capture all the small scale variations. Furthermore, the different

tree species in Borden complicate simulating the forest with QUINCY, as the model needs to rely on a general PFT description. However, to better understand the processes occurring at the site some further observations would be useful. To capture the forest structure and to facilitate estimation of the radiative transfer inside the canopy, LiDAR observations (Balestra et al., 2024) would be beneficial, and if done on temporally continuous scale (such as in StrucNet, see Calders et al. (2023)), also valuable information on allocation of annual net primary production could be obtained. Soil chamber observations of respiration would

help to separate the role of soil in the total ecosystem respiration. Use of isotopes would enable revealing the processes behind the observed hysteresis behaviour of the soil respiration. A rain gauge at the site would help to study potential biases of using precipitation data at a nearby site. The ozone profile concentration observation at the site could help in estimating the potential ozone damages on the vegetation, that could be addressed by a model. Observations of leaf and soil C:N ratios would help to better understand the nitrogen status of the forests.

### 4.9   Revisiting the research questions and future responses

Our first research question was whether the decoupling of LAI and $Chl_{Leaf}$ affects the estimation of annual carbon fluxes. Delaying the development of $Chl_{Leaf}$ did affect the seasonal development of GPP, but the effect on annual fluxes was very small. If the $Chl_{Leaf}$ was more linearly related to the biochemical model parameters in the model, the influence might be somewhat greater.

Our second research question was whether QUINCY is able to simulate any long-term changes in seasonal shifts in carbon fluxes and LAI values. The observed trends in GPP found at the site were quite large and were not captured by our model, even though it does responds to climate and increasing atmospheric $[CO_2]$. Possible reasons for this could be processes that are not included in our model, such as the role of the understory, changes in tree composition, and recovery from S and ozone deposition. The observations showed a small increasing trend in LAI that was not captured by the model. If the reason for the

increase is a change in tree composition, our model would not include the process behind it.

Our third research question was whether nitrogen limitation exists in the Borden Forest and whether it changes over the 22-year period. To answer this question, we compared the nitrogen cycle enabled version to the carbon cycle only version with

fixed stoichiometry. The differences in the annual carbon fluxes between these two simulations were very small, and there was no change in time between these results. Therefore the Borden forest would not be nitrogen limited. This is consistent with the finding that observed GPP continues to increase even though the N deposition at the site has decreased since 1995.

Our fourth research question was whether QUINCY could simulate the effects of drought events on the carbon cycle. In general, the model captures many environmental responses that occur at the site, as the model performance is generally good. However, the effects of the 2007 drought on carbon fluxes in that year and the following year were not captured by the model. The top layer of the soil contains most of the organic matter and most of the heterotrophic respiration in the model comes from this layer. The soil model does not have a wide enough range of soil moisture values in the top layer compared to observations. This may be one reason for the discrepancy. QUINCY has non-structural carbohydrate pools and they get lower values in 2007 and the reserve pool also in 2008 compared to regular years. These changes are not reflected in the summer LAI values, which would be one way to carry out the legacy effect of the drought into other years, but this was also not detected in the observations. The legacy effect of respiration is likely due to a process not currently represented in QUINCY.

QUINCY was generally able to capture the environmental responses of the forest. However, the long-term trends and extreme events pose challenges to the model, that would require more detailed representation of some processes. These challenges highlight the value of long-term data series. The combination of data and simulations shows a decoupling of LAI development and $Chl_{Leaf}$ as well as too strong coupling of LAI and GPP in the model in autumn compared to the observations.

In the future the Borden forest is likely to have warmer temperatures and more precipitation in winter (Bush et al., 2022). A warmer climate is expected to increase the severity of heat waves, which can in turn may contribute to increased droughts (Bush et al., 2022). The 2007 drought likely had a legacy effect on the 2008 component fluxes. However, because the TER was reduced more than the GPP in 2007, the forest was a net sink (-318 $\mathrm{gC\,m^{-2}yr^{-1}}$) in that year, and also in the following year (-282 $\mathrm{gC\,m^{-2}yr^{-1}}$). Thus, the drought did not endanger the sink capacity of the forest, but rather increased it. If droughts become more frequent, they could threaten the sink capacity of the forest. Estimating the effect of warmer temperatures is challenging because the observed trend in GPP is not captured by the model. QUINCY would predict both increased photosynthesis and respiration at warmer temperatures, but whether this would lead to a higher sink or source would depend on the overall conditions.

## 5 Conclusions

In this work we used several data streams measured at the Borden Forest Research Station, some extending over two decades, and aimed to improve and evaluate the terrestrial biosphere model QUINCY. This work demonstrated the usefulness of using different data sources and the importance of observational long time series. The use of leaf chlorophyll content and LAI in parameterizing the model improved simulated GPP in the CN simulations. These changes also decreased the RMSE for TER. Generally the model did capture average seasonal cycle of GPP (daily $r^2$=0.80) and TER (daily $r^2$=0.75). QUINCY was also successful in estimation of the growing season metrics, even though the ending of the season was more coupled between LAI and GPP than in the observations.

Two important data sources used in this work, leaf chlorophyll content and leaf area index (LAI), can also be measured from space. Therefore our work paves the way toward combining terrestrial biosphere models (TBMs) and using remote sensing data for their parameterization, as has been proposed by Rogers et al. (2017). Work in this front has been done by combining leaf chlorophyll to the photosynthesis parameters of models (Lu et al., 2022). In this work we explicitly model the leaf chlorophyll, which links this variable directly to the nitrogen cycle. The unique dataset accessible from the Borden site permitted the assessment and enhancement of the parameterization employed to divide leaf nitrogen to different compartments. In addition to utilising LAI and leaf chlorophyll, sun-induced chlorophyll fluorescence (SIF) represents a pivotal variable observed from space that is linked to the carbon cycle (Sun et al., 2023). SIF is currently being implemented in QUINCY and will, in the future, provide a means of conducting a global-scale assessments of the carbon cycle together with the nitrogen cycle related metrics.

*Data availability.* Data, including model results from the CN:LAI&chl simulations and meteorological forcing used to run the model, can be found at https://fmi.b2share.csc.fi/records/81778e9da06243d5bccdd364cfdb320a.

*Author contributions.* TT designed the study. OS did preliminary analysis and code modifications proposed by SZ. TM performed final analysis and made the figures. HC, RS and CR provided observation data. TT wrote the first version of the manuscript. The interpretation of the results was developed in discussions with all the authors. The manuscript was commented by all the authors.

*Competing interests.* The authors declare no competing interests.

*Acknowledgements.* We acknowledge the CA-Cbo AmeriFlux site for its data records. In addition, funding for AmeriFlux data resources was provided by the U.S. Department of Energy's Office of Science. TT, TM and OS acknowledge funding from Research Council of Finland (RESEMON project, grant number 330165; and 337552), and for TM, also Flagship of Advanced Mathematics for Sensing Imaging and Modelling, grant number 359196). Scientific programmers Dr. Jan Engel and Dr. Julia Nabel are thanked for technical support and maintenance of the QUINCY code. Dr. Manon Sabot is thanked for useful discussions. We wish to thanks two anonymous reviewers whose feedback improved this paper.

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
