# Peer review of "Modelling decadal trends and the impact of extreme events on carbon fluxes in a deciduous temperate forest using the QUINCY model"

_EGUsphere, 2024_

## Referee Comment (RC1)

**General Comments**

In the manuscript "Modelling decadal trends and the impact of extreme events on carbon fluxes in a deciduous temperate forest using the QUINCY model", Thum et al. explore how using in-situ measurements of vegetation traits to parametrize the nitrogen cycle-enabled QUINCY model improves flux simulations at the Borden Forest flux tower. The QUINCY model is modified to allow for a delay in leaf chlorophyll development and the model runs produced in nine different simulations of varying parametrization and nutrient cycle implementations.

When using locally calibrated parameters and enabling the nitrogen cycle in addition to the carbon cycle, QUINCY performed better against observations, especially for GPP, but still lacked some key behaviors – namely it failed to capture drought response and legacy.

Overall, the manuscript is of a high quality, presenting a well-developed study in a clear and scientific manner. It is a substantial contribution to the literature. I recommend publication following minor revisions.

**Specific Comments**

1. I like the specific statement of the research questions in the study at the end of the Introduction. However, these are not referred to again. I would suggest a section of the Discussion is reformulated to explicitly restate these questions and then explore the evidence found for each one. This would provide a good narrative throughout the manuscript and help in synthesizing the findings of the study.

2. Among the limitations of the study is the level of representativeness of the chosen PFT for the flux tower site. According to my reading of Thum et al. (2019) and its supplementary material, QUINCY has the capacity to model an individual gridcell as nested tiles of different PFTs. Does this not provide the necessary flexibility for QUINCY to represent a more accurate mix of vegetation when modelling Borden Forest? What was the inter-species variability in the leaf-level parameters and how much did the species-weighted average differ from the species level values?

3. Studies have shown that a simple representation of carbohydrate pools can help to model drought legacies (Guo et al., 2020; Jones et al., 2020). QUINCY has a representation of these NSC pools and so might be expected to capture the drought legacy in 2008. As such, I'd like to see Section 4.4 in the discussion expanded to discuss in more detail, or at least more focused on, the reasons why QUINCY is

unable to simulate drought and drought legacy. Exploring the processes behind the observations is interesting but this should be framed from the perspective of model evaluation and used to synthesize our understanding of QUINCY's performance.

4. Table S1 is referred to throughout the manuscript at greater frequency (12 times by my count) than every figure and table in the manuscript except Figure 4 which is also referenced 12 times. Alone, it is referred to more times than all three of the tables included in the manuscript! I would suggest moving this table to the manuscript or otherwise incorporating this information in a manner that does not result in the jarring requirement of frequently referring to the supplement.

5. Figure 2 shows the mean yearly values for NEE, TER, soil temperature and soil moisture. This is discussed in detail. My feeling, considering the soil temperature plot and the location of Borden Forest, is that snow cover may be playing a substantial role at this site over winter. Note that the observed soil temperature maintains a constant temperature through winter, consistent with an insulating snow cover. This may also explain the model's earlier TER response as QUINCY fails to simulate any snowmelt period suppressing soil respiration despite increasing temperatures (Teklemariam et al., 2009). Was this potential role of snow considered?

**Technical Comments**

1. L20: Delete "the" from before "ecosystems".
2. L27: Longer growing seasons compared to what?
3. L30: Delete "the" from before "vegetation functioning".
4. L33: Delete "the" from before "forests". Note this is a recurring issue as per comments 1 and 3. Since we are not discussing specific forests or ecosystems, there should not be a definite article "the" in these instances.
5. L51: Add "an" before "increase" and a "the" before "land carbon sink". Modify "changing of" to "change in".
6. L61: GPP has already been defined.
7. L69: Delete the comma after "Borden Forest" or add a comma after "continuous data".
8. L78-79: This sentence needs to be reformulated to make sense.
9. L81: $Chl_{Leaf}$ has already been defined.
10. L83 and throughout: Make sure references are formatted correctly, namely the brackets.

11. L125: GPP has already been defined.

12. L163: The acronym QUINCY has already been defined.

13. L202: Define or explain "OCN".

14. L228: Delete "an" from before "the slope".

15. L243: Add "and" between "humidity" and "wind".

16. L247: Was a single random year used repeatedly in the spin-up or was it a 500 year time series constructed by randomly selecting a year of [CO2] and meteorology for each of the 500 years?

17. L249: What data was used for the meteorology between 1901 and 1996 in the transient runs? ERA-5 is mentioned earlier but this dataset begins in 1940.

18. L264: Change "showed" to "shown".

19. The model abbreviations are quite long and complicated – is there a way to condense them while maintaining the information?

20. L272: As I understand it, LAI was a parameter that was calibrated and this can be seen by the improved seasonality of LAI in Fig 1c&d. Was there no capacity to correct the magnitude of LAI which is too high in summer and too low in winter?

21. L286: Figure S2 shows 2013, not 2014.

22. L297: "The observations show more shallow decrease" - I think this should be "The observations show a more gradual decline in increasing GPP before the peak" or similar?

23. L304: "parameterizations is in" should be "parametrizations are in".

24. L305: It may be worth clarifying that this refers to different simulations within the "C-only" model framework only, not between the "C-only" and "C-only,fix" model simulations.

25. L309: Perhaps specify that the "more accurate representation" is better timing of senescence?

26. Figure 1: GPP and LAI have already been defined, and "leaf chlorophyll" has not been defined as "Chl" but as "Chl$_{Leaf}$". I also do not think that the colors of each model simulation need to be spelled out in the caption as the legend already provides this information. It should be "… leaf chlorophyll WHICH has been smoothed…".

27. L313: Perhaps a column should be added to Table S1 containing the percentage figures of under/overestimation.

28. L316: Should it be "with only a 1.4% larger value"?

29. L320: I think this should refer to Table S1, not Table S2.

30. L322: TER has already been defined.

31. L330: "Table 1" should be "Table S1".

32. Figures S3, S4, and S5 are often used to illustrate points that are difficult to parse from the plots (namely referring to fluxes being over/underestimated in certain months). The plots are too noisy for me to clearly and easily identify these statements so I suggest a different method of plotting this data to illustrate the points made.

33. L339: "The early season pattern observed in the simulated NEE is attributed to heterotrophic respiration". Is not the overestimation of early season NEE due to a GPP that is too low from DOY ~75, as well as TER being too high between DOY 50 and 100? This is what I read from Figure S6.

34. L348: Missing bracket after "(Fig. 2."

35. L355: This sentence is not properly formatted.

36. L365 and surrounding paragraph: This paragraph could be condensed. It is also unnecessary to specify (C-only:LAI&Chl) and (CN:LAI&Chl) after each reference to the models as it has already been stated that these are the models referred to from Line 318 onwards.

37. L376: This statement about SLA is disjointed from the rest of the text.

38. L379: TER is discussed first in Figure 4 but in the plot, GPP comes first. I would ensure these align to improve readability.

39. Figure 4: In this instance, I think it might be better to have the observed and simulated on the same plots, with the different facets instead for the years – as it stands, the figure is better as a comparison between years than for assessing model performance, which I believe to be its main message.

40. Figure 4 and Figure S9: As in other plots, could the standard deviation be plotted as a shaded area around the observation means?

41. L392: Move the sentences about GPP to the next paragraph which discusses GPP more widely, rather than in this paragraph which is about TER. Section 3.4 in general could use some work to improve the flow of the section and maintain a narrative throughout instead of the current text which tends to jump between the discussed variables with little reason. Either discuss each variable in turn, or discuss each time period in turn.

42. Table 4: The QUINCY LOS (GPP) is incorrect – it should be 146 days if calculated as the difference between the mean SOS and mean EOS as the other table elements seem to imply (or perhaps this is coincidence and it is actually the mean length of each individual growing season, in which case this should be clarified).

43. L417: "LAI based estimates" is misspelled.

44. L420: "takes place in average" should be "takes place on average".

45. L429: Clarify what is meant by "make use of different spring and autumn periods".

46. L438: There appears to be a significant breakpoint in GPP around 2009. Are there any explanations for why this might be? Is this a recovery from the 2007 drought? Is there any potential reason why every year post 2010 has higher GPP than any year before? Why were the final 5 years removed as a test?

47. L445: Why is LAI not plotted in Figure S12 if it is discussed here?

48. L451: Can the differences in IAV be quantified somehow, for instance comparing the standard deviations?

49. L469: Incorrect parentheses for reference.

50. Section 4.1: I'd like to see more discussion here about how the continuous LAI measurements improved the model performance. What has been learnt from this study and how can these lessons be implemented in the model? Does the senescence parameter need to be modified in the standard implementation of the model? Do we need to test the model tuned with continuous LAI values at other sites?

51. Section 4.2: Again, I think there could be more discussion here about what the simulated $Chl_{Leaf}$ values can teach us regarding QUINCY and future improvements. What does it mean for the model simulations that $Chl_{Leaf}$ peaks early? What might this imply for QUINCY applied at other sites or globally? While not the objective in this study, what might be learnt if we did attempt to capture the timing of the maximum leaf chlorophyll? What would the tradeoffs be?

52. L511: "observed transitions is more smooth" should be "observed transitions are more smooth".

53. L514: I'd change "the model used here" to "QUINCY".

54. L515: Missing word between "Testing model performance" and "a TBM designed".

55. L570: "It occurs most pronounced" is not grammatically correct – replace with "It occurs most prominently" or "It is most pronounced".

56. L571: "Based on the data available then…" reads as if referring to the 2008 drought due to the prior sentence. I'd recommend "Based on the data available to Lee et al. (1999), they found …" or similar.

57. L583: Delete comma after "exudates".

58. L587: Typo in "whereas is was".

59. L598: "QUINCY does simulate" should be "QUINCY does not simulate".

60. L600: There are quite a few statements throughout the manuscript similar to the sentence here: "One additional cause of model failure might be that the canopy light-saturation point does not reflect the observations, however, there is not robust evidence that this is the case." These require at least some explanation of what potential sources of evidence were explored and discounted.

61. L605: Add "an" before "impact on the N cycle".

62. L619: Add "in" after "cause a change".

63. L623: Add "an" before "unrealistically low value".

64. L630: Add "the" before "N saturated case".

65. L631: Delete "and" before "in line with".

66. L633: Delete "of" in Section 4.8 header.

67. L642: Add "A" before "rain gauge".

68. L648: Specify that the "long time spans" is referring to long time series of observations.

69. L648: Specify that the "use of leaf chlorophyll content and LAI" is in parametrizing the model.

70. L658: Change to "attributed to an increase in PAR which is not visible in the shortwave radiation forcing for QUINCY".

71. L661: Delete "is" from before "paves the way".

References

Guo, J. S., Gear, L., Hultine, K. R., Koch, G. W., & Ogle, K. (2020). Non-structural carbohydrate dynamics associated with antecedent stem water potential and air temperature in a dominant desert shrub. *Plant, Cell & Environment*, *43*(6), 1467–1483. https://doi.org/10.1111/pce.13749

Jones, S., Rowland, L., Cox, P., Hemming, D., Wiltshire, A., Williams, K., Parazoo, N. C., Liu, J., da Costa, A. C. L., Meir, P., Mencuccini, M., & Harper, A. B. (2020). The impact of a simple representation of non-structural carbohydrates on the simulated response of tropical forests to drought. *Biogeosciences*, *17*(13), 3589–3612. https://doi.org/10.5194/bg-17-3589-2020

Teklemariam, T., Staebler, R. M., & Barr, A. G. (2009). Eight years of carbon dioxide exchange above a mixed forest at Borden, Ontario. *Agricultural and Forest Meteorology*, *149*(11), 2040–2053. https://doi.org/10.1016/j.agrformet.2009.07.011

---

## Referee Comment (RC2)

**General comments**

The study of Thum et al. investigates carbon and nitrogen interactions in the Ontario's Borden Forest Research Station, using in-situ measurements to parameterize the QUINCY model. It evaluates carbon flux simulations over 22 years, finding good alignment in some metrics like GPP but identifying key discrepancies in ecosystem respiration trends and legacy drought impacts, underscoring the need to improve TBMs.

The manuscript addresses an important topic: the representation of carbon and nitrogen interactions in TBM models. Overall, it is well-written, and the figures—despite a few editorial issues—are clear and effectively support the results presented. However, I found the study's objectives difficult to discern from the Abstract. The Introduction also requires substantial revision, as it sometimes lacks logical flow, and the paragraphs don't fully cohere. For instance, the first paragraph focuses on the importance of nitrogen, followed abruptly by a discussion on changes in the growing season due to warming, and then by a mention of the value of long-term observations for capturing anomalies (anomalies of what?). The Introduction feels like a series of loosely connected points without a clear narrative thread, which makes it challenging to understand the paper's aim until the objectives are listed in the final paragraph.

Regarding the content of the paper, there are two key aspects that should be addressed:

1. **Quality of Eddy Covariance Flux Post-Processing:**
   The study relies heavily on estimates of GPP and TER derived from eddy covariance measurements, which are not directly measured values. It is essential to assess the quality of the partitioning and gap-filling methods used. A study from 2004 is referenced for both flux partitioning and gap filling, yet there is no description of this approach, its advantages, or why it was chosen. Given that GPP estimates are highly sensitive to partitioning methods, it's critical to first establish a solid foundation, showing that the best possible gap-filled and partitioned fluxes were derived before making further interpretations about model structure or other underlying factors.
2. **Snow Cover Effects:**
   Considering the site's geographic location, persistent winter snow cover is likely; if this is not the case, it should be explicitly mentioned. However, the manuscript does not address snow cover or its potential impact on carbon flux processes. This is especially relevant in the shoulder seasons, where the authors discuss discrepancies between modeled and observed fluxes. Factors such as snow effects, soil freezing and thawing, and changes in soil-air temperature decoupling due to snowmelt could significantly influence these processes but are omitted in the analysis.

Below, I provide more specific comments on the manuscript.

**Specific comments**

Line 10: Please also report the RMSE (and not just r2) when reporting the model performance.

Line 10: You mentioned how you parameterized the model but not how the model was improved.

Line 11: Would be interesting to know the magnitude of this increase

Line 11: NEE not defined yet

Line 35: Grammar ("are" missing)

Line 42: Would be important to add here why the current representation of N limitation of photosynthesis is not sufficient (since this is such a key aspect in this paper)

Line 43: For example, which responses?

These aspects are important to be clarified to convince the readers why "it is paramount that the effects of N constraints on plant productivity are accurately simulated".

Line 50 and 52: Do these observations overlap? Both in terms of time and space? How is the increase in LAI and the decline in N related?

Line 79: Grammar

Section 2.1: Please add a description of the understory vegetation (species and cover). This information is relevant to your discussions of model limitations that we read later in the paper.

Lines 101-104: Is this the species composition within the flux footprint?

Line 107: Mean over what period?

Line 118: Used for what? For calibration of the model? For validation?

Line 126: A brief description of the gapfilling and partitioning method should be given here and the justification why such method is selected over other existing methods.

According to Barr et al 2004 "seasonal onset, rise and fall of photosynthesis from the $F$NEP time series based on the parameter $Px$ in the rectangular hyperbolic model "was this the case too in this study?

Line 128: Grammar

Line 133: How exactly was this scaling done?

How well do the ERA5-Land precipitation product and measured precipitation at the site compare? Perhaps a comparison can be added to the supplementary.

Line 193: Typo

Line 274: Based on what was this level of T selected for the adjustments?

There is a typo just before the caption of Table 2.

Figure S2 has a typo in the legend of panel b (should be LAI not GPP)

In Figure S2 the panels already show GPP or LAI. I would not repeat this again in the legend. Also, the dashed lines can be removed from the legend (to make it less crowded) and keep their description in the caption.

Figure S3- 5 the axis label should be "modelled" to be consistent with the x-axis which says "Observations". The figure title is already mentioning which model was used, not needed to mention this in the title and in the axis label.

Figure S3- 5 wouldn't it make more sense to display mean daily temperature as the third value rather than the number of the month? What is the reasoning to use month here? Maybe can briefly be added to the caption.

Figure S3- 5 it is hard to judge quantitatively the comparison of different model performances. Maybe at least the r2 values can be displayed in the panels?

Figure S8 very hard to read the figure. Consider increasing the font please.

Line 339-345: Could this inaccuracy in modelling the soil temperature be because of the snow? What is the contribution of snow cover at this site? We see from colder sites that snow has an insulating effect on the soil that decouples its temperature from air temperature. If direct measurements are not available at the site perhaps you could explore available remote sensing products (e.g., MODIS/Terra (MOD10A2) and MODIS/AQUA (MYD10A2) (Hall and Riggs 2021) Snow Cover 8-Day L3 Global 500m SIN Grid, Version 6 dataset, which provides maximum snow cover extent at 8-day temporal resolution and 500m spatial resolution).

Line 460-464: Here the Results are repeated. Instead, there should be a discussion of what underpins the observation that although modelled LAI is overestimated, modelled GPP in summer is underestimated.

Section 4.2 I suggest dividing parameters by structural from photosynthetic traits.

Line 491-493: What explains it if the SLA was overestimated but GPP underestimated?

Line 500: "instead, we are estimating the tree traits per average individual for a deciduous forest." Not clear to me what this statement means. Because the methods section (2.2.3) only mentions "we used a species-weighted canopy average of the leaf-level parameters, based on the species composition of the forest „ and is not clear how the parameters are weighted and then aggregated (?). Adding a mathematical description here would be very helpful.

Line 510: Which drought occurrences?

Line 515-517: Grammar check and re-writing needed. Sentence is not clear.

Line 525: "The tree species composition has undergone changes at the site during our study period, e.g., the red maple was reported to have coverage of 36 % in 1995 (Lee et al., 1999) and 52 % in 2006 (Teklemariam et al., 2009). The impacts that these changes in the tree

composition have on the carbon fluxes could be studied by a demographic model with sufficient granularity in the description of tree functional diversity"

Since species compositional shift was not really addressed in this study would suggest to remove this context from the Introduction as currently it reads as if this is one of the aspects this paper addresses.

Section 4.4 it is not clear if the model fails to simulate the (legacy) effect of drought because of its structure (representation of the carbohydrate pools) or the lack of precise soil moistures estimations which directly affect modelled $CO_2$ fluxes. Could this not have been specifically tested (for this particular case of drought conditions) by first calibrating the model using observed soil moisture measurements and testing whether GPP simulations improved during drought? If the model´s limitations during drought is a focus of this paper it deserves a more systematic approach to test this.

Line 556: What is meant here? Isn't a depth-resolved soil texture provided to the model? Otherwise, what description of the soil physics exactly is lacking here?

Line 570-576: Yes, this hysteresis in TER response to temperature (early season lower respiration than late season) could reflect the seasonal patterns of photosynthate allocation to roots. Tree girdling studies have shown that seasonal pattern of below-ground C allocation may be more important than soil temperature in determining root respiration (see for example Högberg et al. 2003) and that earlier in the season respiration from the soil is mostly due to heterotrophic respiration.

Line 591: Increase over time? Sentence reads incomplete.

Line 597: So the PAR that was used as an input to the model was not correct? The model would clearly use PAR, so why give it the wrong forcing?

Line 603: You mean at this site? It is not clear how these findings are relevant to this study.

Line 600-607: The discussion on the ozone effect comes out of blue here, unless my previous comment is clarified.

Lien 614: What is meant by "differences between the annual GPP and carbon balance"?

Line 619 and 620: Grammar

Section 4.8: I would not finish the manuscript on listing technical shortcomings. What have we learnt from long term ecological response of such an ecosystem, observations and model results combined? And what is the outlook under further changes in the climate? (e.g., predicted potential increase in temperature and dryness).

The Conclusion section can be shortened.

**References**

Hall, D. K. and G. A. Riggs. MODIS/Terra Snow Cover 8-Day L3 Global 500m SIN Grid, Version 6. Distributed by NASA National Snow and Ice Data Center Distributed Active Archive Center (2021).

Högberg, P., Nordgren, A., Buchmann, N. et al. Large-scale forest girdling shows that current photosynthesis drives soil respiration. Nature 411, 789–792 (2001). https://doi.org/10.1038/35081058

---

## Author Comment (AC1)

Borden paper review: Reply to Reviewer #1

(The comments by the reviewer are in magenta, the replies to the comments are in black, new text added or modified in the manuscript is written in italics.)

General Comments

In the manuscript "Modelling decadal trends and the impact of extreme events on carbon fluxes in a deciduous temperate forest using the QUINCY model", Thum et al. explore how using in-situ measurements of vegetation traits to parametrize the nitrogen cycle-enabled QUINCY model improves flux simulations at the Borden Forest flux tower. The QUINCY model is modified to allow for a delay in leaf chlorophyll development and the model runs produced in nine different simulations of varying parametrization and nutrient cycle implementations.

When using locally calibrated parameters and enabling the nitrogen cycle in addition to the carbon cycle, QUINCY performed better against observations, especially for GPP, but still lacked some key behaviors – namely it failed to capture drought response and legacy.

Overall, the manuscript is of a high quality, presenting a well-developed study in a clear and scientific manner. It is a substantial contribution to the literature. I recommend publication following minor revisions.

We thank the reviewer for the positive insight towards our study and thank for the detailed, in-depth comments that have helped us to improve the manuscript.

Specific Comments

1. I like the specific statement of the research questions in the study at the end of the Introduction. However, these are not referred to again. I would suggest a section of the Discussion is reformulated to explicitly restate these questions and then explore the evidence found for each one. This would provide a good narrative throughout the manuscript and help in synthesizing the findings of the study.

This is a good idea and we'll follow it.

2. Among the limitations of the study is the level of representativeness of the chosen PFT for the flux tower site. According to my reading of Thum et al. (2019) and its supplementary material, QUINCY has the capacity to model an individual gridcell as nested tiles of different PFTs. Does this not provide the necessary flexibility for QUINCY to represent a more accurate mix of vegetation when modelling Borden Forest? What was the inter-species variability in the leaf-level parameters and how much did the species-weighted average differ from the species level values?

To run QUINCY at site level, we can only describe one PFT at a time. Therefore for sites with clearly two PFTs (e.g. tree-grass savannah) we have done simulations with two different PFTs. This approach has many issues, as there is no way for these simulations to communicate and the simulated ecosystems would end up having e.g. different soil

hydrological conditions and we need to add in the results from these two simulations, thus making an analysis more difficult. We did not have leaf level observations on the evergreen needleleaves, so we couldn't have compared our modelling results against observations.

We will add a table having leaf-level observational values for different species.

3. Studies have shown that a simple representation of carbohydrate pools can help to model drought legacies (Guo et al., 2020; Jones et al., 2020). QUINCY has a representation of these NSC pools and so might be expected to capture the drought legacy in 2008. As such, I'd like to see Section 4.4 in the discussion expanded to discuss in more detail, or at least more focused on, the reasons why QUINCY is unable to simulate drought and drought legacy. Exploring the processes behind the observations is interesting but this should be framed from the perspective of model evaluation and used to synthesize our understanding of QUINCY's performance.

Reasons for QUINCY not to capture the drought legacy could be caused by:
1) the drought response function of photosynthesis and soil moisture response of soil respiration incorrect
2) inaccuracies in the simulated soil moisture
3) the non-structural carbohydrate pool sizes are not correct

There are two ways in which the drought can influence the photosynthesis in QUINCY. The leaf stomatal control can be constrained by air or soil moisture and additionally there is a direct limitation to reducing the photosynthesis levels by soil water potential. This way to take drought effects into account in the terrestrial biosphere models is common, even though it has been criticized (Sabot et al., 2022).

To unveil these effects, we made plots of the drought responses for the simulations and the observations to see if the challenges would be in the way the drought responses are described in the model. Next, we made more thorough analysis on how well the soil moisture is simulated in dry conditions. Finally, we checked the simulated NSC pool sizes in the model in years 2007 and 2008 against regular years to see if there potentially could be an effect and also assessed if there was an influence on the simulated LAI values.

We added discussion based on this new analysis and hope that this will better answer the request by the reviewer.

4. Table S1 is referred to throughout the manuscript at greater frequency (12 times by my count) than every figure and table in the manuscript except Figure 4 which is also referenced 12 times. Alone, it is referred to more times than all three of the tables included in the manuscript! I would suggest moving this table to the manuscript or otherwise incorporating this information in a manner that does not result in the jarring requirement of frequently referring to the supplement.

We thank the reviewer for this comment. We've moved Table S1 to the main text, as suggested.

5. Figure 2 shows the mean yearly values for NEE, TER, soil temperature and soil moisture. This is discussed in detail. My feeling, considering the soil temperature plot and the location of Borden Forest, is that snow cover may be playing a substantial role at this site over winter. Note that the observed soil temperature maintains a constant temperature through winter, consistent with an insulating snow cover. This may also explain the model's earlier TER response as QUINCY fails to simulate any snowmelt period suppressing soil respiration despite increasing temperatures (Teklemariam et al., 2009). Was this potential role of snow considered?

The reviewer has a good point about the snow cover. The soil temperatures will be influenced by the snow cover and also the springtime recovery of TER will be influenced.

In the original simulations we did not have the snow module turned on (we did not have snow included in the original version of the model we started to work with). To separately assess its effects we did not re-do all our simulations. Instead, we did a simulation with the CN:LAI&chl -version of the model with the snow module activated and now report the influence on the soil temperatures and carbon fluxes.

Technical Comments

1. L20: Delete "the" from before "ecosystems".

Thank you, deleted.

2. L27: Longer growing seasons compared to what?

We modified the text to:

"*At large spatial scales, satellite observations have shown a lengthening of the growing season in deciduous forests in recent decades that has been attributed to warming temperatures.*"

3. L30: Delete "the" from before "vegetation functioning".

Thank you, deleted.

4. L33: Delete "the" from before "forests". Note this is a recurring issue as per comments 1 and 3. Since we are not discussing specific forests or ecosystems, there should not be a definite article "the" in these instances.

Thank you, deleted.

5. L51: Add "an" before "increase" and a "the" before "land carbon sink". Modify "changing of" to "change in".

Thank you, we have corrected the sentence.

6. L61: GPP has already been defined.

Yes, we removed the definition from here.

7. L69: Delete the comma after "Borden Forest" or add a comma after "continuous data".

Thanks, comma deleted.

8. L78-79: This sentence needs to be reformulated to make sense.

Apologies for unclear structure. The original sentence was: "*In our research, the data are combined with a terrestrial biosphere model QUINCY (QUantifying Interactions between terrestrial Nutrient CYcles and the climate system) (Thum et al., 2019), which simulates fully coupled cycles of carbon, nitrogen, phosphorus of terrestrial ecosystems coupled representations of the surface and sub-surface budgets of water and energy.*"

The modified version is: "*In our research, the data are combined with a terrestrial biosphere model QUINCY (QUantifying Interactions between terrestrial Nutrient CYcles and the climate system) (Thum et al., 2019). QUINCY simulates fully coupled cycles of carbon, nitrogen, phosphorus of terrestrial ecosystems together with budgets of water and energy.*"

9. L81: ChlLeaf has already been defined.

Thanks, we removed the definition from here.

10. L83 and throughout: Make sure references are formatted correctly, namely the brackets.

Thanks, we've corrected this line and checked the manuscript thoroughly in this respect.

11. L125: GPP has already been defined.

Thanks, we removed its definition from here.

12. L163: The acronym QUINCY has already been defined.

That's right, we removed the definition from here.

13. L202: Define or explain "OCN".

We have added explanation: "*terrestrial biosphere model OCN*"

14. L228: Delete "an" from before "the slope".

Thanks, deleted.

15. L243: Add "and" between "humidity" and "wind".

Thanks, added.

16. L247: Was a single random year used repeatedly in the spin-up or was it a 500 year time series constructed by randomly selecting a year of [CO2] and meteorology for each of the 500 years?

Apologies for the unclear formulation. We have only used the observed meteorological data during the spinup. For the transient simulations starting in 1901 forwards we created a meteorological forcing file by randomly picking a year of the observed years, with the $CO_2$ concentration varying according to GCP (Friedlingstein et al., 2019). For the spinup we used the first 30 years of this forcing file repeatedly.

17. L249: What data was used for the meteorology between 1901 and 1996 in the transient runs? ERA-5 is mentioned earlier but this dataset begins in 1940.

We've only used the meteorology measured at the site in the spinup. We have clarified this in the text:

"*This was followed by a transient simulation starting in 1901, which was using meteorological data randomly picked from the observed meteorology.*"

Thanks, corrected.

19. The model abbreviations are quite long and complicated – is there a way to condense them while maintaining the information?

Thanks for noting this. We decided to change the "C-only" to "C", as the simulation names will still keep their information content. We also changed "C-only,fix" to Cfix".

20. L272: As I understand it, LAI was a parameter that was calibrated and this can be seen by the improved seasonality of LAI in Fig 1c&d. Was there no capacity to correct the magnitude of LAI which is too high in summer and too low in winter?

We only modified the senescence parameter. The deciduous forests don't have leaves in winter in QUINCY, so the only way we could match the wintertime LAI values would be to have in addition evergreen coniferous species simulation. We decided not to do that, as it would complicate this work considerably (also discussed in reply to Comment 2.)

We don't have a simple way to change the summertime maximum LAI in QUINCY. We could try to change the leaf to root ratio, directing more of the biomass allocation going to roots instead of leaves. However, there were several reasons why we did not do this.

1) Had we lowered the LAI, the simulated GPP would have been even lower.
2) The other leaf level parameters would likely have degraded because of this change.

3) The continuously measured LAI was lower compared to other observations, probably being influenced by location of the sensors. Other estimates for the LAI at the site were more in line with our estimate and we therefore also thought that our simulation result would not be so off.

We have added to the text: "*The wintertime LAI will stay overestimated in our current modelling set-up, as only deciduous forest is being simulated. Changing the allocation pattern in QUINCY would have allowed lowering of the simulated LAI, but this would have led to lower GPP values* ."

21. L286: Figure S2 shows 2013, not 2014.

Thanks, we've corrected the text to the year 2013.

22. L297: "The observations show more shallow decrease" - I think this should be "The observations show a more gradual decline in increasing GPP before the peak" or similar?

Thanks, corrected.

23. L304: "parameterizations is in" should be "parametrizations are in".

Thank you, we've made the correction.

24. L305: It may be worth clarifying that this refers to different simulations within the "C-only" model framework only, not between the "C-only" and "C-only,fix" model simulations.

Thanks, that's a good point. We modified the text to: "*The different C model simulations using dynamic stoichiometry (i.e. not including simulations using fixed stoichiometry, Cfix) did not largely impact RMSE and r² values*."

25. L309: Perhaps specify that the "more accurate representation" is better timing of senescence?

Sure, we modified the text to: "*This resulted in a more accurate representation with better timing of senescence of the observed seasonal cycles of GPP, LAI and Chl$_{Leaf}$ in the simulations*."

26. Figure 1: GPP and LAI have already been defined, and "leaf chlorophyll" has not been defined as "Chl" but as "Chl Leaf". I also do not think that the colors of each model simulation need to be spelled out in the caption as the legend already provides this information. It should be "... leaf chlorophyll WHICH has been smoothed...".

Thank you for the careful check on the figure and the caption. We removed the definitions of GPP and LAI from the caption, corrected the y-axis labels for leaf chlorophyll, removed the explanations for the colors and made the mentioned language correction to the caption.

27. L313: Perhaps a column should be added to Table S1 containing the percentage figures of under/overestimation.

That's a good idea, thanks! It has been added.

28. L316: Should it be "with only a 1.4% larger value"?

Yes, it should be, and we've corrected that now, thanks.

29. L320: I think this should refer to Table S1, not Table S2.

Thank you for noticing this, this has now been corrected.

30. L322: TER has already been defined.

Yes, we removed the definition from here.

31. L330: "Table 1" should be "Table S1".

Thanks, we have now made the reference to the correct table.

32. Figures S3, S4, and S5 are often used to illustrate points that are difficult to parse from the plots (namely referring to fluxes being over/underestimated in certain months). The plots are too noisy for me to clearly and easily identify these statements so I suggest a different method of plotting this data to illustrate the points made.

We thank the reviewer for pointing this out. We modified the figures to show the monthly values instead of the daily values to make the plots less noisy. We also added $r^2$-values for these plots as suggested by Reviewer #2.

33. L339: "The early season pattern observed in the simulated NEE is attributed to heterotrophic respiration". Is not the overestimation of early season NEE due to a GPP that is too low from DOY ~75, as well as TER being too high between DOY 50 and 100? This is what I read from Figure S6.

Sure, the reviewer is right, we have changed this text into: "*The early season pattern observed in the simulated NEE is attributed to too late onset of GPP and heterotrophic respiration...*"

34. L348: Missing bracket after "(Fig. 2."

Thanks, corrected.

35. L355: This sentence is not properly formatted.

Thanks. The original sentence was: "*The continuous observations of LAI provide values (3.78 ± 0.43 m2 m−2 in summer, averaged the over June-July period, along with the standard deviation).*"

The modified sentence is: "*The continuous observations of LAI provide values of 3.78 ± 0.43 m2 m−2 in summer (averaged over the June-July period, along with the standard deviation)."*

36. L365 and surrounding paragraph: This paragraph could be condensed. It is also unnecessary to specify (C-only:LAI&Chl) and (CN:LAI&Chl) after each reference to the models as it has already been stated that these are the models referred to from Line 318 onwards.

Sure. We removed the specifics about the simulation runs and shortened the paragraph, as suggested.

37. L376: This statement about SLA is disjointed from the rest of the text.

Thanks for noting this. We tried to accommodate it better to the text by moving it and modifying it to:

"*In the model the specific leaf area (SLA) is an important factor in changing the leaf biomass to LAI and it is kept constant. In the observations the SLA exhibited a dynamic change, with higher values (~303 cm g$^{-1}$) observed in the early season and a subsequent decline to a summertime value of 162 cm g$^{-1}$ within approximately one month.*"

38. L379: TER is discussed first in Figure 4 but in the plot, GPP comes first. I would ensure these align to improve readability.

Sure, we've made this change now.

39. Figure 4: In this instance, I think it might be better to have the observed and simulated on the same plots, with the different facets instead for the years – as it stands, the figure is better as a comparison between years than for assessing model performance, which I believe to be its main message.

We also wanted to show the behavior in the years 2007 and 2008 against the regular years. We have now modified the plot so that the measurements and observations are shown in the same plots and the regular year behaviour in thin lines.

40. Figure 4 and Figure S9: As in other plots, could the standard deviation be plotted as a shaded area around the observation means?

We modified the Figure 4 according to the comment #39 and also added in the

We have added the standard deviation to the line that is the average of several years in Figure 4.

41. L392: Move the sentences about GPP to the next paragraph which discusses GPP

more widely, rather than in this paragraph which is about TER. Section 3.4 in general could use some work to improve the flow of the section and maintain a narrative throughout instead of the current text which tends to jump between the discussed variables with little reason. Either discuss each variable in turn, or discuss each time period in turn.

Sure, we followed the instructions by the reviewer and moved sentences about GPP. Additionally we worked on improving the flow of this section and discussed each variable in turn, as was suggested.

42. Table 4: The QUINCY LOS (GPP) is incorrect – it should be 146 days if calculated as the difference between the mean SOS and mean EOS as the other table elements seem to imply (or perhaps this is coincidence and it is actually the mean length of each individual growing season, in which case this should be clarified).

We thank the reviewer for such careful investigation of the values. We had added a wrong number to the table for the QUINCY SOS (GPP), it should have been 142. We have corrected this now in the table and text.

43. L417: "LAI based estimates" is misspelled.

Thanks, corrected.

44. L420: "takes place in average" should be "takes place on average".

Thanks, corrected.

45. L429: Clarify what is meant by "make use of different spring and autumn periods".

We have changed this into: "*The real forest with several species might have more resilience to different environmental conditions and therefore different species might be able to benefit from differing environmental conditions during shoulder seasons*."

46. L438: There appears to be a significant breakpoint in GPP around 2009. Are there any explanations for why this might be? Is this a recovery from the 2007 drought? Is there any potential reason why every year post 2010 has higher GPP than any year before? Why were the final 5 years removed as a test?

Gonsamo et al. (2015) studied the trend in GPP at the site and found a significant trend in photosynthetically active radiation and attributed this change to cleaner air. They also saw significant trends in length of the carbon uptake period (CUP) and delay of the ending in the carbon uptake period. The delay in autumn might have to do with different responses to environmental conditions by carbon uptake and respiration. We did a simplified analysis of the CUP and noticed also a significant trend in the ending of CUP and length of CUP in our model simulations for the same time period as Gonsamo did in their observations, but when we extended the analysis period until the end of 2018, the significant trend disappeared.

In the observations the significant increase in GPP happens mostly during summer months, with some increase also taking place during autumn (Table S2). If this change was caused by environmental conditions, such as increasing atmospheric CO2 or air temperature, in theory we should be able to capture this increase by our model.

The increase in site observations could be caused by demographic changes in the forest. There are tree seedlings in the forest floor and as they grow, they might be able to photosynthesize more. Their influence would not be captured by the continuous LAI observations nor our model.

One thing that we had not addressed in the manuscript is the role of nitrogen deposition. Our model input data shows a top value in 1995 at the site and then a continuous decrease until our study period, 2018. This is in line with the sulphur deposition change that has been reported for the Egbert site, that is close to the Borden forest (Gonsamo et al., 2015). The increase in GPP could partly be caused by the recovery of the forest from ozone, nitrogen and sulphur depositions, but it would be expected that this recovery is seen as increasing LAI, which is not seen in the observations. Decreasing nitrogen and sulphur depositions might alter the soil pH, causing alterations to the nitrogen cycle. If we had leaf nitrogen observations over different time periods we might be able to assess this. We'll add these points to the new version of the manuscript.

Removal of the last five years was quite arbitrary. We have mentioned this now in the text.

47. L445: Why is LAI not plotted in Figure S12 if it is discussed here?

We have added LAI to plot Fig. S12.

48. L451: Can the differences in IAV be quantified somehow, for instance comparing the standard deviations?

Sure, we added in here the standard deviations from the Table 3 (earlier Table S1) and noticed an inaccuracy in the earlier formulation, as the standard deviations of GPP and TER are comparable in the observations.

49. L469: Incorrect parentheses for reference.

Thanks, corrected.

50. Section 4.1: I'd like to see more discussion here about how the continuous LAI measurements improved the model performance. What has been learnt from this study and how can these lessons be implemented in the model? Does the senescence parameter need to be modified in the standard implementation of the model? Do we need to test the model tuned with continuous LAI values at other sites?

For the standard implementation of the model, we have tested the parameters at several sites (against GPP observations) when deciding on the default parameters. In the current model development phase the QUINCY model developers use satellite observations at site

level for FLUXNET sites as well as the benchmarking system iLamb for global scale evaluations. Therefore we'd not change parameters based on one site observations. An on-going analysis based on flux tower and remote sensing data (Miinalainen et al., in preparation) shows several other sites with the same plant functional type are overestimating both GPP and (satellite-based) LAI in October, so the results from Borden are in-line with these estimates and with the uncertainties of phenological transition dates of remote sensing observations (Wang et al., 2024), are a valuable site-level verification for these results. We've added to the text:

"*The parameterization of QUINCY relies often on several sites, as the aim is to have successful large scale simulations. However, an on-going study evaluating the QUINCY seasonal cycles with flux tower and remote sensing data (Miinalainen et al., pers comm.) has shown that a similar bias in autumn phenology for temperate broadleaf deciduous occurs at several other sites, so this PFT could benefit from parameter tuning. However, the satellite observations have some issues in predicting the autumn phenology (Wang et al., 2024) and the long-term in-situ observations at the Borden site can act as a valuable verification resource.*"

51. Section 4.2: Again, I think there could be more discussion here about what the simulated ChlLeaf values can teach us regarding QUINCY and future improvements. What does it mean for the model simulations that Chl Leaf peaks early? What might this imply for QUINCY applied at other sites or globally? While not the objective in this study, what might be learnt if we did attempt to capture the timing of the maximum leaf chlorophyll? What would the tradeoffs be?

The peak in the observed LeafChl occurs around DOY 180-200 (Fig 1e). This is not seen in the Vcmax and the Jmax observations (Fig. 3). Our aim is to improve the photosynthesis and despite leaf chlorophyll being part in the photosynthesis calculation of the QUINCY model, the Vcmax and Jmax values play a larger part and the LeafChl is considered to have highest relevance because of its link to these parameters. Increasing LeafChl values for the time period visible in the observations (Fig 1e) would not probably improve our results. We could do sensitivity testing with the canopy module of QUINCY to see the influence on GPP of forcing LeafChl to a higher level for this time period. Considering the model structure, this influence would not be very large.

Earlier studies by simpler approaches have shown a more pronounced influence of using leaf chlorophyll in modelling. Croft et al. (2015) replaced LAI in a light use efficiency (LUE) model by leaf chlorophyll and they obtained a better seasonal cycle of GPP by this change. The r2 of daily GPP from the LUE model using LAI was 0.55 and with leaf chlorophyll 0.65 and with canopy scaled chlorophyll 0.69.

In the study by Luo et al. (2018), the LeafChl was directly used to estimate Vcmax and Jmax and they found improvements in model performance, r2 for GPP in 2013-2014 was increased from basic model formulation 0.84 to 0.91. This study was done by the BEPS model, which is usually run with LAI as input.

Our study was partly inspired by these works, as we're now having a model that explicitly includes ChlLeaf. Our approach has a direct coupling of LeafChl to the nitrogen cycle

modelling, therefore we find this variable very interesting. With both carbon and nitrogen cycles included our approach is more constrained than the modelling approaches in the two earlier studies. The impact we obtained was not as pronounced as illustrated by these two earlier studies. One reason is that our model is constrained by many processes. One difference between our study and the study by Luo et al. (2017) is that we don't have a direct linear relationship between the leaf chlorophyll and Vcmax, as has been shown to be the case at the Borden site (Croft et al., 2015). Despite this, our comparison with the biochemical model parameters showed reasonable values and didn't require changes for the formulation. However, the LeafChl formulation of QUINCY might benefit from linear relationship between LeafChl and Vcmax and this is one of the issues tackled in an on-going study (Miinalainen, T., in preparation), where remotely sensed LeafChl is compared to QUINCY estimates across several flux sites.

A recent study combining chlorophyll fluorescence observations with the LeafChl and biochemical model parameter observations at the Borden site reveal that the different light acclimation rates of Vcmax and LeafChl cause changes in their relationship during different seasons (Yu et al., 2024). We could test the influence of changing these relationships with the canopy version of the model, that is modelling only the leaf canopy, so this would be one potential way to further study this topic.

We added these points to the text by: "*The peak in values in observed ChlLeaf during midsummer was not simultaneously accompanied by increases in the biochemical model parameters Vc(max) and Jmax (Fig. 3), which would have a more pronounced effect on the simulated photosynthesis than ChlLeaf. We could test the effect of increased midsummer ChlLeaf on photosynthesis using the canopy module of QUINCY, which calculates only the canopy part of the model. Given the structure of the model, this influence is likely to be small. Luo et al. (2018) used the fact that ChlLeaf and Vc(max) have a linear relationship at the Borden site to improve their model results for GPP and evapotranspiration. In QUINCY we currently do not have this linear relationship, although it has been observed in some studies (e.g. Qian et al. 2021). A further study using remote sensing data and more sites will explore the implementation of such a description of leaf N partitioning in QUINCY. At the Borden site the linear relationship between ChlLeaf and Vc(max) was found to be off due to their different rates of light acclimation (Yu et al., 2024). Once we have a linear relationship between ChlLeaf and Vc(max) implemented in QUINCY, we can use the canopy module to estimate what effect this would have on photosynthesis in our model.*"

52. L511: "observed transitions is more smooth" should be "observed transitions are more smooth".

Thanks, corrected.

53. L514: I'd change "the model used here" to "QUINCY".

Thanks, we did this change.

54. L515: Missing word between "Testing model performance" and "a TBM designed".

Thanks, corrected now to: "Testing the performance of a TBM designed…"

55. L570: "It occurs most pronounced" is not grammatically correct – replace with "It occurs most prominently" or "It is most pronounced".

Thanks, replaced to: "*It is most pronounced in the year following the drought, 2008 (Fig. S10e).*"

56. L571: "Based on the data available then…" reads as if referring to the 2008 drought due to the prior sentence. I'd recommend "Based on the data available to Lee et al. (1999), they found …" or similar.

Thanks, we used the formulation suggested by the reviewer.

57. L583: Delete comma after "exudates".

Thank you, comma is deleted.

58. L587: Typo in "whereas is was".

Thank you, corrected to "whereas it was."

59. L598: "QUINCY does simulate" should be "QUINCY does not simulate".

Thanks, corrected.

60. L600: There are quite a few statements throughout the manuscript similar to the sentence here: "One additional cause of model failure might be that the canopy light-saturation point does not reflect the observations, however, there is not robust evidence that this is the case." These require at least some explanation of what potential sources of evidence were explored and discounted.

The point we wanted to make here was that the responses to light might not be similar in the model because e.g. we don't have the understory saplings in our model set-up. Since we have made this point elsewhere in the manuscript, we remove this point from here and add discussion about the potential recovery of the forest from nitrogen and sulphur deposition, as discussed in point #46. We will try to identify other such statements and add justification.

61. L605: Add "an" before "impact on the N cycle".

Thanks, added.

62. L619: Add "in" after "cause a change".

Thanks, added.

63. L623: Add "an" before "unrealistically low value".

Thanks, added.

64. L630: Add "the" before "N saturated case".

Thanks, added.

65. L631: Delete "and" before "in line with".

Thanks, deleted.

66. L633: Delete "of" in Section 4.8 header.

Thanks, deleted.

67. L642: Add "A" before "rain gauge".

Thanks, added.

68. L648: Specify that the "long time spans" is referring to long time series of observations.

Thanks, we modified the sentence to: "*This work demonstrated the usefulness of using different data sources and the importance of observational long time series.*"

69. L648: Specify that the "use of leaf chlorophyll content and LAI" is in parametrizing the model.

Thanks, the sentence is now: "*The use of leaf chlorophyll content and LAI in parameterizing the model improved simulated GPP in the CN simulations.*"

70. L658: Change to "attributed to an increase in PAR which is not visible in the shortwave radiation forcing for QUINCY".

Thanks, changed.

71. L661: Delete "is" from before "paves the way".

Thanks, corrected.

References

Guo, J. S., Gear, L., Hultine, K. R., Koch, G. W., & Ogle, K. (2020). Non-structural carbohydrate dynamics associated with antecedent stem water potential and air temperature in a dominant desert shrub. Plant, Cell & Environment, 43(6), 1467–1483. https://doi.org/10.1111/pce.13749

Jones, S., Rowland, L., Cox, P., Hemming, D., Wiltshire, A., Williams, K., Parazoo, N. C., Liu, J., da Costa, A. C. L., Meir, P., Mencuccini, M., & Harper, A. B. (2020). The impact of a simple representation of non-structural carbohydrates on the simulated response of

tropical forests to drought. Biogeosciences, 17(13), 3589–3612. https://doi.org/10.5194/bg-17-3589-2020

Teklemariam, T., Staebler, R. M., & Barr, A. G. (2009). Eight years of carbon dioxide exchange above a mixed forest at Borden, Ontario. Agricultural and Forest Meteorology, 149(11), 2040–2053. https://doi.org/10.1016/j.agrformet.2009.07.011

References

Croft, H., Chen, J., Froelich, N., Chen, B., and Staebler, R.: Seasonal controls of canopy chlorophyll content on forest carbon uptake: Implications for GPP modeling, Journal of Geophysical Research: Biogeosciences, 120, 1576–1586, 2015.

Croft, H., Chen, J. M., Luo, X., Bartlett, P., Chen, B., and Staebler, R. M.: Leaf chlorophyll content as a proxy for leaf photosynthetic capacity, Global change biology, 23, 3513–3524, 2017.

Friedlingstein, P., Jones, M. W., O'Sullivan, M., Andrew, R. M., Hauck, J., Peters, G. P., Peters, W., Pongratz, J., Sitch, S., Le Quéré, C., Bakker, D. C. E., Canadell, J. G., Ciais, P., Jackson, R. B., Anthoni, P., Barbero, L., Bastos, A., Bastrikov, V., Becker, M., Bopp, L., Buitenhuis, E., Chandra, N., Chevallier, F., Chini, L. P., Currie, K. I., Feely, R. A., Gehlen, M., Gilfillan, D., Gkritzalis, T., Goll, D. S., Gruber, N., Gutekunst, S., Harris, I., Haverd, V., Houghton, R. A., Hurtt, G., Ilyina, T., Jain, A. K., Joetzjer, E., Kaplan, J. O., Kato, E., Klein Goldewijk, K., Korsbakken, J. I., Landschützer, P., Lauvset, S. K., Lefèvre, N., Lenton, A., Lienert, S., Lombardozzi, D., Marland, G., McGuire, P. C., Melton, J. R., Metzl, N., Munro, D. R., Nabel, J. E. M. S., Nakaoka, S.-I., Neill, C., Omar, A. M., Ono, T., Peregon, A., Pierrot, D., Poulter, B., Rehder, G., Resplandy, L., Robertson, E., Rödenbeck, C., Séférian, R., Schwinger, J., Smith, N., Tans, P. P., Tian, H., Tilbrook, B., Tubiello, F. N., van der Werf, G. R., Wiltshire, A. J., and Zaehle, S.: Global Carbon Budget 2019, Earth Syst. Sci. Data, 11, 1783–1838, https://doi.org/10.5194/essd-11-1783-2019, 2019.

Sabot, M. E. B., De Kauwe, M. G., Pitman, A. J., Medlyn, B. E., Ellsworth, D. S., Martin-StPaul, N. K., et al. (2022). One stomatal model to rule them all? Toward improved representation of carbon and water exchange in global models. Journal of Advances in Modeling Earth Systems, 14, e2021MS002761. https://doi.org/10.1029/2021MS002761

Luo, X., Croft, H., Chen, J. M., Bartlett, P., Staebler, R., and Froelich, N.: Incorporating leaf chlorophyll content into a two-leaf terrestrial biosphere model for estimating carbon and water fluxes at a forest site, Agricultural and Forest Meteorology, 248, 156–168, 2018.

Miinalainen, T., Ojasalo, A., Zaehle, S., Croft, H., Aurela, M., Peltoniemi, M., Caldararu, S. and Thum, T. Evaluating a terrestrial biosphere model carbon and nitrogen cycles via leaf chlorophyll content and remote sensing observations, in preparation.

Qian, X., Liu, L., Croft, H., & Chen, J. (2021). Relationship between leaf maximum carboxylation rate and chlorophyll content preserved across 13 species. Journal of Geophysical Research: Biogeosciences, 126, e2020JG006076. https://doi.org/10.1029/2020JG006076

Yu, L., Luo, X., Croft, H., Rogers, C. A. & Chen, J. M. (2024) Seasonal variation in the relationship between leaf chlorophyll content and photosynthetic capacity. *Plant, Cell & Environment*, 47, 3953–3965. https://doi.org/10.1111/pce.14997

Wang, C., Yang, Y., Yin, G., Xie, Q., Xu, B., Verger, A., et al. (2024). Divergence in autumn phenology extracted from different satellite proxies reveals the timetable of leaf senescence over deciduous forests. *Geophysical Research Letters*, 51, e2023GL107346. https://doi.org/10.1029/2023GL107346

---

## Author Comment (AC2)

Borden paper review: Reply to Reviewer #2

(The comments by the reviewer are in magenta, the replies to the comments are in black, new text added or modified in the manuscript is written in italics.)

General comments

The study of Thum et al. investigates carbon and nitrogen interactions in the Ontario's Borden Forest Research Station, using in-situ measurements to parameterize the QUINCY model. It evaluates carbon flux simulations over 22 years, finding good alignment in some metrics like GPP but identifying key discrepancies in ecosystem respiration trends and legacy drought impacts, underscoring the need to improve TBMs.

The manuscript addresses an important topic: the representation of carbon and nitrogen interactions in TBM models. Overall, it is well-written, and the figures—despite a few editorial issues—are clear and effectively support the results presented. However, I found the study's objectives difficult to discern from the Abstract. The Introduction also requires substantial revision, as it sometimes lacks logical flow, and the paragraphs don't fully cohere. For instance, the first paragraph focuses on the importance of nitrogen, followed abruptly by a discussion on changes in the growing season due to warming, and then by a mention of the value of long-term observations for capturing anomalies (anomalies of what?). The Introduction feels like a series of loosely connected points without a clear narrative thread, which makes it challenging to understand the paper's aim until the objectives are listed in the final paragraph.

We thank the reviewer for finding the topic important and for all the in-depth comments that we think will improve the quality of the manuscript. We apologize for the shortcomings of the introduction and will improve its flow, following the guidance given by the reviewer.

Regarding the content of the paper, there are two key aspects that should be addressed:

**1. Quality of Eddy Covariance Flux Post-Processing:**
The study relies heavily on estimates of GPP and TER derived from eddy covariance measurements, which are not directly measured values. It is essential to assess the quality of the partitioning and gap-filling methods used. A study from 2004 is referenced for both flux partitioning and gap filling, yet there is no description of this approach, its advantages, or why it was chosen. Given that GPP estimates are highly sensitive to partitioning methods, it's critical to first establish a solid foundation, showing that the best possible gap-filled and partitioned fluxes were derived before making further interpretations about model structure or other underlying factors.

We thank the reviewer for this insight. We have added an explanation about the method developed by Barr et al. (2004) and would like to mention that it is the standard method for the Fluxnet-Canada sites (Pierrat et al., 2021). We have added this point now to the manuscript. This is also the method that has been used for the site in several earlier publications.

The post-processing of eddy covariance data is a challenging task and many decisions need to be made during it. We find that to test the different partitioning and gap-filling methods would be a study on its own. There are several papers from the site that have more concentrated on the observations and how they have been processed (e.g. Froelich et al., 2015). Our aim with this study was to use those data for model evaluation.

Studies that have compared gap-filling methods, have not found them to introduce large differences (Mahabbati et al., 2021). A study comparing partitioning methods found the influence on the annual balances to be less than 10 % (Desai et al., 2008), modest differences have also been found in other studies (Moffat et al., 2007).

Another difficulty in assessing what would be the best gap-filling and partitioning method is that we don't know what the truth is. Therefore studies using synthetic data are useful, since then the 'truth' is known. We've been involved and shared our model results from other sites to such a study and fully support these investigations (Vekuri et al., 2024). The field of studying partitioning methods is an exciting one with new ecosystem functioning related findings (Wohlfahrt and Galvagno, 2017) and applications of machine learning (Pastorello et al., 2020), but we consider applying these different methods to be outside the scope of our study.

While the gap-filling and partitioning do influence the annual balance values, for our purposes the seasonality of these fluxes was more on focus. If the annual balance values would shift a bit, the conclusions of our study would not change. We have now added to the discussion text on the uncertainty that the gap-filling and partitioning method cause to the measurements:

"*Another source of uncertainty in the observations are the use of gapfilling (Mahabbati et al., 2021) and partitioning methods (Desai et al., 2008), that cause uncertainty in the annual carbon balance estimates.*"

2. Snow Cover Effects:

Considering the site's geographic location, persistent winter snow cover is likely; if this is not the case, it should be explicitly mentioned. However, the manuscript does not address snow cover or its potential impact on carbon flux processes. This is especially relevant in the shoulder seasons, where the authors discuss discrepancies between modeled and observed fluxes. Factors such as snow effects, soil freezing and thawing, and changes in soil-air temperature decoupling due to snowmelt could significantly influence these processes but are omitted in the analysis.

We thank the reviewer for bringing this point up. The original model simulations did not have snow included (this was partly because of 'historical' reasons: QUINCY did not have snow included when this work started). We agree with the reviewer that it's a relevant process at this site and added a simulation with the snow, analyzed the snow depth against site level observations and its influence on the carbon fluxes.

Below, I provide more specific comments on the manuscript.

Line 10: Please also report the RMSE (and not just r2) when reporting the model performance.

This has now been added.

Line 10: You mentioned how you parameterized the model but not how the model was improved.

Thanks for noticing this, the phrasing had been unclear with "*The improved model captured observed daily gross primary production (GPP) well.*"

We changed this into:
"*Model with the improved parameterization captured observed daily gross primary production (GPP) well.*"

Line 11: Would be interesting to know the magnitude of this increase

Sure, we added the magnitude of the trend here.

Line 11: NEE not defined yet

Thanks, definition added.

Line 35: Grammar ("are" missing)

Thanks, added.

Line 42: Would be important to add here why the current representation of N limitation of photosynthesis is not sufficient (since this is such a key aspect in this paper)

The claim here originally was not that the representation of N limitation is not sufficient, it's that the models have different approaches that lead to different outcomes. We have added some more text here describing these differences:

"*N limitation may directly affect photosynthesis rates or its effects may be buffered via different stoichiometric related implementations (Thomas et al., 2015) and the different hypotheses and parameter values related to N cycle processes lead to differences between models (Medlyn et al., 2015).*"

Line 43: For example, which responses?

We've added here: "*A model intercomparison study of five CMIP6 models showed a wide range of response in net primary productivity for increased atmospheric $CO_2$ and atmospheric N deposition (Davies-Barnard et al., 2020).*"

These aspects are important to be clarified to convince the readers why "it is paramount that

the effects of N constraints on plant productivity are accurately simulated".

We agree with the reviewer and are thankful for these remarks.

Line 50 and 52: Do these observations overlap? Both in terms of time and space? How is the
increase in LAI and the decline in N related?

The study by Chen (2019) uses global remote sensing data starting in 1981 and reaching up to 2016. The responses for central Europe show divergent trends in LAI development (Fig. 1), mostly small or larger increases. Jonard et al. (2015) noticed lowering of leaf N at the European forests during 1992–2009, so the studies do overlap temporally.  The Jonard et al. (2015) study also included foliar mass observations and for some tree species significant increasing trends were included. They conclude that increases in tree productivity have led to higher nutrient demand by trees and as soil nutrient supply is not enough to meet this demand, the tree mineral nutrition deteriorates. Mason et al. (2022) have reported a decline in N availability in many terrestrial ecosystems, supporting also conclusions by Jonard et al.

We have added this point to the text by:

"*Climate change induced changes have caused a increases in LAI (Chen et al., 2019) and tree productivity (Jonard et al., 2015), and the changes in availability and demands for N have been leading to declining N availability in respect to demands in terrestrial ecosystems (Mason et al., 2022).*"

Line 79: Grammar

Thank you for noticing this. We have rephrased this sentence as the reviewer #1 also commented on this.

Section 2.1: Please add a description of the understory vegetation (species and cover). This information is relevant to your discussions of model limitations that we read later in the paper.

Sure. Unfortunately the exact species haven't been determined at the site to our knowledge. We added to the section the following text: "*The understory consists of short ferns, small shrubs and saplings (Halliday, 2010).*"

Lines 101-104: Is this the species composition within the flux footprint?

There is cropland in the northwest direction and these data were excluded and gapfilled. This has been explained in Section 2.2.1.

Line 107: Mean over what period?

Mean over 2000-2014, added .

Line 118: Used for what? For calibration of the model? For validation?

The $CO_2$ fluxes were used for model evaluation. The modified sentence is now: "*$CO_2$ flux data from half-hourly eddy covariance measurements sampled at Borden Forest tower at 33 m height between 1996 and 2018 were used for model evaluation.*"

Line 126: A brief description of the gapfilling and partitioning method should be given here and the justification why such method is selected over other existing methods.

We have added in here the point that this is the standard method used by Fluxnet-Canada (Pierrat et al., 2021) and have added a further explanation on the method as:

"*The procedure first derives the component fluxes from NEE and then uses simple empirical models constrained by the measured data for one year at a time. The other empirical relationship is between TER and soil temperature at shallow depth and the other is between GPP and photosynthetically active radiation (PAR) above the canopy. Parameters for this empirical relationship are first obtained for the annual analysis and after that one parameter per relationship is allowed to vary over time while other parameters stay constant. These time-varying parameters are determined by a flexible moving window approach*."

According to Barr et al 2004 "seasonal onset, rise and fall of photosynthesis from the FNEP time series based on the parameter Px in the rectangular hyperbolic model "was this the case too in this study?

No, here a bit different formulation for this equation was used. We will add a reference to the PhD thesis of C. Rogers.

Thanks, we changed the original sentence ("*measured from instruments on the flux tower*") to: "*measured by instruments on the flux tower*".

The scaling was very simple. The difference between the annual precipitation from the Egbert weather station and the ERA5-Land product was estimated and this scalar was used to multiply all the values in the ERA5-Land dataset. We will clarify this in the text.

How well do the ERA5-Land precipitation product and measured precipitation at the site compare? Perhaps a comparison can be added to the supplementary.

So, we don't have precipitation measurements exactly from the site, but from the nearby Egbert site. We will add one comparison figure to the supplement, as suggested by the reviewer.

Thanks, corrected 'metres' to 'meters'.

The line is "*The soil profile consists of 15 layers, reaching a depth of 9.5 metres. The depth of each layer layer increases exponentially as it*"

Line 274: Based on what was this level of T selected for the adjustments?

The selection was based on the best match of the simulated LAI to the observed LAI.

The original sentence was: "*To adjust the seasonality of LAI, the parameter controlling leaf senescence ($t\_\{air\}^\{sen\}$) was modified from the default value of 8.5 C to 15.0 C.*"

We modified this into: "*The autumn decline of the simulated LAI was adjusted to match the observations by modifying the parameter controlling leaf senescence ($t\_\{air\}^\{sen\}$) from the default value of 8.5 C to 15.0 C.*"

There is a typo just before the caption of Table 2.

The reviewer might be referring to a "t" that has appeared in the pdf-version of this file. It's not in the original file, but likely a result of the pdf conversion.

Figure S2 has a typo in the legend of panel b (should be LAI not GPP)

Thanks, corrected.

In Figure S2 the panels already show GPP or LAI. I would not repeat this again in the legend. Also, the dashed lines can be removed from the legend (to make it less crowded) and keep their description in the caption.

Thank you for these remarks, they have been taken into account in the modified figure.

Figure S3- 5 the axis label should be "modelled" to be consistent with the x-axis which says "Observations". The figure title is already mentioning which model was used, not needed to mention this in the title and in the axis label.

Sure, that's a valid point, we have made this change.

Figure S3- 5 wouldn't it make more sense to display mean daily temperature as the third value rather than the number of the month? What is the reasoning to use month here? Maybe can briefly be added to the caption.

Also using daily temperature as a color code would be a good idea here. Based on these comments and the ones from reviewer #1, we decided to show the monthly values instead of the daily values, to make the figures easier to interpret and to deliver the message that we're using these to make the point that different parameterizations influence the monthly values.

Figure S3- 5 it is hard to judge quantitatively the comparison of different model performances. Maybe at least the r2 values can be displayed in the panels?

We agree with the reviewer. We have added the $r^2$-values in the panels, as suggested.

Figure S8 very hard to read the figure. Consider increasing the font please.

Apologies for the unclear plot. We have increased the font and improved its readability by shortening the titles of the subplots.

Line 339-345: Could this inaccuracy in modelling the soil temperature be because of the snow? What is the contribution of snow cover at this site? We see from colder sites that snow has an insulating effect on the soil that decouples its temperature from air temperature. If direct measurements are not available at the site perhaps you could explore available remote sensing products (e.g., MODIS/Terra (MOD10A2) and MODIS/AQUA (MYD10A2) (Hall and Riggs 2021) Snow Cover 8-Day L3 Global 500m SIN Grid, Version 6 dataset, which provides maximum snow cover extent at 8-day temporal resolution and 500m spatial resolution).

We made an additional simulation including snow and also show a comparison of snow depth against observations. Comparison against observations show that indeed the snow has a role in the springtime soil temperatures and that the model simulation is improved in this respect.

We're planning to do a more in-depth study of snow effects in a separate study including several boreal sites and using remote sensing data of snow cover data (Nagler et al., 2022) and freeze-thaw data from SMOS (Rautiainen and Holmberg, 2023), expanding results from Böttcher et al. (in preparation).

Line 460-464: Here the Results are repeated. Instead, there should be a discussion of what underpins the observation that although modelled LAI is overestimated, modelled GPP in summer is underestimated.

Sure, we'll do this.

Section 4.2 I suggest dividing parameters by structural from photosynthetic traits.

Section 4.2 discusses leaf chlorophyll and specific leaf area (SLA). SLA can be considered to be a structural trait, as it influences the LAI. As this section only discusses these two parameters in follow-up paragraphs, it was a bit unclear how to further divide these if we don't aim for very short sections.

Line 491-493: What explains it if the SLA was overestimated but GPP underestimated?

SLA changes the leaf carbon pool to LAI. The simulated LAI for the site is too large, if compared to the continuous observations, but compared to in situ -observations, it is close to the observed values. Even though the LAI is important in calculating the GPP, the figures 1 and 2 make the point that in the C-simulations the GPP increases compared to CN-simulations (25 % larger annual values, Table 3) because of the high leaf N content, not only because of the LAI (10 % larger, Table 4).

The underestimation of GPP is not pronounced in the early years of the time period, but becomes more pronounced in the later years, contributing to the 17 % underestimation in the annual values (Table 3), which is still within the measurement uncertainty. If we'd only consider the first five years, the observations estimate for annual GPP balance 1367 g C m$^{-2}$ and the simulations 1222 g C m$^{-2}$, an underestimation of 11 %.

Line 500: "instead, we are estimating the tree traits per average individual for a deciduous forest." Not clear to me what this statement means. Because the methods section (2.2.3) only mentions "we used a species-weighted canopy average of the leaf-level parameters, based on the species composition of the forest „ and is not clear how the parameters are weighted and then aggregated (?). Adding a mathematical description here would be very helpful.

We apologize for unclear expression in Section 2.2.3. We're only having traits for one tree in the model, that is representing the whole forest.

The earlier unclear sentence was:
"*Our modelling approach does not allow for species separation; instead, we are estimating the tree traits per average individual for a deciduous forest*."

We re-worded the unclear sentence to:
"*Since we do not have the abiliity to model different tree species, we model a deciduous forest composed of trees with identical traits*."

We have now added in a table of the values for different species to the supplemental material and an equation showing how we've calculated the species-weighted average. This is the same approach that has been used in earlier studies making use of these data (Croft et al., 2015, Luo et al., 2018). We haven't really aggregated the values, but we have smoothed the lines for plotting purposes. The only purpose we used these lines was in delaying the leaf chlorophyll development. For comparison of other traits we only used summertime averages.

Line 510: Which drought occurrences?

We refer here to drought periods taking place in late summer. We modified the text to:

"*The increase in the simulations is more abrupt to the summer levels and decline from early summer values occurs quite early, probably due to dry periods occurring during summer*."

Line 515-517: Grammar check and re-writing needed. Sentence is not clear.

The original sentence was: "*Testing model performance a TBM designed for large-scale simulation at site-level is challenging as the model necessarily needs to apply generalizations in process representation in order to have a model that can be applied across sites and at large scales, due to limited knowledge and data needed for large-scale parameterization*."

We have modified this into: "*Testing the performance of a TBM designed for large-scale, site-level simulation is challenging. The model must necessarily make generalizations in the process representation due to the limited knowledge and data required for large-scale parameterization.*"

Line 525: "The tree species composition has undergone changes at the site during our study period, e.g., the red maple was reported to have coverage of 36 % in 1995 (Lee et al., 1999) and 52 % in 2006 (Teklemariam et al., 2009). The impacts that these changes in the tree composition have on the carbon fluxes could be studied by a demographic model with sufficient granularity in the description of tree functional diversity"
Since species compositional shift was not really addressed in this study would suggest to remove this context from the Introduction as currently it reads as if this is one of the aspects this paper addresses.

We did not find a statement regarding to this in the Introduction, but it was prominent in the Abstract with the line: "*However, how carbon and nitrogen interactions affect both carbon fluxes and plant functional traits in dynamic ecotones, which are experiencing disturbance and species compositional shifts remains unclear.*" and perhaps this was what the reviewer had in mind.

We have modified this into: "*However, how carbon and nitrogen interactions affect both carbon fluxes and plant functional traits in dynamic ecotones, which are experiencing biotic and abiotic changes remains unclear.*"

Section 4.4 it is not clear if the model fails to simulate the (legacy) effect of drought because of its structure (representation of the carbohydrate pools) or the lack of precise soil moistures
estimations which directly affect modelled $CO_2$ fluxes. Could this not have been specifically tested (for this particular case of drought conditions) by first calibrating the model using observed soil moisture measurements and testing whether GPP simulations improved during drought? If the model´s limitations during drought is a focus of this paper it deserves a more systematic approach to test this.

We agree with the reviewer, that this topic would benefit from a deeper dive into it. However, calibrating the model by using observed soil moisture measurements is not straightforward. Instead we made a more thorough analysis to see if the lack of drought effect is caused by: insufficient soil moisture description, wrong drought response of the carbon fluxes or insufficient influence of drought in the non-structural carbohydrate pool. We will include this analysis in the new version of the manuscript.

Line 556: What is meant here? Isn't a depth-resolved soil texture provided to the model? Otherwise, what description of the soil physics exactly is lacking here?

We don't have a depth-resolved soil texture in the model, we are here referring to needs to change model structure. Without a deeper analysis it is difficult to say whether we'd need changes in the water-retention curve, infiltration properties, pedotransfer functions or something else.

We apologize for unclear formulation and have changed the sentence into: "*The drought response at the site could potentially be improved by calibrating the soil moisture response functions in the model, but probably some structural changes in the description of soil physics would also be required. These changes might involve changes in water-retention curve, pedotransfer functions (Weber et al., 2024) or infiltration properties (Vereecken et al., 2019).*"

Line 570-576: Yes, this hysteresis in TER response to temperature (early season lower respiration than late season) could reflect the seasonal patterns of photosynthate allocation to roots. Tree girdling studies have shown that seasonal pattern of below-ground C allocation may be more important than soil temperature in determining root respiration (see for example Högberg et al. 2003) and that earlier in the season respiration from the soil is mostly due to heterotrophic respiration.

We thank the reviewer for this point and have added the mentioned reference to the text.

Line 591: Increase over time? Sentence reads incomplete.

Apologies for the incomplete sentence and thanks for noticing it. The original sentence was: "*Froelich et al. (2015) found a significant increase in summertime GPP and Gonsamo et al. (2015) significant increase in carbon uptake.*"

We have modified this into: "*Froelich et al. (2015) found a significant increase in summertime GPP and Gonsamo et al. (2015) significant increase in carbon uptake between 1996 and 2012.*"

Line 597: So the PAR that was used as an input to the model was not correct? The model would clearly use PAR, so why give it the wrong forcing?

We have used shortwave radiation, not PAR, as the input for the model, and these two variables have been measured by different sensors. The trend in PAR reported in the Gonsamo et al. (2015) was significant, as was stated in the paper. However, when looking at the PAR dataset more closely (we are doing this for the PAR available from the AmeriFlux database, not the gapfilled data from the Gonsamo study), we noticed that the increasing trend was less than 1 % in a year for the time period of their study (1996-2012). Having this kind of increase in model forcing would not lead to that kind of strong trends in GPP as seen in the observations. Furthermore, when we calculated the trend for our study period (1996-2018), the trend was not anymore significant.

In the forcing that we used, the shortwave radiation, we instead had a small declining trend. Since the trends in these radiation variables are so small, we wouldn't expect them to be the cause for large increasing trends in GPP. We instead hypothesize in the current version of the manuscript that the role of understory with potential other effects, e.g. declining nitrogen and sulphur deposition rates.

Line 603: You mean at this site? It is not clear how these findings are relevant to this study.

Apologies for unclear impressions. The study by Gonsamo et al. has been done at two sites (Harvard and Borden) and these conclusions were true for both sites. We have now clarified in the text, that these results are for the Borden sites.

Line 600-607: The discussion on the ozone effect comes out of blue here, unless my previous comment is clarified.

Yes, thanks, we hope the new version of the text makes it clearer in this aspect.

Lien 614: What is meant by "differences between the annual GPP and carbon balance"?

We apologize for the unclear statement. We've corrected this to:

"*It is interesting to note that the differences between the measured and simulated annual GPP are not apparent in the early years of the record, but emerge in the later years (Fig. S12)*."

Line 619 and 620: Grammar

Thank you for noting this. The original sentence was: "*Including the nitrogen cycle in the simulations did not cause a change the net carbon balance of the ecosystem, …*"

We corrected this to: "*Including the nitrogen cycle in the simulations did not cause a change in the net carbon balance of the ecosystem, …*"

Section 4.8: I would not finish the manuscript on listing technical shortcomings. What have we learnt from long term ecological response of such an ecosystem, observations and model results combined? And what is the outlook under further changes in the climate? (e.g., predicted potential increase in temperature and dryness).

The idea of section 4.8 was not to list technical shortcomings (for that we have Section 4.3), but to describe what kind of new observations would be beneficial to better understand the processes at the site.

However, following the idea of the reviewer, we have added a new section 4.9, that discusses the points mentioned by the reviewer.

The Conclusion section can be shortened.

We've shortened the Conclusions.

References

Hall, D. K. and G. A. Riggs. MODIS/Terra Snow Cover 8-Day L3 Global 500m SIN Grid, Version 6. Distributed by NASA National Snow and Ice Data Center Distributed Active Archive Center (2021).

Högberg, P., Nordgren, A., Buchmann, N. et al. Large-scale forest girdling shows that current

photosynthesis drives soil respiration. Nature 411, 789–792 (2001). https://doi.org/10.1038/350810585

References

Böttcher, K. Thum, T., Aurela, M., Rautiainen, K., Holmberg, M., Johnson, B., Koponen, S., Plummer, S. & Pulliainen, J. 2024. Influence of cryosphere dynamics on carbon fluxes in needleleaf boreal forest. In preparation for Remote Sensing of Environment.

Chen, J.M., Ju, W., Ciais, P. *et al.* Vegetation structural change since 1981 significantly enhanced the terrestrial carbon sink. *Nat Commun* **10**, 4259 (2019). https://doi.org/10.1038/s41467-019-12257-8

Croft, H., Chen, J., Froelich, N., Chen, B., and Staebler, R.: Seasonal controls of canopy chlorophyll content on forest carbon uptake: Implications for GPP modeling, Journal of Geophysical Research: Biogeosciences, 120, 1576–1586, 2015.

Davies-Barnard, T., Meyerholt, J., Zaehle, S., Friedlingstein, P., Brovkin, V., Fan, Y., Fisher, R. A., Jones, C. D., Lee, H., Peano, D., Smith, B., Wårlind, D., and Wiltshire, A. J.: Nitrogen cycling in CMIP6 land surface models: progress and limitations, Biogeosciences, 17, 5129–5148, https://doi.org/10.5194/bg-17-5129-2020, 2020.

Froelich, N., Croft, H., Chen, J. M., Gonsamo, A., and Staebler, R. M.: Trends of carbon fluxes and climate over a mixed temperate–boreal transition forest in southern Ontario, Canada, Agricultural and Forest Meteorology, 211, 72–84, 2015.

Halliday, M., 2010. Correlation between sonic anemometers at three heights within a mixed temperate forest. SURG Journal 3, 2291-1367, doi: 10.21083/surg.v3i2.1106.

Luo, X., Croft, H., Chen, J. M., Bartlett, P., Staebler, R., and Froelich, N.: Incorporating leaf chlorophyll content into a two-leaf terrestrial biosphere model for estimating carbon and water fluxes at a forest site, Agricultural and Forest Meteorology, 248, 156–168, 2018.

Jonard, M., Fürst, A., Verstraeten, A., Thimonier, A., Timmermann, V., Potočić, N., Waldner, P., Benham, S., Hansen, K., Merilä, P., Ponette, Q., de la Cruz, A.C., Roskams, P., Nicolas, M., Croisé, L., Ingerslev, M., Matteucci, G., Decinti, B., Bascietto, M. and Rautio, P. (2015), Tree mineral nutrition is deteriorating in Europe. Glob Change Biol, 21: 418-430. https://doi.org/10.1111/gcb.12657

Mahabbati, A., Beringer, J., Leopold, M., McHugh, I., Cleverly, J., Isaac, P., and Izady, A.: A comparison of gap-filling algorithms for eddy covariance fluxes and their drivers, Geosci. Instrum. Method. Data Syst., 10, 123–140, https://doi.org/10.5194/gi-10-123-2021, 2021.

Mason, R. E., Craine, J. M., Lany, N. K., Jonard, M., Ollinger, S. V., Groffman, P. M., Fulweiler, R. W., Angerer, J., Read, Q. D., Reich, P. B., Templer, P. H., and Elmore, A. J.: Evidence, causes, and consequences of declining nitrogen availability in terrestrial ecosystems, Science, 376, eabh3767, https://doi.org/10.1126/science.abh3767, https://www.science.org/doi/10.1126/science.abh3767, 2022.

Medlyn, B., Zaehle, S., De Kauwe, M. *et al.* Using ecosystem experiments to improve vegetation models. *Nature Clim Change* **5**, 528–534 (2015). https://doi.org/10.1038/nclimate2621

Moffat, A. M., Dario Papale, Markus Reichstein, David Y. Hollinger, Andrew D. Richardson, Alan G. Barr, Clemens Beckstein, Bobby H. Braswell, Galina Churkina, Ankur R. Desai, Eva Falge, Jeffrey H. Gove, Martin Heimann, Dafeng Hui, Andrew J. Jarvis, Jens Kattge, Asko Noormets, Vanessa J. Stauch, Comprehensive comparison of gap-filling techniques for eddy covariance net carbon fluxes, Agricultural and Forest Meteorology, Volume 147, Issues 3–4, 2007, https://doi.org/10.1016/j.agrformet.2007.08.011

Nagler, T., Schwaizer, G., Mölg, N., Keuris, L., Hetzenecker, M., & Metsämäki, S., 2022. ESA Snow Climate Change Initiative (Snow_cci): Daily global Snow Cover Fraction - snow on the ground (SCFG) from MODIS (2000-2020), version 2.0. In: NERC EDS Centre for Environmental Data Analysis

Pierrat, Z., Nehemy, M. F., Roy, A., Magney, T., Parazoo, N. C., Laroque, C., et al. (2021). Tower-based remote sensing reveals mechanisms behind a two-phased spring transition in a mixed-species boreal forest. _Journal of Geophysical Research: Biogeosciences_, 126, e2020JG006191. https://doi.org/10.1029/2020JG006191

Rautiainen, K., & Holmberg, M., 2023. SMOS Freeze and Thaw Processing and Dissemination Service, Algorithm Theoretical Baseline Document, ESRIN Contract Nro: 4000124500/18/I-EF. In (p. 18): Finnish Meteorological Institute (FMI)

Thomas, R.Q., Brookshire, E.N.J. and Gerber, S. (2015), Nitrogen limitation on land: how can it occur in Earth system models?. Glob Change Biol, 21: 1777-1793. https://doi.org/10.1111/gcb.12813

Tramontana G, Migliavacca M, Jung M, et al. Partitioning net carbon dioxide fluxes into photosynthesis and respiration using neural networks. _Glob Change Biol_. 2020; 26: 5235–5253. https://doi.org/10.1111/gcb.15203](https://doi.org/10.1111/gcb.15203

Vekuri, Henriikka and Tuovinen, Juha-Pekka and Kulmala, Liisa and Aurela, Mika and Thum, Tea and Liski, Jari and Lohila, Annalea, Improved Uncertainty Estimates for Eddy Covariance-Based Carbon Dioxide Balances Using Deep Ensembles. Available at SSRN: https://ssrn.com/abstract=4999973

Vereecken, H., Weihermüller, L., Assouline, S., Šimůnek, J., Verhoef, A., Herbst, M., Archer, N., Mohanty, B., Montzka, C., Vanderborght, J., Balsamo, G., Bechtold, M., Boone, A., Chadburn, S., Cuntz, M., Decharme, B., Ducharne, A., Ek, M., Garrigues, S., Goergen, K., Ingwersen, J., Kollet, S., Lawrence, D.M., Li, Q., Or, D., Swenson, S., de Vrese, P., Walko, R., Wu, Y. and Xue, Y. (2019), Infiltration from the Pedon to Global Grid Scales: An Overview and Outlook for Land Surface Modeling. Vadose Zone Journal, 18: 1-53 180191. https://doi.org/10.2136/vzj2018.10.0191

Weber, T. K. D., Weihermüller, L., Nemes, A., Bechtold, M., Degré, A., Diamantopoulos, E., Fatichi, S., Filipović, V., Gupta, S., Hohenbrink, T. L., Hirmas, D. R., Jackisch, C., de Jong van Lier, Q., Koestel, J., Lehmann, P., Marthews, T. R., Minasny, B., Pagel, H., van der Ploeg, M., Shojaeezadeh, S. A., Svane, S. F., Szabó, B., Vereecken, H., Verhoef, A., Young, M., Zeng, Y., Zhang, Y., and Bonetti, S.: Hydro-pedotransfer functions: a roadmap for future development, Hydrol. Earth Syst. Sci., 28, 3391–3433, https://doi.org/10.5194/hess-28-3391-2024, 2024.

Wohlfahrt, G. and Galvagno, M., 2017. Revisiting the choice of the driving temperature for eddy covariance $CO_2$ flux partitioning, Agricultural and Forest Meteorology, 237–238, 135-142.

---

## Author Response (AR2)

Replies to Reviewer, 13.1.2024

Reply to Reviewer #1

(The comments by the reviewer are in magenta, the replies to the comments are in black, new text added or modified in the manuscript is written in italics.)

General Comments

In the manuscript "Modelling decadal trends and the impact of extreme events on carbon fluxes in a deciduous temperate forest using the QUINCY model", Thum et al. explore how using in-situ measurements of vegetation traits to parametrize the nitrogen cycle-enabled QUINCY model improves flux simulations at the Borden Forest flux tower. The QUINCY model is modified to allow for a delay in leaf chlorophyll development and the model runs produced in nine different simulations of varying parametrization and nutrient cycle implementations.

When using locally calibrated parameters and enabling the nitrogen cycle in addition to the carbon cycle, QUINCY performed better against observations, especially for GPP, but still lacked some key behaviors – namely it failed to capture drought response and legacy.

Overall, the manuscript is of a high quality, presenting a well-developed study in a clear and scientific manner. It is a substantial contribution to the literature. I recommend publication following minor revisions.

We thank the reviewer for the positive insight towards our study and thank for the detailed, in-depth comments that have helped us to improve the manuscript.

Specific Comments

1. I like the specific statement of the research questions in the study at the end of the Introduction. However, these are not referred to again. I would suggest a section of the Discussion is reformulated to explicitly restate these questions and then explore the evidence found for each one. This would provide a good narrative throughout the manuscript and help in synthesizing the findings of the study.

This is a good idea and we have written section 4.9, that addresses the research questions again at the end of the discussion.

2. Among the limitations of the study is the level of representativeness of the chosen PFT for the flux tower site. According to my reading of Thum et al. (2019) and its supplementary material, QUINCY has the capacity to model an individual gridcell as nested tiles of different PFTs. Does this not provide the necessary flexibility for QUINCY to represent a more accurate mix of vegetation when modelling Borden Forest? What was the inter-species variability in the leaf-level parameters and how much did the species-weighted average differ from the species level values?

To run QUINCY at the site level, we currently can only describe one PFT at a time for one simulation. Therefore for sites with clearly two PFTs (e.g. tree-grass savanna) we have run simulations with two different PFTs. This approach has many challenges, as there is no way for these simulations to communicate and the simulated ecosystems would end up with different soil hydrological conditions, for example, and we would have to add up the results from these two simulations, thus making an analysis more difficult. We didn't have leaf-level observations on the evergreen needleleaf, so we couldn't have compared our modelling results against observations. For the deciduous tree species we only had leaf level observations, so we could have only parameterized the leaf chlorophyll, the LAI observations were for the whole forest. Note that QUINCY does not resolve species, and therefore has limited capacity to account for inter-species variability.

We have added a Table S1 showing the species-weighted average with species-specific leaf-level parameter values. Red maple, with 60.5 % coverage of deciduous species, had generally similar values to the species-weighted average for leaf nitrogen and the biochemical model parameters, while trembling aspen had generally large values when compared to the species-weighted average.

3. Studies have shown that a simple representation of carbohydrate pools can help to model drought legacies (Guo et al., 2020; Jones et al., 2020). QUINCY has a representation of these NSC pools and so might be expected to capture the drought legacy in 2008. As such, I'd like to see Section 4.4 in the discussion expanded to discuss in more detail, or at least more focused on, the reasons why QUINCY is unable to simulate drought and drought legacy. Exploring the processes behind the observations is interesting but this should be framed from the perspective of model evaluation and used to synthesize our understanding of QUINCY's performance.

Reasons for QUINCY not to capture the drought legacy could be caused by:
1) inaccuracies in the simulated soil moisture
2) the drought response function of photosynthesis and soil moisture response of soil respiration are incorrect
3) the non-structural carbohydrate pool sizes are not correct

There are two ways in which the drought can affect the photosynthesis in QUINCY. Leaf stomatal control can be limited by air or soil moisture, and in addition there is a direct limitation to reduce the photosynthesis levels by soil water potential. This way of accounting for drought effects in the terrestrial biosphere models is common, although it has been criticized (Sabot et al., 2022).

To reveal these effects, we compared the simulated soil moisture more closely to the observed soil moisture at different depths. At 5 cm depth, which is the most influential for the TER, the variation in soil moisture was not large enough (Fig. S13a), although this is not as strongly seen at other depths. When plotting GPP and TER against soil moisture in summer (Fig. S13b-c), no clear relationship was seen in the observations, but for the modelled values we see some decrease near the lower end of the soil moisture values. Soil water potential did begin to reduce GPP (Fig. S12), but the magnitude was not great enough.

Finally, we checked the simulated NSC pool sizes in the model in years 2007 and 2008 against normal years to see if there could potentially be an effect and also assessed if there was an effect on the simulated LAI values. We have added the following text in lines 432-435 this: "*The non-structural carbohydrate pools were affected by the drought. The labile pool was 25 % lower in 2007 than in normal years, but had fully recovered by 2008. The reserve pool was in 2007 at 18 % lower in 2007 than in normal years and was still 10 % lower in 2008. However, this did not significantly affect the LAI values or the annual GPP levels.*"

We also thank the reviewer for bringing these studies NSC-related to our attention, and have added a reference to the Jones paper.

4. Table S1 is referred to throughout the manuscript at greater frequency (12 times by my count) than every figure and table in the manuscript except Figure 4 which is also referenced 12 times. Alone, it is referred to more times than all three of the tables included in the manuscript! I would suggest moving this table to the manuscript or otherwise incorporating this information in a manner that does not result in the jarring requirement of frequently referring to the supplement.

We thank the reviewer for this comment. We've moved Table S1 to the main text (now Table 3), as suggested.

5. Figure 2 shows the mean yearly values for NEE, TER, soil temperature and soil moisture. This is discussed in detail. My feeling, considering the soil temperature plot and the location of Borden Forest, is that snow cover may be playing a substantial role at this site over winter. Note that the observed soil temperature maintains a constant temperature through winter, consistent with an insulating snow cover. This may also explain the model's earlier TER response as QUINCY fails to simulate any snowmelt period suppressing soil respiration despite increasing temperatures (Teklemariam et al., 2009). Was this potential role of snow considered?

The reviewer has a good point about the snow cover. Soil temperatures are affected by the snow cover, and the springtime recovery of TER is also affected.

In the original simulations we did not have the snow module turned on (we did not have snow included in the original version of the model we started to work with). In order to evaluate its effects separately, we did not rerun all of our simulations. Instead, we ran a simulation with the CN:LAI&chl -version of the model with the snow module enabled, and now report the influence on the soil temperatures and carbon fluxes. The annual carbon balances from this simulation are added to Table 3, the annual carbon cycles together with snow depth are shown in Fig. S9 and the simulated snow depth with observations are shown in Fig. S10.

Technical Comments

1. L20: Delete "the" from before "ecosystems".

Thank you, deleted.

We have removed this sentence from the new version of the manuscript to clarify the structure of the Introduction, which was criticized by Reviewer 2.

3. L30: Delete "the" from before "vegetation functioning".

Thank you, deleted.

4. L33: Delete "the" from before "forests". Note this is a recurring issue as per comments 1 and 3. Since we are not discussing specific forests or ecosystems, there should not be a definite article "the" in these instances.

Thank you, deleted.

5. L51: Add "an" before "increase" and a "the" before "land carbon sink". Modify "changing of" to "change in".

Thank you, we have corrected the sentence.

Yes, we removed the definition from here.

7. L69: Delete the comma after "Borden Forest" or add a comma after "continuous data".

Thanks, comma deleted.

8. L78-79: This sentence needs to be reformulated to make sense.

Apologies for unclear structure. The original sentence was: "*In our research, the data are combined with a terrestrial biosphere model QUINCY (QUantifying Interactions between terrestrial Nutrient CYcles and the climate system) (Thum et al., 2019), which simulates fully coupled cycles of carbon, nitrogen, phosphorus of terrestrial ecosystems coupled representations of the surface and sub-surface budgets of water and energy.*"

The modified version in lines 87-89 is: "*In our research, the observational data are combined with a terrestrial biosphere model QUINCY (QUantifying Interactions between terrestrial Nutrient CYcles and the climate system) (Thum et al., 2019). QUINCY simulates fully coupled carbon, nitrogen, and phosphorus cycles of terrestrial ecosystems along with water and energy budgets.*"

Thanks, we removed the definition from here.

10. L83 and throughout: Make sure references are formatted correctly, namely the brackets.

Thanks, we've corrected this line and checked the manuscript thoroughly in this respect.

11. L125: GPP has already been defined.

Thanks, we removed its definition from here.

12. L163: The acronym QUINCY has already been defined.

That's right, we removed the definition from here.

13. L202: Define or explain "OCN".

We have added explanation: "*terrestrial biosphere model OCN*"

14. L228: Delete "an" from before "the slope".

Thanks, deleted.

15. L243: Add "and" between "humidity" and "wind".

Thanks, added.

16. L247: Was a single random year used repeatedly in the spin-up or was it a 500 year time series constructed by randomly selecting a year of [CO2] and meteorology for each of the 500 years?

Apologies for the unclear formulation. We have only used the observed meteorological data during the spinup. For the transient simulations starting in 1901 forwards we created a meteorological forcing file by randomly picking a year of the observed years, with the $CO_2$ concentration varying according to GCP (Friedlingstein et al., 2019). For the spinup we used the first 30 years of this forcing file repeatedly.

17. L249: What data was used for the meteorology between 1901 and 1996 in the transient runs? ERA-5 is mentioned earlier but this dataset begins in 1940.

We've only used the meteorology measured at the site in the spinup. We have clarified this in the text:

"*This was followed by a transient simulation starting in 1901, which was using meteorological data randomly picked from the observed meteorology.*"

18. L264: Change "showed" to "shown".

Thanks, corrected.

19. The model abbreviations are quite long and complicated – is there a way to condense them while maintaining the information?

Thanks for noting this. We decided to change the "C-only" to "C", as the simulation names will still keep their information content. We also changed "C-only,fix" to Cfix".

20. L272: As I understand it, LAI was a parameter that was calibrated and this can be seen by the improved seasonality of LAI in Fig 1c&d. Was there no capacity to correct the magnitude of LAI which is too high in summer and too low in winter?

We only modified the senescence parameter. The deciduous forests in QUINCY don't have leaves in winter, so the only way we could match the winter LAI values would be to additionally simulate evergreen conifers. We decided not to do this, as it would complicate the work considerably (also discussed in response to Comment 2.)

We don't have an easy way to change the summer maximum LAI in QUINCY. We could try to change the leaf to root ratio, so that more of the biomass allocation goes to roots instead of leaves. However, there were several reasons why we did not do this.

1) If we had lowered the LAI, the simulated GPP would have been even lower.
2) The other leaf level parameters would likely have been degraded by this change.
3) The continuously measured LAI was lower than other observations, likely influenced by the location of the sensors. Other estimates of LAI at the site were more in line with our estimate, and we therefore also thought that our simulation result would not be so far off.

We have added to the text in lines 548-550 : "*The winter LAI will remain overestimated in our current modelling setup because only deciduous forest is being simulated. Changing the allocation pattern in QUINCY would have allowed a reduction in simulated LAI, but this would have resulted in lower GPP values.*"

21. L286: Figure S2 shows 2013, not 2014.

Thanks, we've corrected the text to the year 2013.

22. L297: "The observations show more shallow decrease" - I think this should be "The observations show a more gradual decline in increasing GPP before the peak" or similar?

Thanks, corrected.

23. L304: "parameterizations is in" should be "parametrizations are in".

Thank you, we've made the correction.

24. L305: It may be worth clarifying that this refers to different simulations within the "C-only" model framework only, not between the "C-only" and "C-only,fix" model simulations.

Thanks, that's a good point. We modified the text in lines 324-326 to: "*The different C model simulations (i.e. C:orig, C:LAI and C:LAI&chl) using dynamic stoichiometry (i.e. not including simulations using fixed stoichiometry, Cfix) did not largely impact RMSE and r² values.*"

25. L309: Perhaps specify that the "more accurate representation" is better timing of senescence?

Sure, we modified the text in lines 330-331 to: "*This resulted in a more accurate representation with better timing of senescence of the observed seasonal cycles of GPP, LAI and Chl$_{Leaf}$ in the simulations.*"

26. Figure 1: GPP and LAI have already been defined, and "leaf chlorophyll" has not been defined as "Chl" but as "Chl Leaf". I also do not think that the colors of each model simulation need to be spelled out in the caption as the legend already provides this information. It should be "... leaf chlorophyll WHICH has been smoothed...".

Thank you for the careful check on the figure and the caption. We removed the definitions of GPP and LAI from the caption, corrected the y-axis labels for leaf chlorophyll, removed the explanations for the colors and made the mentioned language correction to the caption.

27. L313: Perhaps a column should be added to Table S1 containing the percentage figures of under/overestimation.

That's a good idea, thanks! It has now been added (now Table 3), and some references to these values in the text have been removed.

28. L316: Should it be "with only a 1.4% larger value"?

Yes, it should be, and we've corrected that now, thanks.

29. L320: I think this should refer to Table S1, not Table S2.

Thank you for noticing this, this has now been corrected.

30. L322: TER has already been defined.

Yes, we removed the definition from here.

31. L330: "Table 1" should be "Table S1".

Thanks, we have now made the reference to the correct table.

32. Figures S3, S4, and S5 are often used to illustrate points that are difficult to parse from the plots (namely referring to fluxes being over/underestimated in certain months). The plots are too noisy for me to clearly and easily identify these statements so I suggest a different method of plotting this data to illustrate the points made.

We thank the reviewer for pointing this out. We modified the figures to show the monthly values instead of the daily values to make the plots less noisy. We also added $r^2$-values for these plots as suggested by Reviewer #2.

33. L339: "The early season pattern observed in the simulated NEE is attributed to heterotrophic respiration". Is not the overestimation of early season NEE due to a GPP that is too low from DOY ~75, as well as TER being too high between DOY 50 and 100? This is what I read from Figure S6.

Sure, the reviewer is right, we have changed this text in lines 361-362 into: "*The early season pattern observed in the simulated NEE is attributed to too late onset of GPP and heterotrophic respiration...*"

34. L348: Missing bracket after "(Fig. 2."

Thanks, corrected.

35. L355: This sentence is not properly formatted.

Thanks. The original sentence was: "*The continuous observations of LAI provide values (3.78 ± 0.43 m2 m−2 in summer, averaged the over June-July period, along with the standard deviation).*"

The modified sentence in lines 387-388 is: "*The continuous observations of LAI provide values of 3.78 ± 0.43 m2 m−2 in summer (averaged over the June-July period, along with the standard deviation).*"

36. L365 and surrounding paragraph: This paragraph could be condensed. It is also unnecessary to specify (C-only:LAI&Chl) and (CN:LAI&Chl) after each reference to the models as it has already been stated that these are the models referred to from Line 318 onwards.

Sure. We removed the specifics about the simulation runs and shortened the paragraph, as suggested.

37. L376: This statement about SLA is disjointed from the rest of the text.

Thanks for noting this. We tried to accommodate it better to the text by moving it to lines 392-394 and modifying it to:

"*In the model the specific leaf area (SLA) is an important factor in converting the leaf biomass to LAI and is held constant. In the observations the SLA showed a dynamic change, with higher values (~303 cm $g^{-1}$) observed in the early season and a subsequent decrease to a summer value of 162 cm $g^{-1}$ within about one month.*"

38. L379: TER is discussed first in Figure 4 but in the plot, GPP comes first. I would ensure these align to improve readability.

Sure, we've made this change now.

39. Figure 4: In this instance, I think it might be better to have the observed and simulated on the same plots, with the different facets instead for the years – as it stands, the figure is better as a comparison between years than for assessing model performance, which I believe to be its main message.

We also wanted to show the behavior in the years 2007 and 2008 against the regular years. We have now modified the plot so that the measurements and observations are shown in the same plots and the regular year behaviour in thin lines.

40. Figure 4 and Figure S9: As in other plots, could the standard deviation be plotted as a shaded area around the observation means?

We have added the standard deviation to the line that is the average of several years in Figure 4 and the standard deviation to precipitation over several years in Figure S11 (Fig. S9 in the first version).

41. L392: Move the sentences about GPP to the next paragraph which discusses GPP more widely, rather than in this paragraph which is about TER. Section 3.4 in general could use some work to improve the flow of the section and maintain a narrative throughout instead of the current text which tends to jump between the discussed variables with little reason. Either discuss each variable in turn, or discuss each time period in turn.

Sure, we followed the instructions by the reviewer and moved sentences about GPP. Additionally we worked on improving the flow of this section and discussed each variable in turn, as suggested.

42. Table 4: The QUINCY LOS (GPP) is incorrect – it should be 146 days if calculated as the difference between the mean SOS and mean EOS as the other table elements seem to imply (or perhaps this is coincidence and it is actually the mean length of each individual growing season, in which case this should be clarified).

We thank the reviewer for such careful investigation of the values. We had added a wrong number to the table for the QUINCY SOS (GPP), it should have been 142. We have corrected this now in the table and text.

43. L417: "LAI based estimates" is misspelled.

Thanks, corrected.

44. L420: "takes place in average" should be "takes place on average".

Thanks, corrected.

45. L429: Clarify what is meant by "make use of different spring and autumn periods".

We have changed this (now in lines 459-461) into: "*The real forest with several species may be more resilient to different environmental conditions, and therefore different species may be able to benefit from different environmental conditions during the shoulder seasons.*"

46. L438: There appears to be a significant breakpoint in GPP around 2009. Are there any explanations for why this might be? Is this a recovery from the 2007 drought? Is there any potential reason why every year post 2010 has higher GPP than any year before? Why were the final 5 years removed as a test?

Gonsamo et al. (2015) examined trends in GPP at the site and found a significant trend in photosynthetically active radiation, which they attributed to cleaner air. They also saw significant trends in the length of the carbon uptake period (CUP) and the delay in the end of the carbon uptake period. The delay in fall may be related to different responses to environmental conditions by carbon uptake and respiration. We performed a simplified analysis of the CUP and also found a significant trend in the end of the CUP and length of the CUP in our model simulations for the same time period as Gonsamo did in their observations, but when we extended the analysis period to the end of 2018, the significant trend disappeared.

In the observations the significant increase in GPP occurs mostly in the summer months, with some increase also taking place in the fall (Table S3). If this change is caused by environmental conditions, such as increasing atmospheric $CO_2$ or air temperature, we should theoretically be able to capture this increase in our model.

The increase in site observations could be caused by demographic changes in the forest. There are tree seedlings on the forest floor, and as they grow, they may be able to photosynthesize more. Their influence would not be captured by the continuous LAI observations or our model.

One thing that we had not addressed in the manuscript is the role of nitrogen deposition. Our model input data shows a top value in 1995 at the site and then a continuous decrease until the end of our study period, 2018 (Fig. S17). This is in line with the sulphur deposition change that has been reported for the Egbert site, which is close to the Borden forest (Gonsamo et al., 2015). The increase in GPP could partly be caused by the recovery of the forest from ozone, nitrogen, and sulphur depositions, but it would be expected that this recovery is seen as increasing LAI, which is not seen in the observations to a large degree. Decreasing nitrogen and sulphur depositions might alter the soil pH, causing alterations to the nitrogen cycle. If we had leaf nitrogen observations over different time periods, we might be able to assess this. We have added these points to the new version of the manuscript, to lines 659-666:

"*The N and S deposition was found to reduce net primary production in southern Ontario (Aherne et al, 2013), where also the Borden forest is located. Decreasing S deposition was found to increase forest productivity in the northeastern U.S. region (Dalton et al, 2024, Phelan et al., 2024), which is close to the Borden forest. Decreasing S and N deposition will also affect the soil pH (Dalton et al., 2024), which can potentially affect the N dynamics. The N deposition can be useful for forests under N deficit (Horn et al., 2018), however according*

*to our model results, the Borden forest would not suffer from N deficiency. The recovery of S deposition may be one of the reasons for the increasing trend of GPP. Long-term S exposure could also reduce the drought resilience of sites (Dalton et al., 2024), which could be another explanation for why our model failed to capture the drought effects at this site.*"

Removal of the last five years was quite arbitrary. We have mentioned this now in the text.

47. L445: Why is LAI not plotted in Figure S12 if it is discussed here?

We have added LAI to plot Fig. S12.

48. L451: Can the differences in IAV be quantified somehow, for instance comparing the standard deviations?

Sure, we added in here the standard deviations from the Table 3 (earlier Table S1) and noticed an inaccuracy in the earlier formulation, as the standard deviations of GPP and TER are comparable in the observations.

49. L469: Incorrect parentheses for reference.

Thanks, corrected.

50. Section 4.1: I'd like to see more discussion here about how the continuous LAI measurements improved the model performance. What has been learnt from this study and how can these lessons be implemented in the model? Does the senescence parameter need to be modified in the standard implementation of the model? Do we need to test the model tuned with continuous LAI values at other sites?

For the standard implementation of the model, we have tested the parameters at several sites (against GPP observations) when deciding on the default phenological parameters for deciduous forests. In the current model development phase the QUINCY model developers use satellite observations at site level for FLUXNET sites as well as the benchmarking system iLamb for global scale evaluations. Therefore we'd not change parameters based on observations at one site. An on-going analysis based on flux tower and remote sensing data (Miinalainen et al., in preparation) shows that several other sites with the same plant functional type are overestimating both GPP and (satellite-based) LAI in October in the Northern Hemisphere, so the results from Borden are in-line with these estimates and with the uncertainties of phenological transition dates of remote sensing observations (Wang et al., 2024), are a valuable site-level verification for these results. We've added to the text in lines 514-519:

"*The parameterization of QUINCY relies often on several sites in order to perform successful large-scale simulations. An ongoing study evaluating the QUINCY seasonal cycles with flux tower and remote sensing data (T. Miinalainen, pers. comm.) has shown that a similar bias in autumn phenology for temperate broadleaf deciduous trees occurs at several other sites, so this PFT could benefit from parameter tuning. However, the satellite observations have*

*some issues in predicting the autumn phenology (Wang et al., 2024) and the long-term in-situ observations at the Borden site can act as a valuable verification resource.*"

51. Section 4.2: Again, I think there could be more discussion here about what the simulated ChlLeaf values can teach us regarding QUINCY and future improvements. What does it mean for the model simulations that Chl Leaf peaks early? What might this imply for QUINCY applied at other sites or globally? While not the objective in this study, what might be learnt if we did attempt to capture the timing of the maximum leaf chlorophyll? What would the tradeoffs be?

The peak in the observed LeafChl occurs around DOY 180-200 (Fig 1e). This is not seen in the $V_{c(max),25}$ and $J_{max,25}$ observations (Fig. 3). Our goal is to improve the photosynthesis, and although leaf chlorophyll is part of the photosynthesis calculation in the QUINCY model, the $V_{c(max),25}$ and $J_{max,25}$ values play a larger role, and $Leaf_{Chl}$ is considered relevant because of its association with these parameters. Increasing the $Leaf_{Chl}$ values for the time period visible in the observations (Fig 1e) would probably not improve our results. We could perform sensitivity tests with the canopy module of QUINCY to see the influence on GPP of forcing $Leaf_{Chl}$ to a higher level for this time period. Given the structure of the model, this influence would not be very large.

Previous studies using simpler approaches have shown a more pronounced influence of using leaf chlorophyll in modelling. Croft et al. (2015) replaced LAI with leaf chlorophyll in a light use efficiency (LUE) model and they obtained a better seasonal cycle of GPP with this change. The $r^2$ of daily GPP from the LUE model with LAI was 0.55, with leaf chlorophyll it was 0.65 and with canopy scaled chlorophyll it was 0.69.

In the study by Luo et al. (2018), the $Leaf_{Chl}$ was used directly to estimate $V_{cmax25}$ and $J_{max,25}$ and they found improvements in model performance, $r^2$ for GPP in 2013-2014 was increased from 0.84 to 0.91 from basic model formulation with this change. This study was conducted using the BEPS model, which is typically run with LAI as input.

Our study was partly inspired by these previous studies, as we now have a model that explicitly includes $Leaf_{Ch}$. Our modelling approach has a direct coupling of $Leaf_{Chl}$ to the nitrogen cycle modelling, so we find this variable very interesting. By including both carbon and nitrogen cycles, our approach is more constrained than the modelling approaches in the two previous studies. The effects we obtained were not as pronounced as in the two previous studies. One reason is that our model is constrained by many processes. One difference between our study and that of Luo et al. (2017) is that we don't have a direct linear relationship between the leaf chlorophyll and $V_{cmax25}$, as has been shown to be the case at the Borden site (Croft et al., 2015). Nevertheless, our comparison with the biochemical model parameters showed reasonable values and didn't require any changes to the formulation. However, the $Leaf_{Chl}$ formulation of QUINCY may benefit from a linear relationship between $Leaf_{Chl}$ and $V_{cmax25}$ and this is one of the issues being addressed in an ongoing study (Miinalainen, T., in preparation) comparing remotely sensed $Leaf_{Chl}$ with QUINCY estimates at several flux sites.

A recent study combining chlorophyll fluorescence observations with the $Leaf_{Chl}$ and biochemical model parameter observations at the Borden site shows that the different light

acclimation rates of $V_{cmax25}$ and $Leaf_{Chl}$ cause changes in their relationship during different seasons (Yu et al., 2024). We could test the influence of changing these relationships with the canopy version of the model, that is modelling only the leaf canopy, so this would be a potential avenue for further study.

We added these points to the text to lines 530-542: "*The observed $Chl_{Leaf}$ values exhibit elevated levels during the midsummer period, spanning approximately one month. In the modelling conducted, the objective was not to replicate this behaviour but rather to capture the average summertime level. The observed midsummer peak in $Chl_{Leaf}$ was not simultaneously accompanied by increases in the biochemical model parameters $V_{c(max),25}$ and $J_{max,25}$ (Fig. 3), which would have a stronger effect on the simulated photosynthesis than $Chl_{Leaf}$ alone. We could test the effect of increased midsummer $Chl_{Leaf}$ on photosynthesis using the canopy module of QUINCY, which calculates only the canopy part of the model. Given the structure of the model, this influence is likely to be small. Luo et al. (2018) used the fact that $Chl_{Leaf}$ and $V_{c(max),25}$ have a linear relationship at the Borden site to improve their model results for GPP and evapotranspiration. In QUINCY we currently do not have this linear relationship, although it has been observed in some studies (e.g. Qian et al. 2021). A further study with remote sensing data and more sites will explore the implementation of such a description of leaf N partitioning in QUINCY. At the Borden site the linear relationship between $Chl_{Leaf}$ and $V_{c(max),25}$ was found to be off due to their different rates of light acclimation (Yu et al., 2024). Once we have a linear relationship between $Chl_{Leaf}$ and $V_{c(max),25}$ implemented in QUINCY, we can use the canopy module to estimate what effect this would have on photosynthesis in our model.*"

52. L511: "observed transitions is more smooth" should be "observed transitions are more smooth".

Thanks, corrected.

53. L514: I'd change "the model used here" to "QUINCY".

Thanks, we did this change.

54. L515: Missing word between "Testing model performance" and "a TBM designed".

Thanks, corrected now to: "Testing the performance of a TBM designed…"

55. L570: "It occurs most pronounced" is not grammatically correct – replace with "It occurs most prominently" or "It is most pronounced".

Thanks, replaced to: "*It is most pronounced in the year following the drought, 2008 (Fig. S10e).*"

56. L571: "Based on the data available then…" reads as if referring to the 2008 drought due to the prior sentence. I'd recommend "Based on the data available to Lee et al. (1999), they found …" or similar.

Thanks, we used the formulation suggested by the reviewer.

57. L583: Delete comma after "exudates".

Thank you, comma is deleted.

58. L587: Typo in "whereas is was".

Thank you, corrected to "whereas it was."

59. L598: "QUINCY does simulate" should be "QUINCY does not simulate".

Thanks, corrected.

60. L600: There are quite a few statements throughout the manuscript similar to the sentence here: "One additional cause of model failure might be that the canopy light-saturation point does not reflect the observations, however, there is not robust evidence that this is the case." These require at least some explanation of what potential sources of evidence were explored and discounted.

The point we wanted to make here was that the responses to light might not be similar in the model because e.g. we don't have the understory saplings in our model set-up. Since we have made this point elsewhere in the manuscript, we remove this point from here and add discussion about the potential recovery of the forest from nitrogen and sulphur deposition, as discussed in point #46.

There was perhaps another such statement in line 363 where the reviewer was referring to : "*The majority of the simulated heterotrophic respiration originates from the uppermost soil layers*." We added here a further explanation: " *…as most of the organic material is located in these layers in the model.*"

61. L605: Add "an" before "impact on the N cycle".

Thanks, added.

62. L619: Add "in" after "cause a change".

Thanks, added.

63. L623: Add "an" before "unrealistically low value".

Thanks, added.

64. L630: Add "the" before "N saturated case".

Thanks, added.

65. L631: Delete "and" before "in line with".

Thanks, deleted.

66. L633: Delete "of" in Section 4.8 header.

Thanks, deleted.

67. L642: Add "A" before "rain gauge".

Thanks, added.

68. L648: Specify that the "long time spans" is referring to long time series of observations.

Thanks, we modified the sentence in lines 755-756 to: "*This work demonstrated the usefulness of using different data sources and the importance of observational long time series.*"

69. L648: Specify that the "use of leaf chlorophyll content and LAI" is in parametrizing the model.

Thanks, the sentence in lines 756-757 is now: "*The use of leaf chlorophyll content and LAI in parameterizing the model improved simulated GPP in the CN simulations.*"

70. L658: Change to "attributed to an increase in PAR which is not visible in the shortwave radiation forcing for QUINCY".

Thanks, changed.

71. L661: Delete "is" from before "paves the way".

Thanks, corrected.

References

Guo, J. S., Gear, L., Hultine, K. R., Koch, G. W., & Ogle, K. (2020). Non-structural carbohydrate dynamics associated with antecedent stem water potential and air temperature in a dominant desert shrub. Plant, Cell & Environment, 43(6), 1467–1483. https://doi.org/10.1111/pce.13749

Jones, S., Rowland, L., Cox, P., Hemming, D., Wiltshire, A., Williams, K., Parazoo, N. C., Liu, J., da Costa, A. C. L., Meir, P., Mencuccini, M., & Harper, A. B. (2020). The impact of a simple representation of non-structural carbohydrates on the simulated response of tropical forests to drought. Biogeosciences, 17(13), 3589–3612. https://doi.org/10.5194/bg-17-3589-2020

Teklemariam, T., Staebler, R. M., & Barr, A. G. (2009). Eight years of carbon dioxide exchange above a mixed forest at Borden, Ontario. Agricultural and Forest Meteorology, 149(11), 2040–2053. https://doi.org/10.1016/j.agrformet.2009.07.011

References

Croft, H., Chen, J., Froelich, N., Chen, B., and Staebler, R.: Seasonal controls of canopy chlorophyll content on forest carbon uptake: Implications for GPP modeling, Journal of Geophysical Research: Biogeosciences, 120, 1576–1586, 2015.

Croft, H., Chen, J. M., Luo, X., Bartlett, P., Chen, B., and Staebler, R. M.: Leaf chlorophyll content as a proxy for leaf photosynthetic capacity, Global change biology, 23, 3513–3524, 2017.

Friedlingstein, P., Jones, M. W., O'Sullivan, M., Andrew, R. M., Hauck, J., Peters, G. P., Peters, W., Pongratz, J., Sitch, S., Le Quéré, C., Bakker, D. C. E., Canadell, J. G., Ciais, P., Jackson, R. B., Anthoni, P., Barbero, L., Bastos, A., Bastrikov, V., Becker, M., Bopp, L., Buitenhuis, E., Chandra, N., Chevallier, F., Chini, L. P., Currie, K. I., Feely, R. A., Gehlen, M., Gilfillan, D., Gkritzalis, T., Goll, D. S., Gruber, N., Gutekunst, S., Harris, I., Haverd, V., Houghton, R. A., Hurtt, G., Ilyina, T., Jain, A. K., Joetzjer, E., Kaplan, J. O., Kato, E., Klein Goldewijk, K., Korsbakken, J. I., Landschützer, P., Lauvset, S. K., Lefèvre, N., Lenton, A., Lienert, S., Lombardozzi, D., Marland, G., McGuire, P. C., Melton, J. R., Metzl, N., Munro, D. R., Nabel, J. E. M. S., Nakaoka, S.-I., Neill, C., Omar, A. M., Ono, T., Peregon, A., Pierrot, D., Poulter, B., Rehder, G., Resplandy, L., Robertson, E., Rödenbeck, C., Séférian, R., Schwinger, J., Smith, N., Tans, P. P., Tian, H., Tilbrook, B., Tubiello, F. N., van der Werf, G. R., Wiltshire, A. J., and Zaehle, S.: Global Carbon Budget 2019, Earth Syst. Sci. Data, 11, 1783–1838, https://doi.org/10.5194/essd-11-1783-2019, 2019.

Sabot, M. E. B., De Kauwe, M. G., Pitman, A. J., Medlyn, B. E., Ellsworth, D. S., Martin-StPaul, N. K., et al. (2022). One stomatal model to rule them all? Toward improved representation of carbon and water exchange in global models. *Journal of Advances in Modeling Earth Systems*, 14, e2021MS002761. https://doi.org/10.1029/2021MS002761

Luo, X., Croft, H., Chen, J. M., Bartlett, P., Staebler, R., and Froelich, N.: Incorporating leaf chlorophyll content into a two-leaf terrestrial biosphere model for estimating carbon and water fluxes at a forest site, Agricultural and Forest Meteorology, 248, 156–168, 2018.

Miinalainen, T., Ojasalo, A., Zaehle, S., Croft, H., Aurela, M., Peltoniemi, M., Caldararu, S. and Thum, T. Evaluating a terrestrial biosphere model carbon and nitrogen cycles via leaf chlorophyll content and remote sensing observations, in preparation.

Qian, X., Liu, L., Croft, H., & Chen, J. (2021). Relationship between leaf maximum carboxylation rate and chlorophyll content preserved across 13 species. *Journal of Geophysical Research: Biogeosciences*, 126, e2020JG006076. https://doi.org/10.1029/2020JG006076

Yu, L., Luo, X., Croft, H., Rogers, C. A. & Chen, J. M. (2024) Seasonal variation in the relationship between leaf chlorophyll content and photosynthetic capacity. *Plant, Cell & Environment*, 47, 3953–3965. https://doi.org/10.1111/pce.14997

Wang, C., Yang, Y., Yin, G., Xie, Q., Xu, B., Verger, A., et al. (2024). Divergence in autumn phenology extracted from different satellite proxies reveals the timetable of leaf senescence over deciduous forests. *Geophysical Research Letters*, 51, e2023GL107346. https://doi.org/10.1029/2023GL107346

Borden paper review: Reply to Reviewer #2

(The comments by the reviewer are in magenta, the replies to the comments are in black, new text added or modified in the manuscript is written in italics.)

General comments

The study of Thum et al. investigates carbon and nitrogen interactions in the Ontario's Borden Forest Research Station, using in-situ measurements to parameterize the QUINCY model. It evaluates carbon flux simulations over 22 years, finding good alignment in some metrics like GPP but identifying key discrepancies in ecosystem respiration trends and legacy drought impacts, underscoring the need to improve TBMs.

The manuscript addresses an important topic: the representation of carbon and nitrogen interactions in TBM models. Overall, it is well-written, and the figures—despite a few editorial issues—are clear and effectively support the results presented. However, I found the study's objectives difficult to discern from the Abstract. The Introduction also requires substantial revision, as it sometimes lacks logical flow, and the paragraphs don't fully cohere. For instance, the first paragraph focuses on the importance of nitrogen, followed abruptly by a discussion on changes in the growing season due to warming, and then by a mention of the value of long-term observations for capturing anomalies (anomalies of what?). The Introduction feels like a series of loosely connected points without a clear narrative thread, which makes it challenging to understand the paper's aim until the objectives are listed in the final paragraph.

We thank the reviewer for finding the topic important and for all the detailed comments that we believe will improve the quality of the manuscript. We apologize for the shortcomings of the introduction and have improved its flow, following the reviewer's suggestions. We have removed the paragraph that was mentioning the long-term observations to detect anomalies. We added our aims to the abstract in lines #9-11 as:

"*Our goals are to assess the additional value of using $Chl_{Leaf}$ in the model parameterization, to study how well QUINCY can capture observed trends related to the carbon cycle at the site, and investigate how well the processes associated with a drought year and its legacy effects are captured by the model.*"

Regarding the content of the paper, there are two key aspects that should be addressed:

**1. Quality of Eddy Covariance Flux Post-Processing:**
The study relies heavily on estimates of GPP and TER derived from eddy covariance measurements, which are not directly measured values. It is essential to assess the quality of the partitioning and gap-filling methods used. A study from 2004 is referenced for both flux partitioning and gap filling, yet there is no description of this

approach, its advantages, or why it was chosen. Given that GPP estimates are highly sensitive to partitioning methods, it's critical to first establish a solid foundation, showing that the best possible gap-filled and partitioned fluxes were derived before making further interpretations about model structure or other underlying factors.

We thank the reviewer for this insight. We have added an explanation of the method developed by Barr et al. (2004) and would like to mention that this is the standard method for the Fluxnet-Canada sites (Pierrat et al., 2021) and that this method has also been used for this site in several previous publications. We have now added these points to the manuscript.

Post-processing eddy covariance data is a complex task involving many decisions along the way. Testing the various partitioning and gap-filling methods would be a study in itself. There are several papers from the site that have focused on the observations and their processing, providing all the specific details (e.g. Froelich et al., 2015). Our goal in this study was to use these data for model evaluation.

Studies comparing gap-filling methods have not found them to introduce large differences between them(Mahabbati et al., 2021). A study comparing partitioning methods found the influence on the annual balances to be less than 10 % (Desai et al., 2008). Other studies have found modest differences (Moffat et al., 2007).

Assessing what would be the best gap-filling and partitioning method is difficult because there is no independent measure of what the absolute truth is. This is why studies using synthetic data are useful, since then the 'truth' is known. We've participated in one such study, sharing our model results from other sites to such a study, and fully support these investigations (Vekuri et al., 2025). The field of studying partitioning methods is an exciting one with findings related to ecosystem functioning (Wohlfahrt and Galvagno, 2017) and applications of machine learning (Pastorello et al., 2020), but we consider the application of these different methods to be outside the scope of our study.

While the gap-filling and partitioning affect the annual balance values, the seasonality of these fluxes was more important for our purposes. If the annual balance values were shifted slightly, the conclusions of our study would not change. We have now added text (lines 678-679) to the discussion of the uncertainty introduced to the measurements by the gap-filling and partitioning methods:

"*Another source of uncertainty in the observations are the use of gapfilling (Mahabbati et al., 2021) and partitioning methods (Desai et al., 2008), which introduce uncertainty in the annual carbon balance estimates.*"

2. Snow Cover Effects:

Considering the site's geographic location, persistent winter snow cover is likely; if this is not the case, it should be explicitly mentioned. However, the manuscript does not address snow cover or its potential impact on carbon flux processes. This is especially relevant in the shoulder seasons, where the authors discuss discrepancies between modeled and observed fluxes. Factors such as snow effects, soil freezing

We thank the reviewer for bringing this point up. The original model simulations did not include snow (this was partly for "historical" reasons: QUINCY did not include snow when this work began). We agree with the reviewer that it's a relevant process at this site and have added a simulation with the snow. We show the annual carbon balances of these simulations in Table 3, the seasonal cycle of snow depth with the carbon fluxes (Fig. S9), and time series of observed vs. simulated snow depth (Fig. S10).

Below, I provide more specific comments on the manuscript.

Specific comments

Line 10: Please also report the RMSE (and not just r2) when reporting the model performance.

This has now been added.

Line 10: You mentioned how you parameterized the model but not how the model was improved.

Thanks for noticing this, the phrasing had been unclear with "*The improved model captured observed daily gross primary production (GPP) well.*"

We changed this (lines 13-14) into:
"The *model with the improved parameterization captured observed daily gross primary production (GPP) well.*"

Line 11: Would be interesting to know the magnitude of this increase

Sure, we added the magnitude of the trend here.

Line 11: NEE not defined yet

Thanks, definition added.

Line 35: Grammar ("are" missing)

Thanks, added.

Line 42: Would be important to add here why the current representation of N limitation of photosynthesis is not sufficient (since this is such a key aspect in this paper)

The original claim here wasn't that the representation of the N-limitation was insufficient, but that the models have different approaches that lead to different results. We have added some more text (lines 45-47) here describing these differences:

"*N limitation may directly affect photosynthesis rates or its effects may be buffered via different stoichiometric related implementations (Thomas et al., 2015) and the different hypotheses and parameter values related to N cycle processes lead to differences between models (Medlyn et al., 2015).*"

Line 43: For example, which responses?

We've added (lines 49-51) here: "*A model intercomparison study of five CMIP6 models showed a wide range of response in net primary productivity for increased atmospheric $CO_2$ and atmospheric N deposition (Davies-Barnard et al., 2020).*"

These aspects are important to be clarified to convince the readers why "it is paramount that the effects of N constraints on plant productivity are accurately simulated".

We agree with the reviewer and are thankful for these remarks.

Line 50 and 52: Do these observations overlap? Both in terms of time and space? How is the
increase in LAI and the decline in N related?

The study by Chen (2019) uses global remote sensing data from 1981 to 2016. The responses for central Europe show different trends in LAI development (Fig. 1 in Chen's study), mostly small or larger increases. Jonard et al. (2015) found a decrease in leaf N in the European forests from 1992 to 2009, so the studies do overlap in time. The Jonard et al. (2015) study also included leaf mass observations, and for some tree species, significant increasing trends were included. They conclude that increases in tree productivity have led to increased tree nutrient demand, and as soil nutrient supply is insufficient to meet this demand, the tree mineral nutrition is deteriorating. Mason et al. (2022) have reported a decrease in N availability in many terrestrial ecosystems, which also supports the conclusions of Jonard et al. (2015).

We have added this point to the text (lines 58-60) by:

"*Climate change-induced changes led to increases in leaf area index (LAI) (Chen et al., 2019) and tree productivity (Jonard et al., 2015), and the changes in N availability and demand have led to declining N availability relative to demand in terrestrial ecosystems (Mason et al., 2022).*"

Line 79: Grammar

Thank you for noticing this. We have rephrased this sentence as the reviewer #1 also commented on this.

Section 2.1: Please add a description of the understory vegetation (species and cover). This information is relevant to your discussions of model limitations that we read later in the paper.

Sure. Unfortunately the exact species haven't been determined at the site to our knowledge. We added to the section the following text in lines 114-115: "*The understory consists of short ferns, small shrubs and saplings (Halliday, 2010).*"

Lines 101-104: Is this the species composition within the flux footprint?

There is cropland in the northwest direction and these data were excluded and gapfilled. This was explained in Section 2.2.1.

Line 107: Mean over what period?

Mean over 2000-2014, added .

Line 118: Used for what? For calibration of the model? For validation?

The $CO_2$ fluxes were used for model evaluation. The modified sentence is now in line 128-129: "*$CO_2$ flux data from half-hourly eddy covariance measurements sampled at Borden Forest tower at 33 m height between 1996 and 2018 were used for model evaluation.*"

Line 126: A brief description of the gapfilling and partitioning method should be given here and the justification why such method is selected over other existing methods.

We have added in here the point that this is the standard method used by Fluxnet-Canada (Pierrat et al., 2021) and that it has been used in previous studies at this site. We have added a further explanation of the method in lines 138-143:

"*This procedure first derives the component fluxes from the NEE and then uses simple empirical models constrained by the measured data (Barr et al., 2024; Rogers, 2024). The other empirical relationship is between TER and soil temperature at shallow depth, and the other is between GPP and photosynthetically active radiation (PAR) above the canopy, including also a time-varying parameter (Barr et al., 2024; Rogers, 2024). These time-varying parameters are determined using a flexible moving window approach, including 100 points (Barr et al., 2024; Rogers, 2024). The exact formulations used are as in Rogers (2024).*"

According to Barr et al 2004 "seasonal onset, rise and fall of photosynthesis from the FNEP time series based on the parameter Px in the rectangular hyperbolic model "was this the case too in this study?

No, here a bit of a different formulation for this equation was used. We have added a reference to the PhD thesis of C. Rogers, where this is clearly shown.

Line 128: Grammar

Thanks, we changed the original sentence ("*measured from instruments on the flux tower*") to: "*measured by instruments on the flux tower*".

Line 133: How exactly was this scaling done?

The scaling was very simple. The difference between the annual precipitation from the Egbert weather station (for years 2014-2018 that had the hourly data) and the ERA5-Land product was estimated for each month separately and this scalar was used to multiply all the values in the ERA5-Land dataset. We now explain this in Section S1.

How well do the ERA5-Land precipitation product and measured precipitation at the site compare? Perhaps a comparison can be added to the supplementary.

There is a large difference in the annual values. We have added a table showing the differences in S1.

Line 193: Typo

Thanks, corrected 'metres' to 'meters'.

The text in line 210 is "*The soil profile consists of 15 layers, reaching a depth of 9.5 metres. The depth of each layer layer increases exponentially as it*"

Line 274: Based on what was this level of T selected for the adjustments?

The selection was based on the best match of the simulated LAI to the observed LAI.

The original sentence was: "*To adjust the seasonality of LAI, the parameter controlling leaf senescence ($t_{air}^{sen}$) was modified from the default value of 8.5 °C to 15.0 °C.*"

We modified this into in lines 294-296: "*The autumn decline of the simulated LAI was adjusted to match the observations by modifying the parameter controlling leaf senescence ($t_{air}^{sen}$) from the default value of 8.5 °C to 15.0 °C.*"

There is a typo just before the caption of Table 2.

The reviewer might be referring to a "t" that has appeared in the pdf-version of this file (or then the right typo was not found). It's not in the original file, but likely a result of the pdf conversion.

Figure S2 has a typo in the legend of panel b (should be LAI not GPP)

Thanks, corrected.

In Figure S2 the panels already show GPP or LAI. I would not repeat this again in the legend. Also, the dashed lines can be removed from the legend (to make it less crowded) and keep their description in the caption.

Thank you for these remarks, they have been taken into account in the modified figure.

Figure S3- 5 the axis label should be "modelled" to be consistent with the x-axis which says "Observations". The figure title is already mentioning which model was used, not needed to

mention this in the title and in the axis label.

Sure, that's a valid point, we have made this change.

Figure S3- 5 wouldn't it make more sense to display mean daily temperature as the third value rather than the number of the month? What is the reasoning to use month here? Maybe can briefly be added to the caption.

Also using the daily temperature as a color code would be a good idea here. Based on these comments and those of reviewer #1, we decided to show the monthly values instead of the daily values, to make the figures easier to interpret and to convey the message that we're using them to show that different parameterizations affect the monthly values.

Figure S3- 5 it is hard to judge quantitatively the comparison of different model performances. Maybe at least the r2 values can be displayed in the panels?

We agree with the reviewer. We have added the $r^2$-values in the panels, as suggested.

Figure S8 very hard to read the figure. Consider increasing the font please.

Apologies for the unclear plot. We have increased the font and improved its readability by shortening the titles of the subplots.

Line 339-345: Could this inaccuracy in modelling the soil temperature be because of the snow? What is the contribution of snow cover at this site? We see from colder sites that snow has an insulating effect on the soil that decouples its temperature from air temperature. If direct measurements are not available at the site perhaps you could explore available remote sensing products (e.g., MODIS/Terra (MOD10A2) and MODIS/AQUA (MYD10A2) (Hall and Riggs 2021) Snow Cover 8-Day L3 Global 500m SIN Grid, Version 6 dataset, which provides maximum snow cover extent at 8-day temporal resolution and 500m spatial resolution).

We ran an additional simulation including snow and also show a comparison of snow depth with observations (Fig. S10). The comparison with observations shows that snow does indeed play a role in the spring soil temperatures and that the model simulation is improved in this respect. This is in lines 383-385:

"*Snow in the simulations caused a slight delay in the increase of TER during spring (Fig. S9b), and the RMSE of TER during mid-March to mid-April (DOY 74-105) improved by 0.1 µmol m$^2$ s$^1$ for the 2005-2015 period.*"

We plan to investigate snow effects in more detail in a separate study including several boreal sites and using remotely sensed snow cover data (Nagler et al., 2022) and freeze-thaw data from SMOS (Rautiainen and Holmberg, 2023), extending the results of Böttcher et al. (in preparation).

Line 460-464: Here the Results are repeated. Instead, there should be a discussion of what underpins the observation that although modelled LAI is overestimated, modelled GPP in

summer is underestimated.

We shortened the reference to the Results here, as suggested. We added the point that the magnitude of the biochemical parameters is more influencing the differences between the C and CN-simulations than LAI alone.

Section 4.2 I suggest dividing parameters by structural from photosynthetic traits.

Section 4.2 discusses leaf chlorophyll and specific leaf area (SLA). SLA can be considered to be a structural trait, because it influences the LAI. Since this section only discusses these two parameters in subsequent paragraphs, it was a bit unclear how to further divide them if we don't aim for very short sections.

Line 491-493: What explains it if the SLA was overestimated but GPP underestimated?

SLA changes the leaf carbon pool to LAI. The simulated LAI for the site is too high when compared to the continuous observations, but when compared to the in situ observations, it is close to the observed values. Although LAI is important in calculation of GPP, Figures 1 and 2 show that GPP increases in the C simulations the compared to the CN-simulations (25 % higher annual values, Table 3) due to the high leaf N content, not only due to LAI (10 % higher, Table 4).

The underestimation of GPP is not pronounced in the early years of the time period, but becomes more pronounced in the later years, contributing to the 17 % underestimation in the annual values (Table 3), which is still within the measurement uncertainty. If we consider only the first five years, the annual GPP balance is estimated to be 1367 g C m$^{-2}$ in the observations and 1222 g C m$^{-2}$ in the simulations, an underestimation of 11 %.

Line 500: "instead, we are estimating the tree traits per average individual for a deciduous forest." Not clear to me what this statement means. Because the methods section (2.2.3) only mentions "we used a species-weighted canopy average of the leaf-level parameters, based on the species composition of the forest „ and is not clear how the parameters are weighted and then aggregated (?). Adding a mathematical description here would be very helpful.

We apologize for the unclear expression in Section 2.2.3. We're only having traits for one tree in the model, that is representing the whole forest.

The earlier unclear sentence was:
"*Our modelling approach does not allow for species separation; instead, we are estimating the tree traits per average individual for a deciduous forest.*"

We re-worded the unclear sentence to (lines 553-554):
"*Since we do not have the ability to model different tree species, we model a deciduous forest composed of trees with identical traits.*"

We've now added in a table of the values for different species in the supplemental material, as well as an equation showing how we calculated the species-weighted average (Section

S2). This is the same approach that has been used in previous studies using these data (Croft et al., 2015, Luo et al., 2018). We didn't really aggregate the values, but we did smooth the lines for plotting purposes. The only purpose we used these lines was to delay leaf chlorophyll development. For comparison of other traits we only used summer averages.

Line 510: Which drought occurrences?

We refer here to drought periods taking place in late summer. We modified the text in lines 564-565 to:

"*The increase in the simulations is more abrupt to the summer levels and decline from early summer values occurs quite early, probably due to dry periods occurring during summer.*"

Line 515-517: Grammar check and re-writing needed. Sentence is not clear.

The original sentence was: "*Testing model performance a TBM designed for large-scale simulation at site-level is challenging as the model necessarily needs to apply generalizations in process representation in order to have a model that can be applied across sites and at large scales, due to limited knowledge and data needed for large-scale parameterization.*"

We have modified this in lines 569-571 into: "*Testing the performance of a TBM designed for large-scale, site-level simulation is challenging. The model must necessarily make generalizations in the process representation due to the limited knowledge and data required for large-scale parameterization.*"

Line 525: "The tree species composition has undergone changes at the site during our study period, e.g., the red maple was reported to have coverage of 36 % in 1995 (Lee et al., 1999) and 52 % in 2006 (Teklemariam et al., 2009). The impacts that these changes in the tree composition have on the carbon fluxes could be studied by a demographic model with sufficient granularity in the description of tree functional diversity"
Since species compositional shift was not really addressed in this study would suggest to remove this context from the Introduction as currently it reads as if this is one of the aspects this paper addresses.

We did not find a statement regarding to this in the Introduction, but it was prominent in the Abstract with the line: "*However, how carbon and nitrogen interactions affect both carbon fluxes and plant functional traits in dynamic ecotones, which are experiencing disturbance and species compositional shifts remains unclear.*" and perhaps this was what the reviewer had in mind.

We have modified this into: "*However, how carbon and nitrogen interactions affect both carbon fluxes and plant functional traits in dynamic ecotones, which are experiencing biotic and abiotic changes remains unclear.*"

Section 4.4 it is not clear if the model fails to simulate the (legacy) effect of drought because of its structure (representation of the carbohydrate pools) or the lack of precise soil moistures

estimations which directly affect modelled CO2 fluxes. Could this not have been specifically tested (for this particular case of drought conditions) by first calibrating the model using observed soil moisture measurements and testing whether GPP simulations improved during drought? If the model´s limitations during drought is a focus of this paper it deserves a more systematic approach to test this.

We agree with the reviewer that this topic would benefit from a deeper dive into it. However, calibrating the model by using observed soil moisture measurements is not straightforward. Instead, we conducted a more thorough analysis to determine whether the lack of drought effect is caused due to: inadequate description of soil moisture, incorrect drought response of the carbon fluxes (Fig. S13) or insufficient influence of drought in the non-structural carbohydrate pool.

Soil moisture in the top layer does not reach as low values as observed (Fig. S13a) and this may contribute to the problem. However, the issue is not as severe in the lower soil layers.

The non-structural carbohydrate pools were affected by drought in the model, but perhaps the effect was not large enough. However, the observed LAI did not decrease in the observations, so it is likely that the legacy effect seen in TER is not driven by changes in GPP, but by some other process that we do not have in our model. We have also discussed this in reply #3 to reviewer 1.

Line 556: What is meant here? Isn't a depth-resolved soil texture provided to the model? Otherwise, what description of the soil physics exactly is lacking here?

We don't have a depth-resolved soil texture in the model, we're referring here to the need to change the model structure. Without a deeper analysis, it is hard to say if we'd need to change the water-retention curve, infiltration properties, pedotransfer functions or something else.

We apologize for unclear formulation and have changed the sentence in lines 610-613 into: "*The drought response at the site could potentially be improved by calibrating the soil moisture response functions in the model, but probably some structural changes in the description of soil physics would also be required. These changes may involve modifications in water-retention curve, pedotransfer functions (Weber et al., 2024) or infiltration properties (Vereecken et al., 2019).*"

Line 570-576: Yes, this hysteresis in TER response to temperature (early season lower respiration than late season) could reflect the seasonal patterns of photosynthate allocation to roots. Tree girdling studies have shown that seasonal pattern of below-ground C allocation may be more important than soil temperature in determining root respiration (see for example Högberg et al. 2003) and that earlier in the season respiration from the soil is mostly due to heterotrophic respiration.

We thank the reviewer for this point and have added the mentioned reference to the text.

Line 591: Increase over time? Sentence reads incomplete.

Apologies for the incomplete sentence and thanks for noticing it. The original sentence was: "*Froelich et al. (2015) found a significant increase in summertime GPP and Gonsamo et al. (2015) significant increase in carbon uptake.*"

We have modified this in lines 648-649 into: "*Froelich et al. (2015) found a significant increase in summertime GPP and Gonsamo et al. (2015) significant increase in carbon uptake between 1996 and 2012.*"

Line 597: So the PAR that was used as an input to the model was not correct? The model would clearly use PAR, so why give it the wrong forcing?

We have used shortwave radiation, not PAR, as in input to the model, and these two variables were measured by different sensors. The trend in PAR reported in the Gonsamo et al. (2015) was significant, as was stated in the paper. However, when we looked more closely at the PAR dataset (we did this for the PAR available from the AmeriFlux database, not the gapfilled data from the Gonsamo study), we found that the increasing trend was less than 1 % per year for the time period of their study (1996-2012). Such an increase in model forcing would not lead to such strong trends in GPP as seen in the observations. Furthermore, when we calculated the trend for our study period (1996-2018), the trend was no longer significant.

In the forcing that we used, the shortwave radiation, we instead had a small decreasing trend. Because the trends in these radiation variables are so small, we wouldn't expect them to be the cause of large increasing trends in GPP. Instead, we hypothesize in the current version of the manuscript that the role of the understory with potential other effects, e.g. such as decreasing nitrogen and sulphur deposition rates (see lines 659-666 and response #46 to reviewer #1).

Line 603: You mean at this site? It is not clear how these findings are relevant to this study.

Apologies for unclear impressions. The study by Gonsamo et al. has been done at two sites (Harvard and Borden) and these conclusions were true for both sites. We have now clarified in the text, that these results are for the Borden sites.

Line 600-607: The discussion on the ozone effect comes out of blue here, unless my previous comment is clarified.

Yes, thanks, we hope the new version of the text makes it clearer in this aspect.

Lien 614: What is meant by "differences between the annual GPP and carbon balance"?

We apologize for the unclear statement. We've corrected this in lines 681-684 to:

"*It is interesting to note that  the differences between the measured and simulated annual GPP are not apparent in the early years of the record, but emerge in the later years (Fig. S16).*"

Line 619 and 620: Grammar

Thank you for noting this. The original sentence was: "*Including the nitrogen cycle in the simulations did not cause a change the net carbon balance of the ecosystem, …*"

We corrected this to: "*Including the nitrogen cycle in the simulations did not cause a change in the net carbon balance of the ecosystem, …*"

Section 4.8: I would not finish the manuscript on listing technical shortcomings. What have we learnt from long term ecological response of such an ecosystem, observations and model results combined? And what is the outlook under further changes in the climate? (e.g., predicted potential increase in temperature and dryness).

The idea of section 4.8 was not to list technical shortcomings (we have Section 4.3 for that), but to describe what kind of new observations would be useful to better understand the processes at the site.

However, following the reviewer's idea, we have added a new section 4.9, that discusses the points mentioned by the reviewer.

The Conclusion section can be shortened.

We've shortened the Conclusions by one paragraph.

References

Hall, D. K. and G. A. Riggs. MODIS/Terra Snow Cover 8-Day L3 Global 500m SIN Grid, Version 6. Distributed by NASA National Snow and Ice Data Center Distributed Active Archive Center (2021).

Högberg, P., Nordgren, A., Buchmann, N. et al. Large-scale forest girdling shows that current photosynthesis drives soil respiration. Nature 411, 789–792 (2001). https://doi.org/10.1038/350810585

References

Böttcher, K. Thum, T., Aurela, M., Rautiainen, K., Holmberg, M., Johnson, B., Koponen, S., Plummer, S. & Pulliainen, J. 2024. Influence of cryosphere dynamics on carbon fluxes in needleleaf boreal forest. In preparation for Remote Sensing of Environment.

Chen, J.M., Ju, W., Ciais, P. *et al.* Vegetation structural change since 1981 significantly enhanced the terrestrial carbon sink. *Nat Commun* **10**, 4259 (2019). https://doi.org/10.1038/s41467-019-12257-8

Croft, H., Chen, J., Froelich, N., Chen, B., and Staebler, R.: Seasonal controls of canopy chlorophyll content on forest carbon uptake: Implications for GPP modeling, Journal of Geophysical Research: Biogeosciences, 120, 1576–1586, 2015.

Davies-Barnard, T., Meyerholt, J., Zaehle, S., Friedlingstein, P., Brovkin, V., Fan, Y., Fisher, R. A., Jones, C. D., Lee, H., Peano, D., Smith, B., Wårlind, D., and Wiltshire, A. J.: Nitrogen cycling in CMIP6 land surface models: progress and limitations, Biogeosciences, 17, 5129–5148, https://doi.org/10.5194/bg-17-5129-2020, 2020.

Froelich, N., Croft, H., Chen, J. M., Gonsamo, A., and Staebler, R. M.: Trends of carbon fluxes and climate over a mixed temperate–boreal transition forest in southern Ontario, Canada, Agricultural and Forest Meteorology, 211, 72–84, 2015.

Halliday, M., 2010. Correlation between sonic anemometers at three heights within a mixed temperate forest. SURG Journal 3, 2291-1367, doi: 10.21083/surg.v3i2.1106.

Luo, X., Croft, H., Chen, J. M., Bartlett, P., Staebler, R., and Froelich, N.: Incorporating leaf chlorophyll content into a two-leaf terrestrial biosphere model for estimating carbon and water fluxes at a forest site, Agricultural and Forest Meteorology, 248, 156–168, 2018.

Jonard, M., Fürst, A., Verstraeten, A., Thimonier, A., Timmermann, V., Potočić, N., Waldner, P., Benham, S., Hansen, K., Merilä, P., Ponette, Q., de la Cruz, A.C., Roskams, P., Nicolas, M., Croisé, L., Ingerslev, M., Matteucci, G., Decinti, B., Bascietto, M. and Rautio, P. (2015), Tree mineral nutrition is deteriorating in Europe. Glob Change Biol, 21: 418-430. https://doi.org/10.1111/gcb.12657

Mahabbati, A., Beringer, J., Leopold, M., McHugh, I., Cleverly, J., Isaac, P., and Izady, A.: A comparison of gap-filling algorithms for eddy covariance fluxes and their drivers, Geosci. Instrum. Method. Data Syst., 10, 123–140, https://doi.org/10.5194/gi-10-123-2021, 2021.

Mason, R. E., Craine, J. M., Lany, N. K., Jonard, M., Ollinger, S. V., Groffman, P. M., Fulweiler, R. W., Angerer, J., Read, Q. D., Reich, P. B., Templer, P. H., and Elmore, A. J.: Evidence, causes, and consequences of declining nitrogen availability in terrestrial ecosystems, Science, 376, eabh3767, https://doi.org/10.1126/science.abh3767, https://www.science.org/doi/10.1126/science.abh3767, 2022.

Medlyn, B., Zaehle, S., De Kauwe, M. *et al.* Using ecosystem experiments to improve vegetation models. *Nature Clim Change* **5**, 528–534 (2015). https://doi.org/10.1038/nclimate2621

Moffat, A. M., Dario Papale, Markus Reichstein, David Y. Hollinger, Andrew D. Richardson, Alan G. Barr, Clemens Beckstein, Bobby H. Braswell, Galina Churkina, Ankur R. Desai, Eva Falge, Jeffrey H. Gove, Martin Heimann, Dafeng Hui, Andrew J. Jarvis, Jens Kattge, Asko Noormets, Vanessa J. Stauch, Comprehensive comparison of gap-filling techniques for eddy covariance net carbon fluxes, Agricultural and Forest Meteorology, Volume 147, Issues 3–4, 2007, https://doi.org/10.1016/j.agrformet.2007.08.011

Nagler, T., Schwaizer, G., Mölg, N., Keuris, L., Hetzenecker, M., & Metsämäki, S., 2022. ESA Snow Climate Change Initiative (Snow_cci): Daily global Snow Cover Fraction - snow on the ground (SCFG) from MODIS (2000-2020), version 2.0. In: NERC EDS Centre for Environmental Data Analysis

Pierrat, Z., Nehemy, M. F., Roy, A., Magney, T., Parazoo, N. C., Laroque, C., et al. (2021). Tower-based remote sensing reveals mechanisms behind a two-phased spring transition in a mixed-species boreal forest. _Journal of Geophysical Research: Biogeosciences_, 126, e2020JG006191. https://doi.org/10.1029/2020JG006191

Rautiainen, K., & Holmberg, M., 2023. SMOS Freeze and Thaw Processing and Dissemination Service, Algorithm Theoretical Baseline Document, ESRIN Contract Nro: 4000124500/18/I-EF. In (p. 18): Finnish Meteorological Institute (FMI)

Thomas, R.Q., Brookshire, E.N.J. and Gerber, S. (2015), Nitrogen limitation on land: how can it occur in Earth system models?. Glob Change Biol, 21: 1777-1793. https://doi.org/10.1111/gcb.12813

Tramontana G, Migliavacca M, Jung M, et al. Partitioning net carbon dioxide fluxes into photosynthesis and respiration using neural networks. _Glob Change Biol_. 2020; 26: 5235–5253. https://doi.org/10.1111/gcb.15203](https://doi.org/10.1111/gcb.15203

Vekuri, Henriikka and Tuovinen, Juha-Pekka and Kulmala, Liisa and Aurela, Mika and Thum, Tea and Liski, Jari and Lohila, Annalea, Improved Uncertainty Estimates for Eddy Covariance-Based Carbon Dioxide Balances Using Deep Ensembles. Available at SSRN: https://ssrn.com/abstract=4999973

Vereecken, H., Weihermüller, L., Assouline, S., Šimůnek, J., Verhoef, A., Herbst, M., Archer, N., Mohanty, B., Montzka, C., Vanderborght, J., Balsamo, G., Bechtold, M., Boone, A., Chadburn, S., Cuntz, M., Decharme, B., Ducharne, A., Ek, M., Garrigues, S., Goergen, K., Ingwersen, J., Kollet, S., Lawrence, D.M., Li, Q., Or, D., Swenson, S., de Vrese, P., Walko, R., Wu, Y. and Xue, Y. (2019), Infiltration from the Pedon to Global Grid Scales: An Overview and Outlook for Land Surface Modeling. Vadose Zone Journal, 18: 1-53 180191. https://doi.org/10.2136/vzj2018.10.0191

Weber, T. K. D., Weihermüller, L., Nemes, A., Bechtold, M., Degré, A., Diamantopoulos, E., Fatichi, S., Filipović, V., Gupta, S., Hohenbrink, T. L., Hirmas, D. R., Jackisch, C., de Jong van Lier, Q., Koestel, J., Lehmann, P., Marthews, T. R., Minasny, B., Pagel, H., van der Ploeg, M., Shojaeezadeh, S. A., Svane, S. F., Szabó, B., Vereecken, H., Verhoef, A., Young, M., Zeng, Y., Zhang, Y., and Bonetti, S.: Hydro-pedotransfer functions: a roadmap for future development, Hydrol. Earth Syst. Sci., 28, 3391–3433, https://doi.org/10.5194/hess-28-3391-2024, 2024.

Wohlfahrt, G. and Galvagno, M., 2017. Revisiting the choice of the driving temperature for eddy covariance CO2 flux partitioning, Agricultural and Forest Meteorology, 237–238, 135-142.

List of relevant changes

-aims clearly mentioned in the abstract
-introduction modified
-added section S1 (explaining the preparation of precipitation data from ERA before site observations were available) and S2 (the species composition, their respective values and how they were averaged)
-table S1 moved to the main text
-simulation with snow added (two related figures added to SI)
-Fig. 4 modified
-added a plot on N deposition to SI
-analysis and discussion about the drought response of the model added
-Section 4.9 discussing the research questions and future climatic changes at the site
-conclusions shortened